# Double Descent and Overfitting under Noisy Inputs and Distribution Shift for Linear Denoisers

**Chinmaya Kausik** *ckausik@umich.edu*
*Department of Mathematics*
*University of Michigan, Ann Arbor*

**Kashvi Srivastava** *kashvi@umich.edu*
*Department of Mathematics*
*University of Michigan, Ann Arbor*

**Rishi Sonthalia** *rsonthal@math.ucla.edu*
*Department of Mathematics*
*University of California, Los Angeles (UCLA)*

**Reviewed on OpenReview:** *https://openreview.net/forum?id=HxfqTdLIRF*

## Abstract

Despite the importance of denoising in modern machine learning and ample empirical work on supervised denoising, its theoretical understanding is still relatively scarce. One concern about studying supervised denoising is that one might not always have noiseless training data from the test distribution. It is more reasonable to have access to noiseless training data from a different dataset than the test dataset. Motivated by this, we study supervised denoising and noisy-input regression under distribution shift. We add three considerations to increase the applicability of our theoretical insights to real-life data and modern machine learning. First, while most past theoretical work assumes that the data covariance matrix is full-rank and well-conditioned, empirical studies have shown that real-life data is approximately low-rank. Thus, we assume that our data matrices are low-rank. Second, we drop independence assumptions on our data. Third, the rise in computational power and dimensionality of data have made it important to study non-classical regimes of learning. Thus, we work in the non-classical proportional regime, where data dimension $d$ and number of samples $N$ grow as $d/N = c + o(1)$.

For this setting, we derive data-dependent, instance specific expressions for the test error for both denoising and noisy-input regression, and study when overfitting the noise is benign, tempered or catastrophic. We show that the test error exhibits double descent under general distribution shift, providing insights for data augmentation and the role of noise as an implicit regularizer. We also perform experiments using real-life data, where we match the theoretical predictions with under 1% MSE error for low-rank data.

## 1 Introduction

Denoising and noisy-input problems have a rich history in machine learning and signal processing (Vincent et al., 2010a; Tian et al., 2020; Elad et al., 2023). Aside from its natural application to noisy input data, the idea of noise as a regularizer has led to denoising being tied to many areas of modern machine learning, such as pretraining and feature extraction (Krizhevsky et al., 2012), data-augmentation for representation learning Chen et al. (2020), and generative modeling (Rombach et al., 2022). While unsupervised methods like PCA (F.R.S., 1901) and low rank matrix recovery (Davenport & Romberg, 2016) have been addressed in prior theoretical work (Baldi & Hornik, 1989), *supervised* methods like denoising autoencoders are theoretically less well-understood.

One of the biggest practical qualms to studying a supervised setting is that a learner needs access to noiseless data sampled from the test distribution. This is resolved by considering *distribution shift*, which is when the training and test data can come from different distributions. As a toy example, one might have access to noiseless dog pictures containing mostly German shepherds and collies, but would want to denoise noisy dog pictures containing mostly pomeranians and poodles. As long as the training data "spans" the structured subset of features containing dog pictures, meaningful learning is possible. The same argument can be made for regression with noisy inputs, say for predicting click-rates for these pictures. Given this practical motivation, we study supervised denoising and noisy-input regression under distribution shift.

It is well understood that non-trivial denoising is made possible by the presence of additional structure in the data (see, for example, section 3.2 of Vincent et al. (2010a)). One of the most natural such structures is low rank, specifically the idea that the true inputs live in a low dimensional space. In fact, past work (Udell & Townsend, 2019) has demonstrated that *a lot of real-life data is approximately low-rank* – that is, its covariance matrix only has a few significant eigenvalues.

In this paper we look at the following error in variables regression task (Hirshberg, 2021). Let $X_{trn} \in \mathbb{R}^{d \times N}$ be a noiseless data matrix of $N$ samples in $d$ dimensional space. Let $\beta \in \mathbb{R}^{d \times k}$ be a target multivariate linear regressor, and let $Y_{trn} = \beta^T X_{trn}$. Note that when $\beta = I$, our target is the noiseless data $X_{trn}$ itself. Let $A_{trn} \in \mathbb{R}^{d \times N}$ be a $d \times N$ noise matrix. For this paper, we assume that we have access to $Y_{trn}$ and $X_{trn} + A_{trn}$ while training. The goal is to study the test error of the minimum norm linear function $W_{opt}$ that minimizes the *MSE training error*. The MSE training error is a natural choice – it is one of the most common targets for non-linear denoising autoencoders (Vincent et al., 2010a). We formalize the definition of $W_{opt}$ below.

$$W_{opt} = \arg\min_{W} \left\{ \|W\|_F^2 \,\Big|\, W \in \arg\min_{W} \|Y_{trn} - W(X_{trn} + A_{trn})\|_F^2 \right\}$$

Given test data $X_{tst} \in \mathbb{R}^{d \times N_{tst}}$ and $Y_{tst} = \beta^T X_{tst}$, we formally define the test error for arbitrary linear functions $W$ by $\mathcal{R}(W, X_{tst})$ below. Since we are not assuming anything about the distribution of the training or test data, we only take the expectation over the training and test noise. The error is dependent on the specific $X_{trn}$ and $X_{tst}$ that are given.

$$\mathcal{R}(W, X_{tst}) := \mathbb{E}_{A_{trn}, A_{tst}} \left[ \frac{\|Y_{tst} - W(X_{tst} + A_{tst})\|_F^2}{N_{tst}} \right]. \tag{1}$$

The classical theory of learning problems would keep the data dimension $d$ fixed and lets the number of samples $N$ grow to $\infty$. These can be theoretically analysed using elementary tools. However, with growing access to computational power and richness of data, it has become important to study *non-classical regimes*. One important and popular example is the proportional regime, where $d \propto N$ and so $d$ is comparable to N. Phenomena like double-descent and benign overfitting discovered in noiseless-input settings demonstrate the surprising advantages of this kind of high dimensionality. There has been a lot of recent work on whether overfitting output noise is beneficial or harmful in these regimes (Mallinar et al., 2022).

Testing on real data is a challenging issue. As argued above and in Cheng & Montanari (2022), a big reason for this issue is that past work assumes that the data covariance matrix is well-conditioned, while low-rank assumptions better model real-life data covariance matrices. In real life, one has little control over the independence or even the distribution of the data (Kirchler et al., 2023). There is also a growing need to be robust to adversarially chosen data in machine learning (Kotyan, 2023). Some recent work has dropped IID assumptions by looking at time series data (Nakakita & Imaizumi, 2022). Specifically, Nakakita & Imaizumi (2022), shows benign overfitting when the given data $x_1, \ldots, x_n$ come from a time dependent stochastic process. However, in many cases, we have dependent data, without the data forming a time series. We would thus like to drop the assumption that the data is IID or even independent without assuming a particular dependence structure. However, to account for this, we would need to have data-dependent theoretical results. We thus aim to address the following question:

> Q.1. Can we derive test error expressions for linear denoising and regression with noisy inputs that:
>   (a) work with data from a low-dimensional subspace under a non-classical regime,
>   (b) make minimal assumptions on the training data, test data and how they are related,
>   (c) match experiments that use real-life data distributions?
> Q.2. What insights can we obtain from these?

**Contributions.** The major contributions of the paper are as follows.

Q.1 (a) (**Low-Rank Data, Proportional Regime**) We work with low-rank data instead of needing full-rank data. This better models the approximate low-rank structure empirically seen in real-life data.

Q.1 (b) (**Dropping Independence Assumptions, Arbitrary Test Data**) We do not assume that the training data is IID or even independent, and we work with arbitrary test data from our low-dimensional subspace. Specifically, we obtain theoretical expressions that depend on the singular values of the given instantiation of the training and test data as well as the alignment between their left singular vectors.

Q.1 (c) (**Experiments using Real-Life Like Data**) We have a relative error of under 1% in experiments using real-life like data, establishing that our theoretical results can be used with data similar to real data.[1]

Q.2 (i) (**Double Descent and Overfitting Under Distribution Shift**) We show theoretically and empirically that our test error curves exhibit double descent, even for general distribution shifts. We relate this to the role of noise as an implicit regularizer. Further, we generalize ideas about overfitting to our setting and examine what conditions lead to benign, tempered, or catastrophic overfitting.

Q.2 (ii) (**Data Augmentation**) We examine when data augmentation improves in-distribution and out-of-distribution generalization, giving theoretical results and practical insights.

**Paper Structure.** Section 2 provides details and theoretical assumptions for the model we analyze. Section 3 discusses our main theoretical results for the denoising problem as well as experiments verifying these results. Section 4 discusses insights obtained from these results for double descent, data augmentation, and overfitting paradigms.

**Denoising.** Significant work has also been done to understand the role of noise as a regularizer and its impact on generalization (Bishop, 1995; Camuto et al., 2020; Hu et al., 2020; Neelakantan et al., 2015; Sonthalia & Nadakuditi, 2023). Other related works include the use of noise to learn features (Chen et al., 2020; Vincent et al., 2010b; Poole et al., 2014; Srivastava et al., 2014), to improve robustness (Ben-Tal et al., 2009; Rakin et al., 2019; Bertsimas et al., 2011), to prevent adversarial attacks (Chakraborty et al., 2021), and the connection to training dynamics (Pretorius et al., 2018). There has also been work to understand Bayes optimal denoiser using matrix factorization (Nadakuditi, 2014; Lelarge & Miolane, 2017; Lesieur et al., 2017; Maillard et al., 2022; Troiani et al., 2022). Finally, there has been some preliminary recent work on the learning dynamics of supervised denoising (Cui et al., 2021; Pretorius et al., 2018). *In this paper, we continue the study of understanding supervised denoising from the lens of test error.*

**Theoretical Work on Generalization in Non-Classical Regimes.** There has been significant work in understanding the generalization error and benign overfitting. Most prior work looks at the case where $x \sim \mathcal{D}$, for some data distribution $\mathcal{D}$ and we have $y = \beta_*^T x + \xi$ where $\xi \sim \mathcal{N}(0, \tau^2)$. The following least squares problem is commonly considered

$$\beta_{opt} = \arg\min_{\beta} \left\{ \|\hat{\beta}\|_F^2 \,\middle|\, \hat{\beta} \in \arg\min_{\hat{\beta}} \|Y_{trn} - \Xi_{trn} + \hat{\beta}^T X_{trn}\|_F^2 \right\}$$

The above setting was analyzed in Dobriban & Wager (2018); Mel & Ganguli (2021); Muthukumar et al. (2019); Bartlett et al. (2020); Belkin et al. (2020); Derezinski et al. (2020); Hastie et al. (2022); Loureiro et al. (2021b). The version for kernelized regression was studied in Mei et al. (2022); Mei & Montanari (2021); Mei et al. (2018); Tripuraneni et al. (2021a); Gerace et al. (2020); Woodworth et al. (2020); Loureiro et al. (2021a); Ghorbani et al. (2019; 2020).

---

[1]The code for the experiments can be found in the following anonymized repository [Link].

We note that our setup is similar to prior work. However, the major difference is the placement of the noise. That is, we have error in our independent variables $x$ while prior work assumes error in the dependent variables $y$. While standard error in variables (Hirshberg, 2021) assumes error in both, we simplify to only assume error in $x$.

The above prior work assumes full dimensional data. Generalization for low dimensional data is partially addressed by work on Principal Component Regression (Huang et al., 2022; Xu & Hsu, 2019), although they do not test on real data. These works use a PCA-based low-rank approximation of the data and provide theoretical results in terms of the effective rank or eigenvalue decay of the data covariance matrix. *In this paper, we continue the study of linear models and their test errors in non-classical regimes. However, we do so in the context of denoising and low rank data.*

**Testing on Real-Life Data.** Some past theoretical work in the high-dimensional regime tests on real data Paquette et al. (2022); Loureiro et al. (2021b;a); Wei et al. (2022). Loureiro et al. (2021b) assumes that the data distribution is given by a Gaussian mixture model. Loureiro et al. (2021a) uses a Gaussian equivalence principle to argue that their teacher-student model can capture realistic data assumptions. Finally, Wei et al. (2022) show that their appropriately rescaled training error asymptotically converges to the true risk under some assumptions. Cheng & Montanari (2022) also study a more general model of ill-conditioned covariances. *All* of the results require that the training data is *I.I.D.* and that the test data is from the *same distribution* as the training data. Further, they do not do this in the denoising setup considered in this paper. However, in contrast to us, they provide data-independent theoretical results.

**OOD Generalization and Distribution Shift.** Empirically, studies such as Miller et al. (2021); Recht et al. (2019); Miller et al. (2020); McCoy et al. (2020) have noticed that there is a strong linear correlation between in-distribution accuracy and out-of-distribution accuracy. There has been some theoretical work (Tripuraneni et al., 2021a;b; Mania & Sra, 2020; Wu & Xu, 2020; Darestani et al., 2021; LeJeune et al., 2022) that has looked into this phenomenon. In other examples, Ben-David et al. (2006) approaches this from a domain adaptation perspective and computes bounds dependent on an abstract notion of distribution distance; Lei et al. (2021) provides analysis for the minimax risk; and Lampinen & Ganguli (2019) studies the generalization when training models with gradient descent. *In this paper, we continue theoretically understanding distribution shift. We do so in the novel setting of denoising.* While past results usually only depend on some measure of distance between the two distributions. We provide instance-specific results that depend on the data.

**Double Descent and Implicit Regularization.** The generalization curves in our paper display double descent (Opper & Kinzel, 1996; Belkin et al., 2019), which is believed to occur due to implicit bias due to some form of regularization. There has been significant theoretical work on understanding implicit biases (Jacot et al., 2020; Shamir, 2022; Neyshabur, 2017) and gradient descent (Soudry et al., 2018; Ji & Telgarsky, 2019; Du et al., 2019a;b). *However, there is little work on double descent under distribution shift or for noisy inputs, which we establish in this work in relation to implicit regularization due to input noise.*

## 1.1 Limitations

While we provide new results, our work still has limitations. As with most prior work in the area, our work is limited to linear models. Extensions to shallow non-linear models for very different data assumptions have been considered Cui & Zdeborová (2023). However, extensions to deep networks with realistic data assumptions are still quite challenging. One possible way to extend to non-linear models is to study kernel methods. The kernel most closely associated with deep neural networks is the Neural Tangent Kernel (NTK). It has been shown that the feature vectors obtained from the NTK kernel are approximately low rank.

While we make progress toward more realistic data assumptions, another limitation is that the theory needs exactly low rank data. This is not quite true for real data, which is only *approximately* low rank or lives in some non-linear low dimensional space. Approximately low-rank data is difficult to deal with due to the ill-conditioning of the covariance matrix. Recent work such as Cheng & Montanari (2022); Wang et al. (2024) have started studying ill-conditioned covariance matrices.

Another limitation is our theoretical result for the over-parameterized regime has non-trivial error only if one assumes that $\|X_{trn}\|_F^2 = o(N_{trn})$. That is, the norm of the data points must decay as $N_{trn}$ grows. This is not a concern for many prior works (Hastie et al., 2022; Bartlett et al., 2020; Cheng & Montanari, 2022). However, we study the problem in a different setup.

## 2 Problem Setup and Notation

Recall the problem setup. Let $X_{trn} \in \mathbb{R}^{d \times N}$ be a noiseless data matrix of $N$ samples in $d$ dimensional space. Let $\beta \in \mathbb{R}^{d \times k}$ be a target multivariate linear regressor, and let $Y_{trn} = \beta^T X_{trn}$. Note that when $\beta = I$, our target is the noiseless data $X_{trn}$ itself. Let $A_{trn} \in \mathbb{R}^{d \times N}$ be a $d \times N$ noise matrix. For this paper, we assume that we have access to $Y_{trn}$ and $X_{trn} + A_{trn}$ while training. We then solve the following problem.

$$W_{opt} = \arg\min_{W} \left\{ \|W\|_F^2 \,\middle|\, W \in \arg\min_{W} \|Y_{trn} - W(X_{trn} + A_{trn})\|_F^2 \right\}$$

Given test data $X_{tst} \in \mathbb{R}^{d \times N_{tst}}$ and $Y_{tst} = \beta^T X_{tst}$, we have the following test error.

$$\mathcal{R}(W, X_{tst}) := \mathbb{E}_{A_{trn}, A_{tst}} \left[ \frac{\|Y_{tst} - W(X_{tst} + A_{tst})\|_F^2}{N_{tst}} \right].$$

We would like to study the test error $\mathcal{R}(W_{opt}, X_{tst})$ of $W_{opt}$ in terms of properties of the data matrices $X_{trn}$ and $X_{tst}$ as well as the noise distributions. All prior work besides Sonthalia & Nadakuditi (2023); Cui & Zdeborová (2023); Pretorius et al. (2018) only considers models where we have access to noisy outputs $Y$ and noiseless inputs $X$, while we consider the case where we have noisy inputs $X$. For simplicity, we assume access to noiseless outputs $Y$. It is easy to see from elementary linear algebra that $W_{opt} = Y_{trn}(X_{trn} + A_{trn})^{\dagger}$. Notice that when $\beta = I$, we are studying the linear denoising problem, and when $\beta \in \mathbb{R}^d$, we are studying real-valued regression with noisy inputs. We work in the *proportional regime*, where $d/N = c + o(1)$ as $N$ grows, for some constant $c > 0$. Notice that we are not *in* the limit $N \to \infty$; we will in fact bound the deviation from our estimates by $o(1/N)$ as $N$ grows. We now detail our assumptions on data and noise.

**Assumptions about the data.** The assumptions below formalize three natural requirements on the data – (1) that it lies in a low-dimensional subspace as argued above; (2) that the norm of the training data does not grow too much faster than the norm of the training noise, otherwise there will not be enough noise to train on; (3) that the training data "sees enough" of the subspace containing the data.

**Assumption 1.** *We have $d$-dimensional data $X_{trn} \in \mathbb{R}^{d \times N}$ and $X_{tst} \in \mathbb{R}^{d \times N_{tst}}$ so that*

1. *Low-rank: There is a fixed $r > 0$ so that $X_{trn}$ and $X_{tst}$ have data-points lying in an $r$-dimensional subspace $\mathcal{V} \subset \mathbb{R}^d$, and the column span of $X_{trn}$ is $\mathcal{V}$.*
2. *Data growth: $\|X_{trn}\|_F^2 = O(N)$.*
3. *Low-rank well-conditioning: For the $r$ singular values $\sigma_i$ of $X_{trn}$, $\frac{\sigma_j}{\sigma_i} = \Theta(1)$ and $\frac{1}{\sigma_i} = o(1)$ as $N$ grows, for any $i, j$.*

Notice that we don't assume that $X_{trn}$ is IID or even independent, and $X_{tst}$ is completely arbitrary, besides lying in the subspace $\mathcal{V}$. In our results, we will characterize the dependence of the error on $X_{trn}$ and $X_{tst}$ using their singular values. These intuitively measure "how much each direction is sampled," and don't depend on the distribution of the data. Finally, let $X_{trn} = U\Sigma_{trn}V_{trn}^T$ be the SVD of $X_{trn}$ with $U \in \mathbb{R}^{d \times r}$, $\Sigma_{trn} \in \mathbb{R}^{r \times r}$ and $V_{trn}^T \in \mathbb{R}^{r \times N}$. Note that the columns of $U$ span $\mathcal{V}$. Then there exists a matrix $L$ such that $X_{tst} = UL$. For Theorem 2, we will relax our assumption on $X_{tst}$ to say that there exists $L$ and $\alpha > 0$ so that $\|X_{tst} - UL\| < \alpha$.

**Comparison with assumptions in prior work.** Most prior works on theoretical regression assume that the data comes from a Gaussian or Gaussian-like distribution. Specifically, Tripuraneni et al. (2021a); Dobriban & Wager (2018); Mel & Ganguli (2021); Belkin et al. (2020); Mei & Montanari (2021); Huang et al. (2022); Xu & Hsu (2019); Tripuraneni et al. (2021b) assume that $x \sim \mathcal{N}(0, \Sigma)$. Most real data cannot be modeled as Gaussian data. Another common assumption is that $x = \Sigma^{1/2}z$ where the coordinates of $z$ are independent, centered, and have a variance of 1. This setting is a little bit more general than the previous

setting. The independence of data is still a limiting assumption that prevents it from modeling real-life data well. In addition, as the dimension increases, due to the (Lyapunov's) central limit theorem, the data's higher moments tend towards those of a Gaussian distribution again. This makes this assumption nearly as limiting as the first one. Papers with this (or very similar) assumption include Wu & Xu (2020); Bartlett et al. (2020); Hastie et al. (2022); Cheng & Montanari (2022). These papers have many important and seminal results and provide the foundation for our study.

We provide results on test error in a very different low-rank setting inspired by real-life data, and drop many restrictive assumptions. A small number of papers (Huang et al., 2022; Xu & Hsu, 2019; Sonthalia & Nadakuditi, 2023) that do assume a low-rank structure. However, the first two *further* assume that the data is low-rank Gaussian, while the third only provides results for one-dimensional data.

**Assumptions about the training noise.** We recall the definition of the Marchenko-Pastur distribution with shape $c$, for completeness.

**Definition 1.** *Let $c \in (0, \infty)$ be a shape paramter. Then the Marchenko-Pastur distribution with shape $c$ is the measure $\mu_c$ supported on $[c_-, c_+]$, where $c_\pm = (1 \pm \sqrt{c})^2$ is such that*

$$\mu_c = \begin{cases} \left(1 - \frac{1}{c}\right)\delta_0 + \nu & c > 1 \\ \nu & c \leq 1 \end{cases}$$

*where $\nu$ has density $d\nu(x) = \frac{1}{2\pi x c}\sqrt{(c_+ - x)(x - c_-)}$.*

Our assumptions on noise are fairly natural and general. Informally, we require the training noise to (1) have finite second moments, (2) be uncorrelated across entries, (3) be isotropic, and (4) follow a natural limit theorem. On the other hand, the test noise only needs (1) finite second moments and (2) uncorrelated entries. Our assumptions include a broad class of noise distributions (see Proposition 1 of (Sonthalia & Nadakuditi, 2023)). One of the many examples of noise distributions satisfying these is Gaussian noise, with each coordinate having variance $1/d$. Formally, we assume the following.

**Assumption 2.** *Let the train and test noise matrices $A_{trn}, A_{tst} \in \mathbb{R}^{d \times N}$ be sampled from distributions $\mathcal{D}_{trn}$ and $\mathcal{D}_{tst}$ such that $A_{trn}$ satisfies points $1-4$ below and $A_{tst}$ satisfies points $1, 2$.*

1. *For all $i, j$, $\mathbb{E}_{\mathcal{D}}[A_{ij}] = 0$, and $\mathbb{E}_{\mathcal{D}}[A_{ij}^2] = \eta^2/d$. Here $\eta = \Theta(1)$ as $N$ grows.*
2. *For all $\{i_1, j_1\} \neq \{i_2, j_2\}$, $\mathbb{E}_{\mathcal{D}}[A_{i_1 j_1} A_{i_2 j_2}] = \mathbb{E}_{\mathcal{D}}[A_{i_1 j_1}]\mathbb{E}_{\mathcal{D}}[A_{i_2 j_2}]$.*
3. *$\mathcal{D}$ is a rotationally bi-invariant distribution[2] and $A \sim \mathcal{D}$ is full rank with probability one.*
4. *Suppose $A^{d,N}$ is a sequence of matrices such that with $d/N = c + o(1)$ as $N$ grows, for $c > 0$. Let $\lambda_1^{d,N}, \ldots, \lambda_N^{d,N}$ be the eigenvalues of $(A^{d,N})^T A^{d,N}$. Let $\mu_{d,N} = \sum_i \delta_{\lambda_i^{d,N}}$ be the sum of dirac delta measures for the eigenvalues. Then we shall assume that $\mu_{d,N}$ converges weakly in probability to the Marchenko-Pastur measure with shape $c$ as $N$ grows.*

Assumption 2.4 may seem odd, but the Marchenko Pastur is the natural universal limiting distribution for many classes of random matrices.

**Comparison with assumptions in prior work.** There are three papers in denoising to compare to, namely Sonthalia & Nadakuditi (2023); Cui & Zdeborová (2023); Pretorius et al. (2018). Our assumptions on noise are strictly more general than the first two. However, the model for Cui et al. (2021) is more general than ours. Pretorius et al. (2018) has the same assumptions as ours, except that they do not require rotational invariance of noise. In contrast to our general closed form results, they analyse learning dynamics for denoising by choosing a specific orthogonal initialization for the coupled ODE that they derive.

**Terminology.** We introduce some terminology that we will use throughout the paper.

**Definition 2.** • *We call a linear model* overparametrized *if $d > N$ and* underparametrized *if $d < N$.*
• In-subspace *distributions refer to test data distributions whose support is in $\mathcal{V}$, while* out-of-subspace *distributions are test data distributions whose support is not contained in $\mathcal{V}$.*

---

[2]A distribution over matrices $A \in \mathbb{R}^{m \times n}$ is rotationally bi-invariant if for all orthogonal $U_1 \in \mathbb{R}^{m \times m}$ and all orthogonal $U_2 \in \mathbb{R}^{n \times n}$, $U_1 A U_2$ has the same distribution as $A$. Another way to phrase rotational bi-invariance is if the SVD of $A$ is given by $A = U_A \Sigma_A V_A^T$, then $U_A$ and $V_A$ are uniformly random orthogonal matrices and are independent of $\Sigma_A$ and each other.

- *A curve exhibits* double descent *if it has a local maximum or a peak.*
- *By* test error *we mean* $\mathcal{R}(W_{opt}, X_{tst})$, *and by* generalization error *we mean* $\mathbb{E}_{X_{tst}}[\mathcal{R}(W_{opt}, X_{tst})]$, *assuming that $X_{tst}$ is sampled (possibly dependently) from some distribution.*

We now define the overfitting paradigms that we will study. Motivated by past work on benign overfitting, we present a reasonable generalization of overfitting paradigms (benign, tempered and catastrophic, see Mallinar et al. (2022)) to our setting. Consider the minimum norm denoiser that minimizes *expected* MSE training error, similar in spirit to $\theta^*$ in Bartlett et al. (2020).

$$W^* = \arg\min_{W} \left\{ \|W\|_F^2 \middle| W \in \arg\min_{W} \mathbb{E}_{A_{trn}}[\|Y_{trn} - W(X_{trn} + A_{trn})\|_F^2] \right\}$$

Recall that we obtain $W_{opt}$ by instead minimizing the MSE error for a single noise instance $A_{trn}$, which means that $W_{opt}$ overfits $A_{trn}$ in the overparametrized regime. We would like to see if this overfitting is benign, tempered or catastrophic for test error. Following the definition of overfitting paradigms in Mallinar et al. (2022), we want to take $N \to \infty$. Since we are in the proportional regime, we must let $d \to \infty$ as well, maintaining the relation $d/N = c + o(1)$. For studying overfitting, a natural goal would be to study how the excess error $\mathcal{R}(W_{opt}, X_{tst}) - \mathcal{R}(W^*, X_{tst})$ behaves as $d, N \to \infty$. This is analogous to the excess risk studied in overfitting for noiseless inputs (Bartlett et al., 2020). However, we will see that both errors in our difference individually tend to zero as $d, N \to \infty$, making this a somewhat meaningless criterion. As noted in Shamir (2022), benign overfitting is traditionally restricted to scenarios where the minimum possible error is non-zero. A natural generalization to consider then is to instead study the limit of *relative excess error*

$$\frac{\mathcal{R}(W_{opt}, X_{tst}) - \mathcal{R}(W^*, X_{tst})}{\mathcal{R}(W^*, X_{tst})}$$

as $d, N \to \infty$ with $d/N = c + o(1)$.

**Definition 3.** *We say that overfitting is* benign *when this limit is* 0, tempered *when it is finite and positive, and* catastrophic *when it is* $\infty$.

## 2.1 Models Captured

In this section, we explore various scenarios captured by our assumptions, demonstrating the wide applicability of our theoretical results.

**Denoising Low Dimensional Data** The simplest setting is when $X$ lives in some low dimensional subspace $V$. Let $\beta = I$. Then, this models the least squares supervised denoising problem.

**Error in Variables Regression** Suppose $\beta \in \mathbb{R}^d$. Recall that $Y_{trn} = \beta^T X_{trn}$. In this case, the standard linear regression problem is to find $\hat{\beta}$ that minimizes $\|Y_{trn} - \hat{\beta}^T X_{trn}\|_F^2$. For error in variables regression, we have noise on the independent variable $X_{trn}$. Hence, we would be solving the following problem.

$$\|Y_{trn} - \hat{\beta}^T(X_{trn} + A_{trn})\|_F^2.$$

This situation satisfies our assumptions. Note that the situation extends to multivariate regressions, i.e., when $\beta \in \mathbb{R}^{d \times k}$ for $k > 1$.

**Remark 1.** *While this is not the full error-in-variables regression, our proof techniques can also cover cases with errors in $Y_{trn}$.*

**Gaussian Mixture Models Classification** Our assumptions also capture classification problems for Gaussian mixtures. Specifically, let $\mu_1, \ldots, \mu_k$ be the $k$ mean vectors. Let us also assume that the mean vectors are pairwise orthogonal. Let $z^i$ be the $i$th data point and assume that $z^i$ belongs to the $j$th cluster. Then we know that $z^i - \mu_j \sim \mathcal{N}(0, I)$.

Thus, we can see that we can write our data as $Z = X + A$, where $A$ is a Gaussian random matrix, $X$ is a low rank matrix whose columns live in $\{\mu_1, \ldots, \mu_k\}$. Hence, our data can be modeled at low rank (rank $k$) plus Gaussian noise. We also need to see that $Y$ can be modeled as $\beta^T X$ for some $\beta$. Let the label for the $j$th cluster be a $k$-dimensional vector with a one in the $j$th coordinate and let $\beta = [\mu_1 \ldots \mu_k]$, we get that $Y = \beta^T X$. Hence, we can model specific classification problems.

**Remark 2.** *The training and test metrics differ from typical classification settings, utilizing MSE instead of cross-entropy and accuracy, respectively. However, the two are related in many instances. See Muthukumar et al. (2021) for a more in-depth theoretical study of the two loss functions.*

**Spiked Covariance Models**  Another common model is the spiked covariance model. This model assumes that the covariance matrix for data has $K$ (much smaller than $N$ and $d$) large eigenvalues, and the rest of the eigenvalues are of smaller order (Ke et al., 2023). If we consider data $Z = X + A$, where the singular values of $X$ are much bigger than the singular values of $Z$, then we see that

$$\mathbb{E}[ZZ^T] = \mathbb{E}[(X+A)(X+A)^T] = \mathbb{E}[XX^T + AA^T].$$

Hence, we see that $Z$ has a spiked covariance structure. In this case, we can think of the principal $K$ eigenspace as providing the relevant signal and the rest being noise. Hence, if our regression targets only depend on the principal $K$ eigenvectors (i.e. $Y = \beta^T X$), then our model captures this scenario. Spiked covariance models have been studied in prior theoretical works on generalization (Liang et al., 2020; Ba et al., 2023).

## 3  Theoretical Results

This section presents our main result – Theorem 1. We present the results here and discuss insights in section 4. All proofs are in Appendix C.

**Theorem 1** (In-Subspace Test Error). *Let $r < |d - N|$. Let the SVD of $X_{trn}$ be $U\Sigma_{trn}V_{trn}^T$, let $L := U^T X_{tst}$, $\beta_U := U^T\beta$, and $c := d/N$. Under our setup and Assumptions 1 and 2, the test error (Equation 1) is given by the following. If $c < 1$ (under-parameterized regime)*

$$\mathcal{R}(W_{opt}, UL) = \frac{\eta_{trn}^4}{N_{tst}} \left\| \beta_U^T(\Sigma_{trn}^2 c + \eta_{trn}^2 I)^{-1} L \right\|_F^2$$
$$+ \frac{\eta_{tst}^2}{d} \frac{c^2}{1-c} \operatorname{Tr}\left( \beta_U \beta_U^T \Sigma_{trn}^2 \left( \Sigma_{trn}^2 + \frac{1}{\eta_{trn}^2} I \right) (\Sigma_{trn}^2 c + \eta_{trn}^2 I)^{-2} \right) + o\left( \frac{1}{N} \right)$$

*If $c > 1$ (over-parameterized regime)*

$$\mathcal{R}(W_{opt}, UL) = \frac{\eta_{trn}^4}{N_{tst}} \left\| \beta_U^T(\Sigma_{trn}^2 + \eta_{trn}^2 I)^{-1} L \right\|_F^2$$
$$+ \frac{\eta_{tst}^2}{d} \frac{c}{c-1} \operatorname{Tr}(\beta_U \beta_U^T (I + \eta_{trn}^2 \Sigma_{trn}^{-2})^{-1}) + O\left( \frac{\|\Sigma_{trn}\|^2}{N^2} \right) + o\left( \frac{1}{N} \right)$$

*Proof Sketch:* The proof can be broken into multiple steps. First, the generalization error can be decomposed as follows (Sonthalia & Nadakuditi, 2023).

$$\mathcal{R}(W, X_{tst}) = \underbrace{\frac{1}{N_{tst}} \|Y_{tst} - W X_{tst}\|_F^2}_{\text{Bias}} + \underbrace{\frac{\eta_{tst}^2}{d} \|W\|_F^2}_{\text{Variance}}.$$

Second, we solve for $W_{opt}$. From classical theory we know that $W_{opt} = Y_{trn}(X_{trn} + A_{trn})^\dagger$. Using Wei (2001) and some careful linear algebraic observations, we expand these to get

$$W_{opt} = \begin{cases} U\Sigma_{trn}(P^TP)^{-1}Z^TK_1^{-1}H - U\Sigma_{trn}Z^{-1}HH^TK_1^{-1}ZP^+ & c > 1 \\ -U\Sigma_{trn}H_1^{-1}K^TA_{trn}^+ + U\Sigma_{trn}H_1^{-1}Z^T(QQ^T)^{-1}H. & c < 1 \end{cases}$$

The exact expression for the variables in the expansion can be found in Appendix C.

Third, we substitute this back into our expressions for the bias and variance and expand the norms in terms of trace. Then, we group quadratic terms together. Specifically, we shall see that for $c > 1$, the error is decomposed in the sums and products of the following terms - $HH^T$, $P^TP$, and $Z$.

Finally, we estimate these terms using random matrix theory. We compute both their expectations and variances and show that the terms concentrate to values that only depend on the spectrum of $A_{trn}$. Then, using assumption 2.4, we approximate the spectrum of $A_{trn}$ using the Marchenko Pastur distribution. Then, since the terms concentrate, we can approximate the expectation of the product with the product of the expectation. □

**Comments on Expression.** The asymmetry between $c > 1$ and $c < 1$ comes from the asymmetry in the expressions for $W_{opt}$ and the need for renormalizing things in the case when $c > 1$. We further note that, in general, the expression does not depend on the right singular vectors of $X_{trn}$ as $X_{trn}$ only appears as its gram matrix $X_{trn}X_{trn}^T$. Finally, $L$ is the coordinates of $X_{tst}$ in the basis given by $U$. Then suppose $X_{tst} = U_{tst}\Sigma_{tst}V_{tst}^T$. Then we see that $L = U^T U_{tst}\Sigma_{tst}V_{tst}^T$. Then, using the unitary invariance of the norm, we have that

$$\|\beta_U^T(\Sigma\_trn^2 c + \eta\_trn^2 I)^{-1}L\|_F^2 = \|\beta_U^T(\Sigma_{trn}^2 c + \eta\_trn^2 I)^{-1}U^T U_{tst}\Sigma_{tst}\|_F^2.$$

Hence, the error depends on an alignment term $U^T U_{tst}$ and the singular values $\Sigma_{tst}$. Let us also examine the scale of the terms in the test error expressions. In Theorem 1, the bias is the norm term, and the variance is the trace term. Consider the setting where $\Sigma_{trn}^2$ grows as fast as possible and is $\Theta(N)$. This is also what happens when $X_{trn}$ is low-rank Gaussian (Huang et al., 2022). Then, the bias term is $O(1/N^2)$, while the variance term is $O(1/d) = O(1/N)$. The estimation error is $o(1/N)$ in the underparametrized case, so the only significant term, in this case, is the variance term. In the overparametrized case, the only significant terms are the variance term and the $O(\|\Sigma_{trn}\|^2/N^2)$ deviation from the estimate. If $\|X_{trn}\|_F^2 = o(N)$ instead of $\Theta(N)$, then the variance term is the only significant term again. Specific instantiations of Theorem 1 can be found in Appendix A and Appendix B. It is important to note that while the result may seem involved at first, we do in fact use it to develop insights in Section 4.

Theorem 1 is significant for various reasons. First, it can be considered the noisy-input counterpart to Dobriban & Wager (2018); Hastie et al. (2022), which works with noisy outputs instead. Second, it is one of the few results that do not require high dimensional well-conditioned covariance (Cheng & Montanari, 2022). Third, it generalizes Sonthalia & Nadakuditi (2023) to higher rank settings. Further, Theorem 1 estimates the test error in terms of the test instance instead of the generalization error. The latter is the expectation of the test error over a distribution of test instances. This is significant since it allows us to work with arbitrary test data in $\mathcal{V}$, provide results for distribution shift, and provide insights for data augmentation.

**Out-of-Distribution and Out-of-Subspace Generalization.** Theorem 1 can be used to understand OOD and out-of-subspace test error. For the former, Corollary 1 below bounds the change in generalization error in terms of in-subspace distribution shift.

**Corollary 1** (Distribution Shift Bound). *Consider a linear denoiser $W_{opt}$ trained on training data $X_{trn} = U\Sigma_{trn}V_{trn}^T$. Let it be tested on test data $X_{tst,1} = UL_1$ and $X_{tst,2} = UL_2$ generated possibly dependently from distributions supported in the span of $U$ with mean $U\mu_i$ and covariance $\Sigma_{U,i} = U\Sigma_i U^T$ respectively. Then, the difference in* generalization errors $\mathcal{G}_i := \mathbb{E}_{X_{tst,i}}[\mathcal{R}(W_{opt}, X_{tst,i})]$ *is bounded for $c < 1$ by*

$$|\mathcal{G}_2 - \mathcal{G}_1| \leq \frac{\sigma_1(\beta)^2 \eta_{trn}^4 r}{(\sigma_r(X_{trn})^2 f(c) + \eta_{trn}^2)^2}\|\Sigma_2 - \Sigma_1 + \mu_2\mu_2^T - \mu_1\mu_1^T\|_F + o\left(\frac{1}{N}\right)$$

*where $f(c) = c$ for $c < 1$ and $f(c) = 1$ otherwise. We add $O(\|\Sigma_{trn}\|_F^2/N^2)$ when $c \geq 1$.*

Corollary 1 is interesting as it implies that out-of-distribution error is primarily governed by the different between the means and covariances of the two distributions. Since these are relatively constant (population means and covariance changes slightly based on the data), this implies that in practice the error curves versus $c$ are very similar. So far, we have considered in-subspace distributional shifts. However, the distributional shift can take the test data out of the low-dimensional subspace $\mathcal{V}$.

**Theorem 2** (Out-of-Subspace Shift Bound). *If we have the same training data and solution $W_{opt}$ assumptions as in Theorem 1. Then, for **any** $X_{tst}$ for which there exists an $L$ and an $\alpha > 0$ such that $\|X_{tst} - UL\|_F \leq \alpha$, and $A_{tst}$ that satisfies 1,2 from Assumption 2, we have that the generalization error $\mathcal{R}(W_{opt}, X_{tst})$ satisfies*

$$|\mathcal{R}(W_{opt}, X_{tst}) - \mathcal{R}(W_{opt}, UL)| \leq \alpha^2 \sigma_1(W_{opt} + I)^2.$$

Theorem 2 shows us the surprising result that even if our test data is not in the subspace, we can exhibit double descent. This is also verified experimentally in the next section.

**Transfer Learning.** So far, we have considered the case where we have a covariate shift. That is, we have a distribution shift for $X$. However, we still assumed that the targets were given by the same function of $X$. That is, $\beta$ did not change. We now consider the scenario where $\beta$ can change to $\beta_{tst}$. Concretely, we are in the scenario where we train our model with $X_{trn} + A_{trn}$ and $Y_{trn} = \beta^T X_{trn}$ and test the model with data $X_{tst} + A_{tst}$ and $Y_{tst} = \beta_{tst}^T X_{tst}$. In this case, we have the following theorem.

**Theorem 3** (Transfer Learning). *Let $r < |d - N|$. Let the SVD of $X_{trn}$ be $U\Sigma_{trn}V_{trn}^T$, let $L := U^T X_{tst}$, $\beta_U := U^T\beta$, $\beta_{tst,U} = U^T\beta_{tst}$, and $c := d/N$. Under our setup and Assumptions 1 and 2, the test error (Equation 1) is given by the following. If $c < 1$ (under-parameterized regime)*

$$
\mathbb{E}_{A_{trn},A_{tst}}\left[\frac{1}{N_{tst}}\|Y_{tst} - W_{opt}(X_{tst} + A_{tst})\|_F^2\right] = \frac{1}{N_{tst}}\|(\beta_{tst,U} - \beta_U)^T L\|_F^2
$$
$$
- \frac{2\eta_{tst}^2}{N_{tst}}\operatorname{Tr}(\beta_U^T(\Sigma_{trn}^2 c + \eta_{trn}^2 I)^{-1}LL^T(\beta_{tst,U} - \beta_U))
$$
$$
+ \frac{\eta_{tst}^2}{d}\frac{c^2}{1-c}\operatorname{Tr}\left(\beta_U\beta_U^T\Sigma_{trn}^2\left(\Sigma_{trn}^2 + \frac{1}{\eta_{trn}^2}I\right)(\Sigma_{trn}^2 c + \eta_{trn}^2 I)^{-2}\right)
$$
$$
+ \frac{\eta_{trn}^4}{N_{tst}}\left\|\beta_U^T(\Sigma_{trn}^2 c + \eta_{trn}^2 I)^{-1}L\right\|_F^2 + o\left(\frac{1}{N}\right)
$$

*If $c > 1$ (over-parameterized regime)*

$$
\mathbb{E}_{A_{trn},A_{tst}}\left[\frac{1}{N_{tst}}\|Y_{tst} - W_{opt}(X_{tst} + A_{tst})\|_F^2\right] = \frac{1}{N_{tst}}\|(\beta_{tst,U} - \beta_U)^T L\|_F^2
$$
$$
- \frac{2\eta_{tst}^2}{N_{tst}}\operatorname{Tr}(\beta_U^T(\Sigma_{trn}^2 + \eta_{trn}^2 I)^{-1}LL^T(\beta_{tst,U} - \beta_U))
$$
$$
+ \frac{\eta_{tst}^2}{d}\frac{c}{c-1}\operatorname{Tr}(\beta_U\beta_U^T(I + \eta_{trn}^2\Sigma_{trn}^{-2})^{-1}) + O\left(\frac{\|\Sigma_{trn}\|^2}{N^2}\right)
$$
$$
+ \frac{\eta_{trn}^4}{N_{tst}}\left\|\beta_U^T(\Sigma_{trn}^2 + \eta_{trn}^2 I)^{-1}L\right\|_F^2 + o\left(\frac{1}{N}\right)
$$

*Proof.* Note that

$$
\mathbb{E}_{A_{trn},A_{tst}}\left[\frac{1}{N_{tst}}\|Y_{tst} - W_{opt}(X_{tst} + A_{tst})\|_F^2\right] = \mathbb{E}_{A_{trn},A_{tst}}\left[\frac{1}{N_{tst}}Y_{tst} - \beta^T X_{tst} + \beta^T X_{tst} - W_{opt}(X_{tst} + A_{tst})\|_F^2\right].
$$

Then, expand the norm twice to get the above result. $\qquad\square$

Here it is easy to see that since $\beta, \beta_{tst}$ are constant and $\|X_{tst}\|^2 = \|L\|^2 = O(N_{tst})$, we see that the first two terms in both expressions are $O_c(1)$ as a function of $c$. **Hence, we see that as a function of $c$, this error also has double descent.**

**Overfitting Paradigms.** We now compute the test error from $W^*$ in Theorem 4, from which we compute the limit of the relative excess error in Corollary 2, when data does not grow too slowly.

**Theorem 4** (Test Error for $W^*$). *In the same setting as Theorem 1, we have that $W^* = \beta_U^T\left(I + \frac{\eta_{trn}^2}{c}\Sigma_{trn}^{-2}\right)^{-1}U^T$ and*

$$
\mathcal{R}(W^*, UL) = \frac{\eta_{trn}^4 N^2}{N_{tst}d^2}\left\|\beta_U^T\left(\Sigma_{trn}^2 + \frac{\eta_{trn}^2 N}{d}I\right)^{-1}L\right\|_F^2 + \frac{\eta_{tst}^2}{d}Tr\left(\beta_U\beta_U^T\left(I + \frac{\eta_{trn}^2 N}{d}\Sigma_{trn}^{-2}\right)^{-2}\right).
$$

**Corollary 2** (Relative Excess Error). *Let $\|\Sigma_{trn}\|_F^2 = \Omega(N^{1/2+\epsilon})$. As $d, N \to \infty$ with $d/N \to c$, the relative excess error tends to $\frac{c}{1-c}$ in the underparametrized regime. In the overparametrized regime, when $\|\Sigma_{trn}\|_F^2 = o(N)$, it tends to $\frac{1}{c-1}$ and to $\frac{1}{c-1} + k$ for some constant $k$ when $\|\Sigma_{trn}\|_F^2 = \Theta(N)$.*

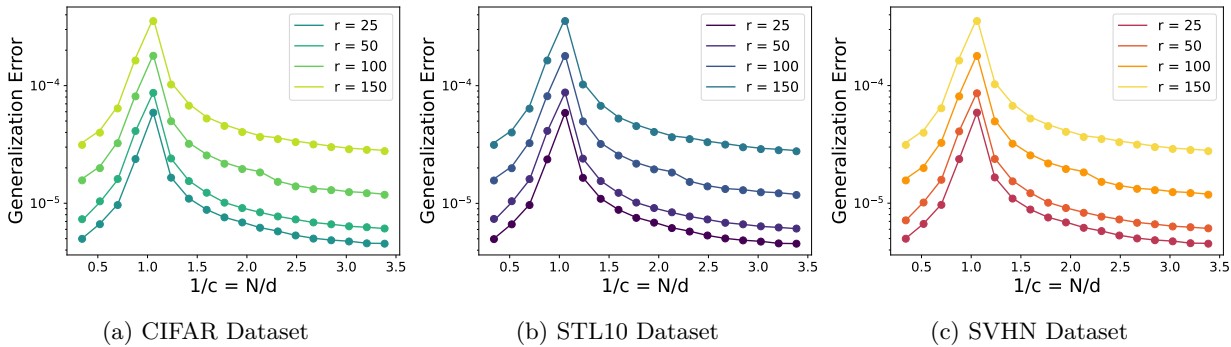

Figure 1: Figures showing the test error for Linear Regression vs $1/c = N/d$. Training data from the CIFAR dataset is projected onto its first $r$ principal components for $r = 25, 50, 100, 150$. 2500 test data points from CIFAR, STL10, and SVHN datasets are projected onto the same low-dimensional subspace. For empirical data points, shown by markers, we report the mean test error over at least 200 trials.

**Experimental Verification**  To experimentally verify our test error predictions using real-life data with distribution shift, we train a linear function $W_{opt}$ on CIFAR (Krizhevsky, 2009) and test on CIFAR, STL10 (Coates et al., 2011), and SVHN (Netzer et al., 2011). We **emphasize** that we run all experiments in the covariate shift case. That is, all models are trained on CIFAR, but then test posted different datasets. For computing test error, we simply compute $W_{opt}$ and plot the empirical average of $\frac{1}{N_{tst}}\|X_{tst} - W_{opt}(X_{tst} + A_{tst})\|_F^2$ over 200 trials. We run three main kinds of experiments. (a) First, we enforce the low-rank assumption to isolate the effect of distribution shift. To do this, we use principal component regression or PCR (Xu & Hsu, 2019; Huang et al., 2022). In PCR, instead of working with the true (and approximately low-rank) training data matrices $X_{tst}$, we find the best low-rank approximation $\hat{X}_{trn}$ of the training data by projecting it to an embedded subspace of the highest principal components. When testing, we project the test datasets to the same subspace to enforce the low-rank assumption before computing the empirical test error. (b) Second, to explicitly control the amount of deviation $\alpha$ from the low-rank assumption, we perturb the low-rank testing data from setting (a) and test using $\tilde{X}_{tst} := \hat{X}_{tst} + K_{tst}$, where $K_{tst}$ is Gaussian noise with covariance designed to control $\alpha$. (c) In the third case, we do not try to artificially enforce the low-rank assumption, relying on the approximate low-rank nature of real-life data. To do this, we report the test error for the matrices $X_{tst}$ themselves. Since $d$ is fixed, we vary $c$ by varying $N$. Figure 2 shows that the theoretical curves and the empirical results align perfectly for experimental setup (a) and that we have tight bounds for experimental setup (b). Numerically, we find that the relative error between the generalization error estimate and the average empirical error in experimental setup (a) is under 1% on average. For setup (c), since real-life data is only *approximately* low rank, we see a non-negligible error. However, the predictions still align well with the empirical results. **This shows that our low-rank assumption is meaningful, and aligns well with the behaviour of real-life data.**

**Single-variable Regression**  Theorem 1 is for any $\beta \in \mathbb{R}^{d \times k}$. Above, we presented figures for when $\beta = I_d$. Here, we present a similar figure for the single variable regression case. That is when $\beta \in \mathbb{R}^{d \times 1}$. Figure 1 shows the results. As before, we train all models on CIFAR and then test on CIFAR, SVHN, and STL10 with the same target $\beta$. As we can see the theory closely matches the empirical results.

**Classification**  We verify our theoretical predictions in the case of classification via two experiments. For the first experiment, we have two datasets. First, we have two Gaussian clusters with means $\pm\mu$ and have labels $\pm 1$. Second, we have three clusters with mutually orthogonal means and labels $e_1, e_2, e_3$. For this model, we train the model using mean squared error and then plot the test mean squared error and the test classification accuracy. Figure 3 shows that our theoretical prediction of the MSE exactly aligns with the empirical observations. Further, we see that the double descent in the MSE results in double descent in the classification error.

Since we are also interested in showing these phenomena also occur during transfer learning, we conduct the following experiment. We again have three mean vectors $\mu_1, \mu_2, \mu_3$ that are mutually orthogonal. We also

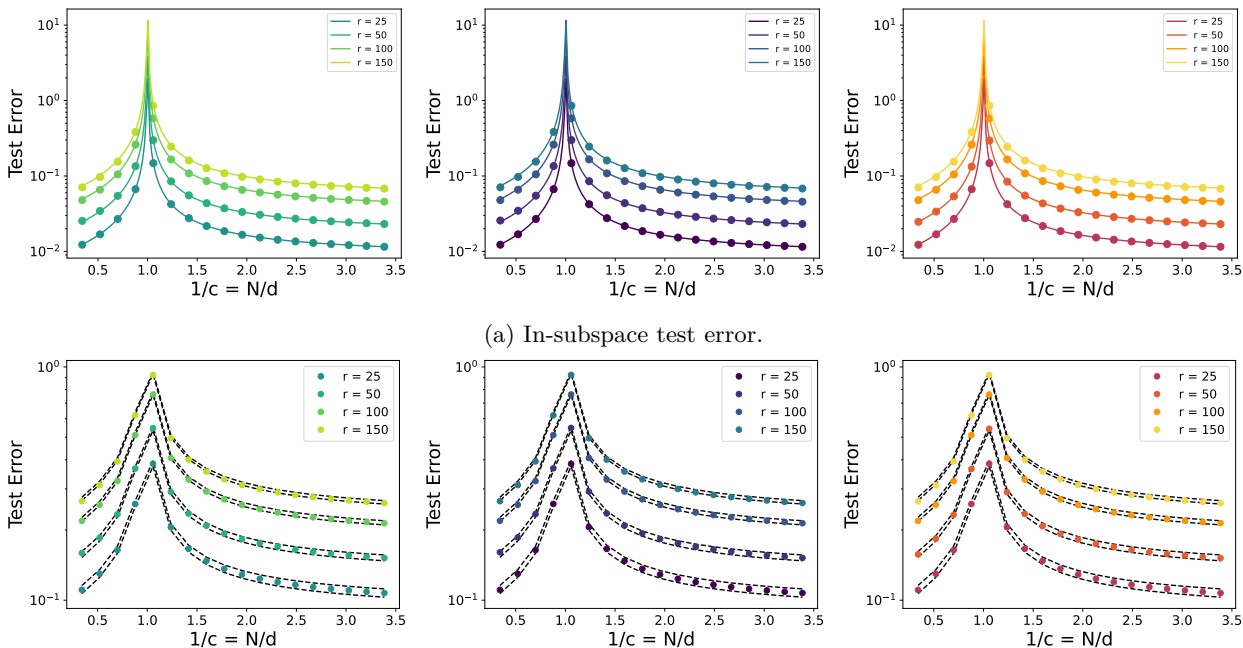

(a) In-subspace test error.

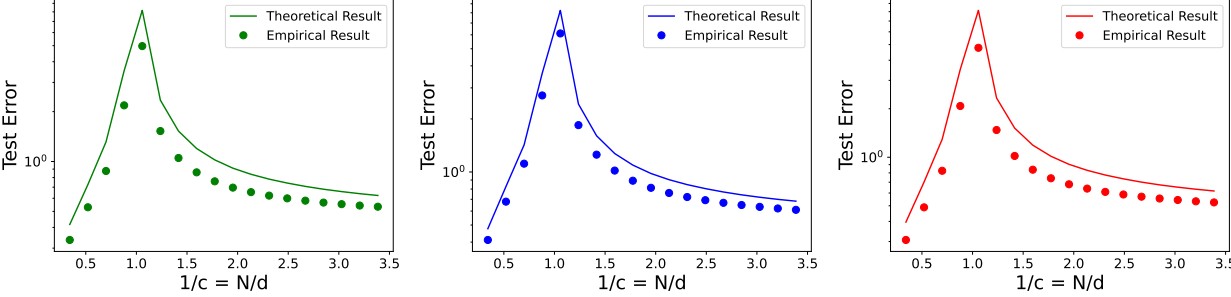

(b) For the out-of-subspace curves, we add full-dimensional Gaussian noise such that $\alpha = 0.1$. The upper and lower bounds for the empirical markers are given by Theorem 2).

(c) Test error estimated without projecting data, relying on the approximate low-rank structure of real-life data.

Figure 2: Figures showing the test error for $\beta = I$ vs $1/c = N/d$. In (a) and (b), training data from the CIFAR dataset is projected onto its first $r$ principal components for $r = 25, 50, 100, 150$. 2500 test data points from CIFAR (Green, Left col.), STL10 (Blue, Middle col.), and SVHN (Red, Right col.) datasets are projected onto the same low-dimensional subspace. (a) is in-subspace test error and (b) is out-of-subspace test error. In (c), we don't project the test data and report the standard test error, relying on the approximate low-rank structure in data instead of imposing it. For empirical data points, shown by markers, we report the mean test error over at least 200 trials. Similar results are obtained for single-variable regression with $\beta \in \mathbb{R}^d$.

have that the target for $\mu_i$ is the vector $e_i$. Then, during the test time, we assume that the means vectors for the three clusters have moved to $\mu_{tst,1}, \mu_{tst,2}, \mu_{tst,3}$ and we interested in mapping $\mu_{tst,i}$ to $e_i$. Hence, we have both covariate shifts, and the target function has changed. **Thus, we are predicting the error for transfer learning with covariates shift**. Figure 4 shows that Theorem 3 exactly predicts the error. *Again, we see double descent in the test mean squared error.* However, in this case, this double descent doesn't always translate to double descent in the classification accuracy (for the new classes). Two different shifts, with different trends, can be seen in Figure 4.

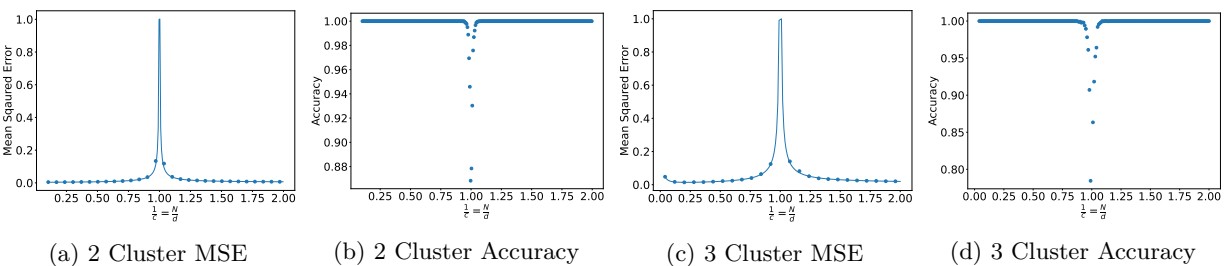

(a) 2 Cluster MSE     (b) 2 Cluster Accuracy     (c) 3 Cluster MSE     (d) 3 Cluster Accuracy

Figure 3: MSE and Accuracy when we only have covariate shift and $\beta_{tst} = \beta$.

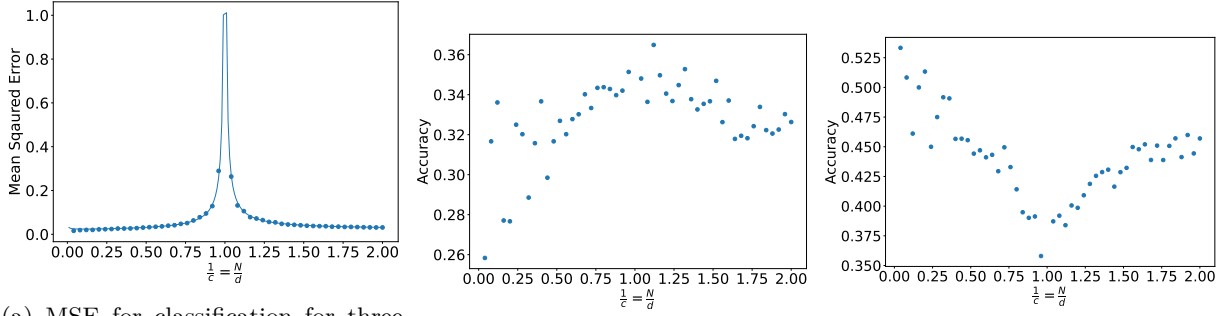

(a) MSE for classification for three Gaussian clusters with covariates shift and transfer learning    (b) Accuracy for first covariate shift    (c) Accuracy for second covariate shift

Figure 4: Experiment when we have *both* covariate shift and a shift in the target function.

## 4 Insights For Noisy-Input Problems

**Double Descent under Distribution Shift**   Notice that all our curves plotting test error against $1/c$ have a similar shape – they rise when $c$ approaches 1 from either side, and there is a peak at $c = 1$. This matches our theoretical results and establishes that denoising test error curves exhibit double descent, even for arbitrary test data in $\mathcal{V}$. To understand why this is happening, consider the denoising target, given by the MSE error below.

$$\mathbb{E}_{A_{tst}}[\|Y_{trn} - W(X_{trn} + A_{trn})\|_F^2] = \|Y_{trn} - WX_{trn}\|_F^2 + 2Tr(Y_{trn} - WX_{trn})^T A_{trn}) + \|WA_{trn}\|_F^2.$$

Notice that minimizing this sum forces us to reduce *both* $\|Y_{trn} - WX_{trn}\|_F$ and $\|W\|_F$, due to the third term (the "variance" term). We see that the noise is regularizing $\|W\|_F$ through the variance term $Tr(W^T W A_{trn} A_{trn}^T)$. This is the implicit regularization of $W$ due to noise. However, the strength of regularization due to the noise instance $A_{trn}$ is not the same across different values of $c$. When $c$ is close to 1, the distribution of the spectrum of $A_{trn}A_{trn}^T$ (the Marchenko-Pastur distribution) has support very close to zero. On the other hand, for $c$ far from 1, the non-zero eigenvalues of $A_{trn}A_{trn}^T$ are all bounded away from zero. This establishes that the effect of regularization weakens most near $c = 1$,[3] leading to a spike in the test error coming from the large norm of the learnt $W_{opt}$. This explanation is similar in spirit to the explanations for double descent in Xu & Hsu (2019) and others, but crucially adapts to implicit regularization due to noise.

**Amount of Training noise**   It was highlighted in Sonthalia & Nadakuditi (2023) that optimally picking the training noise level does not mitigate the double-descent phenomena observed in the generalization error for a linear model. In this section, we support this claim using our result from Theorem 1. Figure 5 shows the double descent curve of $\eta_{trn}$ and Figure 6 shows the generalization error when using the optimal amount

---

[3]The eigenvalues that are exactly zero do not contribute to weakening of the regularization. This is because we are choosing the minimum-norm optimizer $W^*$ for expected MSE error, and more zero eigenvalues increases flexibility, creating a larger set of optimizers to minimize the norm over. This helps decrease the components of $W^*$ by spreading them into more dimensions. This is identical in spirit to arguments about variance in overparametrized regimes in section 1.1 of Hastie et al. (2022).

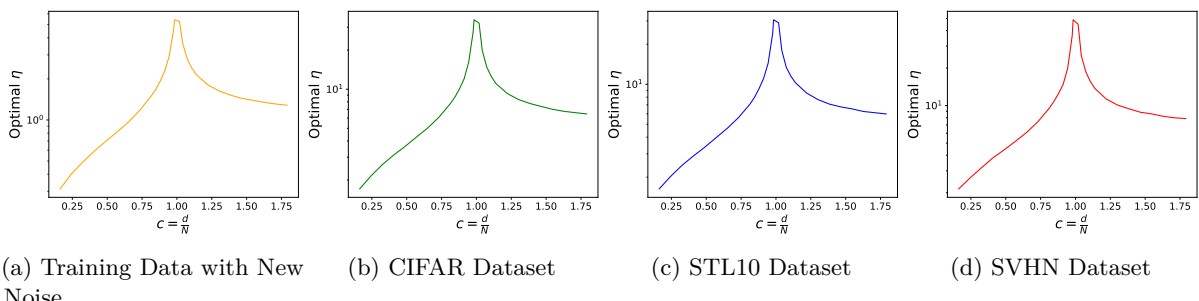

(a) Training Data with New Noise     (b) CIFAR Dataset     (c) STL10 Dataset     (d) SVHN Dataset

Figure 5: Optimal $\eta_{trn}$ that minimizes the test error given in Theorem 1 versus $c = d/N_{trn}$.

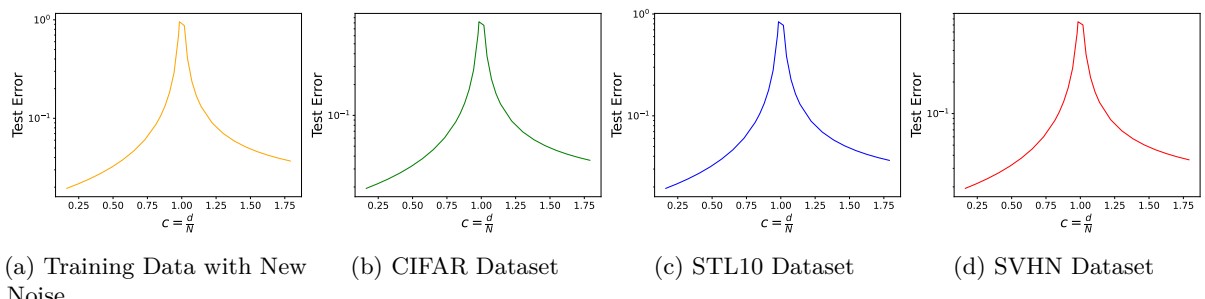

(a) Training Data with New Noise     (b) CIFAR Dataset     (c) STL10 Dataset     (d) SVHN Dataset

Figure 6: Test Error using Theorem 1 versus $1/c$ with optimal $\eta_{trn}$.

of training noise. As in other works such as Sonthalia & Nadakuditi (2023); Yilmaz & Heckel (2022), we see double descent in the regularization strength.

**Data Augmentation to Reduce Test Error.** In contrast with Nakkiran et al. (2020), but similar to Sonthalia & Nadakuditi (2023), optimally picking the noise parameter will not remove the peak in the test error. Instead, we use data augmentation and increase $N$ to try to move away from the peak, studying Theorem 1 to understand how this will affect test error. We take two approaches to data augmentation that individually exploit the absence of the IID assumptions. Since *the data does not have to be independent*, we can take the same training data and add fresh noise to increase $N$. Alternatively, since *the data does not have to be sampled from a specific distribution*, we can combine two different datasets into a larger training dataset to increase $N$. The subtlety to account for is that while increasing $N$ this way decreases $c$, it also increases $\Sigma_{trn}$. However, recall from the insights on the scale of terms above that the variance term is commonly the dominant one, and note that the variance term is roughly constant as $\Sigma_{trn}$ grows[4]. When $c < 1$, applying data augmentation increases $N$, thus decreasing $c$ further away from the peak at 1 and decreasing test error. When $c > 1$, applying data augmentation increases $N$, decreasing $c$ towards the peak at 1 and increasing test error. Of course, the latter phenomenon could be mitigated by adding other regularizers or by further augmenting the data. Figures 7d and 7h empirically verify the validity of Theorem 1 for the training data obtained from data augmentation. *We also see that increasing the number of in-distribution training data points reduces the out-of-distribution test error.*

**Overfitting Paradigms.** Recall from Corollary 2 that the relative excess error is given by $\frac{c}{1-c}$ for the underparameterized regime. This means that we approach benign overfitting as $c$ becomes arbitrarily small (which is essentially the classical regime). On the other hand, for the overparameterized case, we have that the relative excess error is given by $\frac{1}{1-c}$ when $\|\Sigma_{trn}\|^2 = o(N)$. Then as $c$ becomes arbitrarily large we again approach benign overfitting. This is interesting as this is the case when the norm of the signal grows *slower* than the norm of the noise. Hence we see benign overfilling when the signal-to-noise ratio goes to zero. If we had $\|\Sigma_{trn}\|^2 = \Theta(N)$, then our results suggest that the relative excess error *may* increase by a constant, leading to no benign overfitting. However, we believe that this is an artifact of the proof technique.

---

[4]The bias term decreases or stays constant with $\Sigma_{trn}$, but the bias term is irrelevant at common scales of $\Sigma_{trn}$.

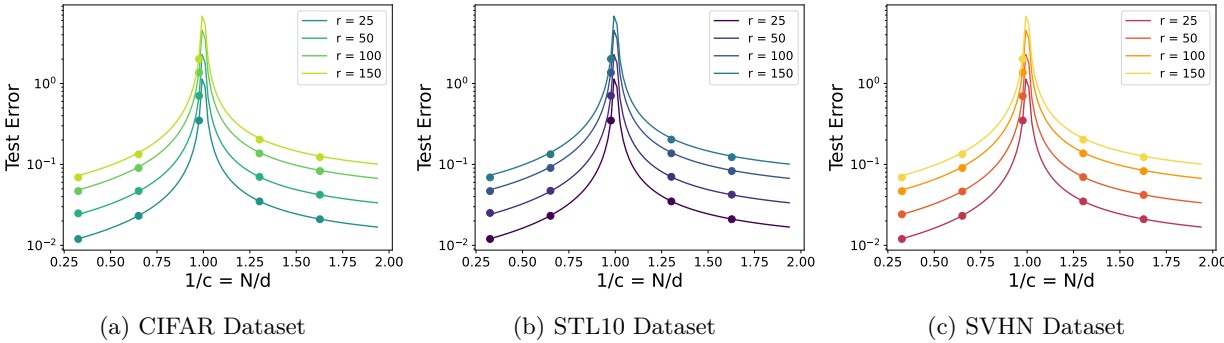

(a) CIFAR Dataset      (b) STL10 Dataset      (c) SVHN Dataset

(d) Data augmentation exploiting non-independence. For different $N_{trn}$ the training data is formed by repeating the same 1000 images from the CIFAR dataset.

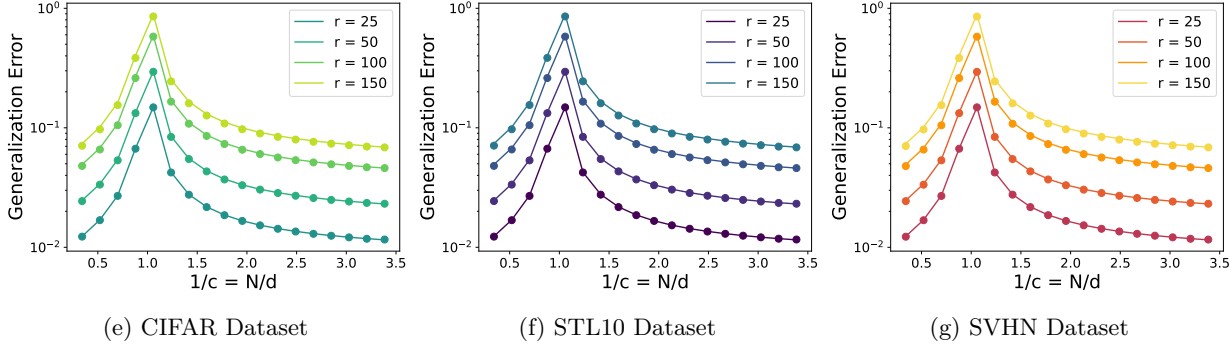

(e) CIFAR Dataset      (f) STL10 Dataset      (g) SVHN Dataset

(h) Data augmentation exploiting non-identicality of the distribution. The training data is formed by mixing CIFAR train split with STL10 train split dataset.

Figure 7: Data augmentation exploiting non-independence. For different $N_{trn}$ the training data is formed by repeating the same 1000 images from the CIFAR dataset.

Notice we don't observe benign overfitting except in the limit of arbitrarily large or arbitrarily small $c$. We make sense of this phenomenon using the following argument. Recall that while $W_{opt}$ is the minimum-norm matrix minimizing the MSE error for a *single noise instance*, $W^*$ is the minimum-norm matrix minimizing its *expectation* over noise. We can use this observation to view $W^*$ as $W_{opt}$ over an augmented dataset. Taking the expectation over noise in the training target is in spirit like augmenting the data with "infinitely many" copies of itself, each with fresh noise. So, $W^*$ can be viewed as the outcome of training $W_{opt}$ over an augmented dataset with a "changed $c$," where $c$ is replaced with a vanishingly small value while keeping $\Sigma^2_{trn}/N = \Sigma^2_{trn}c/d$ constant.

## 5 Conclusion and Future Work

We studied the problem of denoising low-dimensional input data perturbed with high-dimensional noise. Under very general assumptions, we provided estimated test error in terms of the specific instantiations of the training data and test data. This result is significant, as there is scarce prior work in the area of generalization for noisy inputs as well as generalization for low-rank data. Further, we tested our results with *real data* and achieved a relative MSE of 1%. Finally, the instance-specific estimate lets us provide many insights that would be harder to get with results on generalization error, such as showing double descent for arbitrary test data in our low-dimensional subspace, theoretically understanding data augmentation, and provably demonstrating as well as explaining the lack of benign overfitting.

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

# A   Additional Theoretical Results

## A.1   Test Error and Generalization Error

Recall from the introduction that the work of LeJeune et al. (2022) requires the simultaneous diagonalizability of the covariance matrices of training and test data. In a similar spirit, if we assume that the training and test data have the same left singular vectors, we recover the conjectured formula in (Sonthalia & Nadakuditi, 2023) as an immediate consequence of Theorem 1.

**Corollary 3** (Conjecture of Sonthalia & Nadakuditi (2023))**.** *Let the SVD of $X_{tst}$ be $U_{tst}\Sigma_{tst}V_{tst}^T$. In Theorem 1, if we further assume that $U^T U_{tst} = I$, then If $c < 1$ (under-parameterized regime)*

$$\mathcal{R}(W_{opt}, UL) = \eta_{trn}^4 \left\| \beta_U^T (\Sigma_{trn}^2 c + \eta_{trn}^2 I)^{-1} \Sigma_{tst} \right\|_F^2$$
$$+ \frac{\eta_{tst}^2}{d} \frac{c^2}{1-c} \operatorname{Tr}\left( \beta_U \beta_U^T \Sigma_{trn}^2 \left( \Sigma_{trn}^2 + \frac{1}{\eta_{trn}^2} I \right) (\Sigma_{trn}^2 c + \eta_{trn}^2 I)^{-2} \right) + o\left(\frac{1}{N}\right)$$

*If $c > 1$ (over-parameterized regime)*

$$\mathcal{R}(W_{opt}, UL) = \eta_{trn}^4 \left\| \beta_U^T (\Sigma_{trn}^2 + \eta_{trn}^2 I)^{-1} \Sigma_{tst} \right\|_F^2$$
$$+ \frac{\eta_{tst}^2}{d} \frac{c}{c-1} \operatorname{Tr}(\beta_U \beta_U^T (I + \eta_{trn}^2 \Sigma_{trn}^{-2})^{-1}) + O\left( \frac{\|\Sigma_{trn}\|^2}{N^2} \right) + o\left(\frac{1}{N}\right)$$

Additionally, we can use Theorem 1 to give an expression for generalization error when the test data points are drawn from a distribution, possibly dependently.

**Corollary 4** (Generalization Error)**.** *Let $r < |d - N|$. Let the SVD of $X_{trn}$ be $U\Sigma_{trn}V_{trn}^T$, let $L := U^T X_{tst}$, $\beta_U := U^T \beta$, and $c := d/N$. Under our setup and Assumptions 1 and 2, with the further assumption that the columns of $L$ are drawn IID from a distribution with mean $\mu$ and Covariance $\Sigma$, the test error (Equation 1) is given by the following.*
*If $c < 1$ (under-parameterized regime)*

$$\mathbb{E}_L[\mathcal{R}(W_{opt}, UL)] = \eta_{trn}^4 \left\| \beta_U^T (\Sigma_{trn}^2 c + \eta_{trn}^2 I)^{-1} (\Sigma + \mu\mu^T)^{1/2} \right\|_F^2$$
$$+ \frac{\eta_{tst}^2}{d} \frac{c^2}{1-c} \operatorname{Tr}\left( \beta_U \beta_U^T \Sigma_{trn}^2 \left( \Sigma_{trn}^2 + \frac{1}{\eta_{trn}^2} I \right) (\Sigma_{trn}^2 c + \eta_{trn}^2 I)^{-2} \right) + o\left(\frac{1}{N}\right)$$

*If $c > 1$ (over-parameterized regime)*

$$\mathbb{E}_L[\mathcal{R}(W_{opt}, UL)] = \eta_{trn}^4 \left\| \beta_U^T (\Sigma_{trn}^2 + \eta_{trn}^2 I)^{-1} (\Sigma + \mu\mu^T)^{1/2} \right\|_F^2$$
$$+ \frac{\eta_{tst}^2}{d} \frac{c}{c-1} \operatorname{Tr}(\beta_U \beta_U^T (I + \eta_{trn}^2 \Sigma_{trn}^{-2})^{-1}) + O\left( \frac{\|\Sigma_{trn}\|^2}{N^2} \right) + o\left(\frac{1}{N}\right)$$

## A.2   Out-of-Distribution Generalization

Consider the following theorem bounding the difference in generalization error in terms of the change in the test set. Our main distribution shift result is a corollary of its proof.

**Theorem 5** (Test Set Shift Bound)**.** *Under the assumptions of Theorem 1, consider a linear regressor $W_{opt}$ trained on training data $X_{trn} = U\Sigma_{trn}V_{trn}^T$ with $\Sigma_{trn}$ such that $\sigma_r(X_{trn}) > M$, and tested on test data $X_{tst,1} = UL_1$ and $X_{tst,2} = UL_2$ with noise $A_{tst,1}, A_{tst,2}$ with the same variance $\eta_{tst^2}/d$. Then, the generalization errors $\mathcal{R}_1$ and $\mathcal{R}_2$ differ for $c < 1$ by*

$$|\mathcal{R}_2 - \mathcal{R}_1| \le \frac{\sigma_1(\beta)^2}{N_{tst}} \frac{\eta_{trn}^4 r}{(\sigma_r(X_{trn})^2 f(c) + \eta_{trn}^2)^2} \|L_2 L_2^T - L_1 L_1^T\|_F + o\left(\frac{1}{N}\right)$$

*where $f(c) = c$ for $c < 1$ and $f(c) = 1$ for $c \ge 1$. We add $O(\|\Sigma_{trn}\|_F^2/N^2)$ to the bound when $c > 1$.*

# B   Additional Implications of our Results

## B.1   Out-of-Subspace Generalization

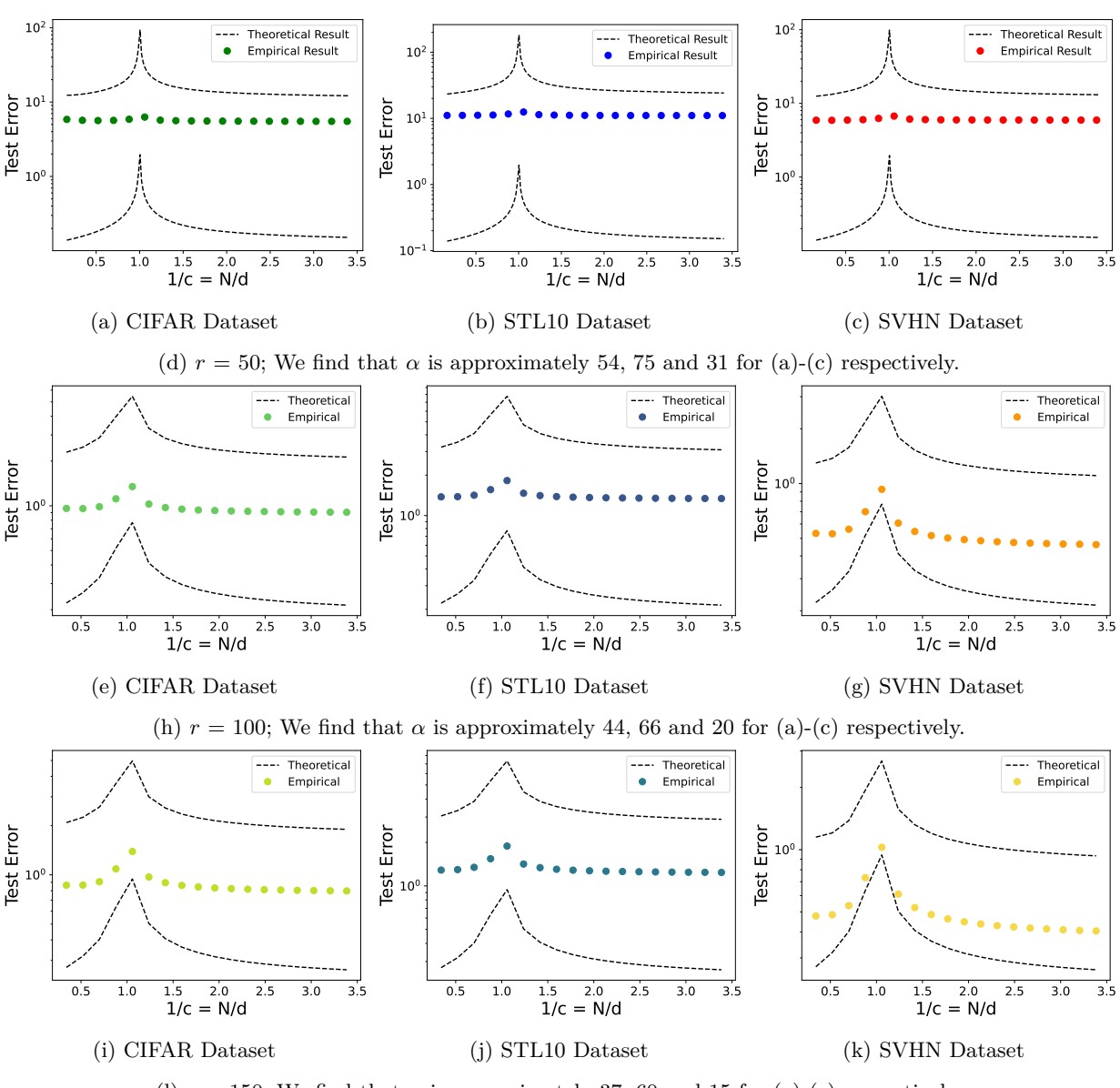

(d) $r = 50$; We find that $\alpha$ is approximately 54, 75 and 31 for (a)-(c) respectively.

(h) $r = 100$; We find that $\alpha$ is approximately 44, 66 and 20 for (a)-(c) respectively.

(l) $r = 150$; We find that $\alpha$ is approximately 37, 60 and 15 for (a)-(c) respectively.

Figure 8: Figure showing the test error vs $1/c$ when the test datasets retain their high dimensions. The training data is projected onto its first $r$ principal components. The markers denote the square root of test error obtained from empirical experiments. The dashed black lines, which act as the upper bounds for the empirical results, are given by $\sqrt{\mathcal{R}(UL)} + \alpha\sigma_1(W_{opt} + I)$ where $\mathcal{R}(UL)$ is the theoretical generalization error (refer Theorem 2). The dashed black lines, which act as the lower bounds, are given by $\sqrt{\mathcal{R}(UL)}$.

As mentioned in Section 3, we numerically verify Theorem 2 in two out-of-distribution setups namely small $\alpha$ and large $\alpha$. The application of our result to the small $\alpha$ case was already presented in the main paper; see Figure 2b. Here, we present the additional numerical results when the value of $\alpha$ is relatively large. We do not project the test datasets onto the low-dimensional subspace for this. The training dataset from the CIFAR train split is projected onto its first $r$ principal components where $r = 25, 50, 100$ and $150$. Figure 8

shows the theoretical bounds on the generalization error from Theorem 2. Unfortunately, for the large $\alpha$ case, the proposed lower bound in Theorem 2 is negative. However, we conjecture that $\mathcal{R}(UL)$ is a lower bound instead. The results for the large $\alpha$ case, shown in Figure 8, suggest the same. However, these bounds do not tell us anything about the shape of the generalization error curve.

The following corollary follows immediately from Theorem 2 and Theorem 5.

**Corollary 5.** *If $X_{tst,1}$ and $X_{tst,2}$ are two different test datasets and $X_{trn} = U\Sigma_{trn}V_{trn}^T$ is the training data such that there exists $L_i$ with $\alpha_i = \|X_{tst,i} - UL_i\|_F$, then for $\mathcal{R}_i := \mathcal{R}(W_{opt}, X_{tst,i})$*

$$|\mathcal{R}_2 - \mathcal{R}_1| \leq (\alpha_1^2 + \alpha_2^2)\sigma_1(W_{opt} + I)^2$$
$$+ \frac{\sigma_1(\beta)^2}{N_{tst}} \frac{\eta_{trn}^4 r}{(\sigma_r(X_{trn})^2 f(c) + \eta_{trn}^2)^2}\|L_2 L_2^T - L_1 L_1^T\|_F + o\left(\frac{1}{N}\right)$$

## B.2 Independent Identical Test data

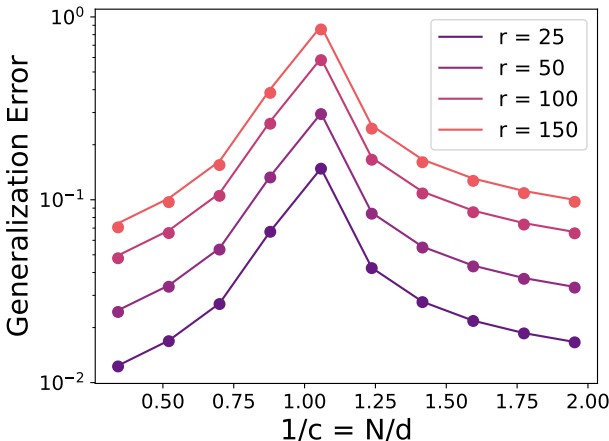

Figure 9: Figure showing the generalization error vs $1/c$ obtained for IID test data for $r = 25, 50, 100, 150$. The theoretical solid line curve is given by Corollary 6. We report the mean generalization error over at least 200 trials for empirical data points, shown by markers.

Let us assume that the test data is identically and independently drawn from some distribution $\mathcal{D}_{tst}$ with mean zero and covariance $\Sigma$. Then the generalization error is given by the following corollary.

**Corollary 6** (IID Test Data). *Let $r < |d - N|$. Let the SVD of $X_{trn}$ be $U\Sigma_{trn}V_{trn}^T$, let $L := U^T X_{tst}$, $\beta_U := U^T \beta$, and $c := d/N$. Under our setup and Assumptions 1 and 2, with the further assumption that the columns of $L$ are drawn IID from a distribution with mean zero and Covariance $\Sigma$, the test error (Equation 1) is given by the following.*
*If $c < 1$ (under-parameterized regime)*

$$\mathbb{E}_L[\mathcal{R}(W_{opt}, UL)] = \eta_{trn}^4 \left\|\beta_U^T(\Sigma_{trn}^2 c + \eta_{trn}^2 I)^{-1}\Sigma^{1/2}\right\|_F^2$$
$$+ \frac{\eta_{tst}^2}{d}\frac{c^2}{1-c}\operatorname{Tr}\left(\beta_U \beta_U^T \Sigma_{trn}^2\left(\Sigma_{trn}^2 + \frac{1}{\eta_{trn}^2}I\right)(\Sigma_{trn}^2 c + \eta_{trn}^2 I)^{-2}\right) + o\left(\frac{1}{N}\right)$$

*If $c > 1$ (over-parameterized regime)*

$$\mathbb{E}_L[\mathcal{R}(W_{opt}, UL)] = \eta_{trn}^4 \left\|\beta_U^T(\Sigma_{trn}^2 + \eta_{trn}^2 I)^{-1}\Sigma^{1/2}\right\|_F^2$$
$$+ \frac{\eta_{tst}^2}{d}\frac{c}{c-1}\operatorname{Tr}(\beta_U \beta_U^T(I + \eta_{trn}^2 \Sigma_{trn}^{-2})^{-1}) + O\left(\frac{\|\Sigma_{trn}\|^2}{N^2}\right) + o\left(\frac{1}{N}\right)$$

**Remark 3.** *Given any distribution on $\mathcal{V}$, we can consider the diffeomorphism that changes the basis to $U$. Hence, making assumptions on the distribution of $L$ versus the distribution of $X_{tst}$ does not cost us any generality.*

Figure 9, shows that the theoretical error aligns perfectly with the empirical result. The model is trained on the CIFAR dataset and tested on data drawn from an anisotropic Gaussian. The case of IID training data is presented in Appendix B.3.

### B.3 Independent Isotropic Identical Training Data

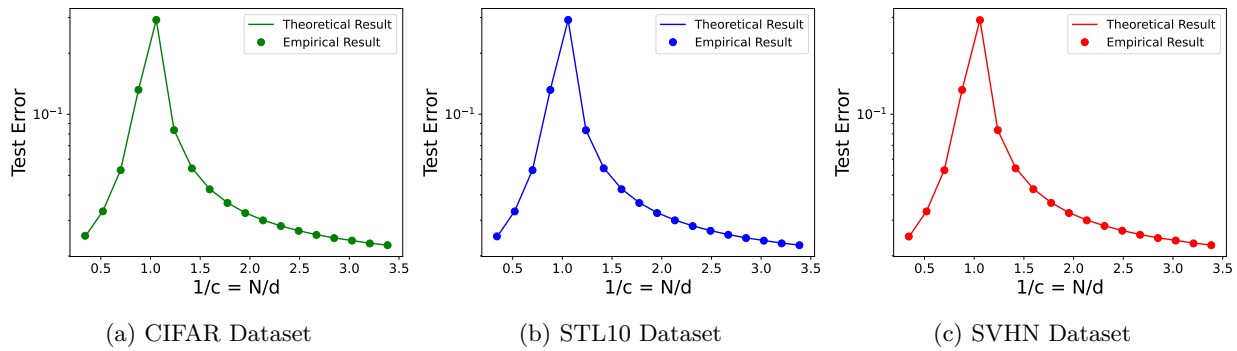

(a) CIFAR Dataset          (b) STL10 Dataset          (c) SVHN Dataset

Figure 10: Figure showing the test error vs $1/c$ for I.I.D. training data. The theoretical solid curves are obtained from the formula in Theorem 6. We report the mean test error over at least 200 trials for empirical data points, shown by markers.

Next, we consider the case of I.I.D training data. Let $U \in \mathbb{R}^{d \times r}$ be a matrix whose columns form an orthonormal basis for an $r$-dimensional space $\mathcal{V}$. Suppose the data is of the form $Uz$ for $z \in \mathbb{R}^r$ such that the coordinates of $z$ are sampled independently, have mean 0, variance $1/r$, and have bounded forth moments. Hence, in this case, we get the following theorem. Proof in Section C.8.

**Theorem 6** (I.I.D. Training Data With Isotropic Covariance)**.** *Let $c = d/N$ and $c_r = r/N$. Then if $c < 1$*

$$\mathbb{E}_{X_{trn}}[\mathcal{R}] = \frac{\eta_{trn}^4}{N_{tst}}\|(\Sigma_{trn}^2 c + \eta_{trn}^2 I)^{-1}L\|_F^2$$
$$+ \eta_{tst}^2 \frac{r}{d}\frac{1}{1-c}\left(T_1(c_r, \eta_{trn}^2/c) + \frac{1}{\eta_{trn}^2}T_2(c_r, \eta_{trn}^2/c)\right) + o\left(\frac{1}{N}\right)$$

*and if $c > 1$*

$$\mathbb{E}_{X_{trn}}[\mathcal{R}] = \frac{\eta_{trn}^4}{N_{tst}}\|(\Sigma_{trn}^2 + \eta_{trn}^2 I)^{-1}L\|_F^2 + \eta_{tst}^2 \frac{r}{d}\frac{c}{c-1}T_3(c_r, \eta_{trn}^2) + O\left(\frac{1}{N}\right)$$

*where $T_1(c_r, z) = T_3(cr, z) - zT_2(c_r, z)$, and*

$$T_2(c_r, z) = \frac{1 + c_r + zc_r}{2\sqrt{(1-c_r+c_r z)^2 + 4c_r^2 z}} - \frac{1}{2}, \;\; T_3(c_r, z) = \frac{1}{2} + \frac{1 + zc_r - \sqrt{(1-c_r+zc_r)^2 + 4c_r^2 z}}{2c_r}.$$

Figure 10 shows that the theoretical curves align perfectly with the empirical results where the training data is I.I.D. from a Gaussian with dimension 50. The test datasets from CIFAR, STL10, and SVHN datasets are also projected onto the low-dimensional subspace.

**I.I.D Test and Training Data**   We can combine the two cases where training and test data are I.I.D.. Specifically, for the case when $X_{tst}$ has $\kappa I$ as the covariance and $X_{trn}$ is as in the previous instantiation Section. Then the generalization error is given by the following corollary.

**Corollary 7** (I.I.D. Train and Tests Data With Isotropic Covariance). *Let $c = d/N$ and $c_r = r/N$. Then if $c < 1$*

$$\mathbb{E}_{X_{trn}}[\mathcal{R}] = \eta_{trn}^4 \cdot r \cdot \kappa \cdot T_4(c_r, \eta_{trn}^2/c)$$
$$+ \frac{r}{d}\frac{1}{1-c}\left(T_1(c_r, \eta_{trn}^2/c) + \frac{1}{\eta_{trn}^2}T_2(c_r, \eta_{trn}^2/c)\right) + o\left(\frac{1}{N}\right)$$

*and if $c > 1$*

$$\mathbb{E}_{X_{trn}}[\mathcal{R}] = \eta_{trn}^4 \cdot r \cdot \kappa \cdot T_4(c_r, \eta_{trn}^2) + \frac{r}{d}\frac{c}{c-1}T_3(c_r, \eta_{trn}^2) + O\left(\frac{1}{N}\right)$$

*where $T_1(c_r, z) = T_3(c_r, z) - zT_2(c_r, z)$, and*

$$T_2(c_r, z) = \frac{1 + c_r + zc_r}{2\sqrt{(1 - c_r + c_r z)^2 + 4c_r^2 z}} - \frac{1}{2}, \; T_3(c_r, z) = \frac{1}{2} + \frac{1 + zc_r - \sqrt{(1 - c_r + zc_r)^2 + 4c_r^2 z}}{2c_r},$$

$$T_4(c_r, z) = \frac{zc_r^2 + c_r^2 + zc_r - 2c_r + 1}{2z^2 c_r \sqrt{(1 - c_r + c_r z)^2 + 4c_r^2 z}} - \frac{1}{2z^2}\left(1 - \frac{1}{c_r}\right).$$

Figure 11 shows that the theoretical error aligns perfectly with the empirical result.

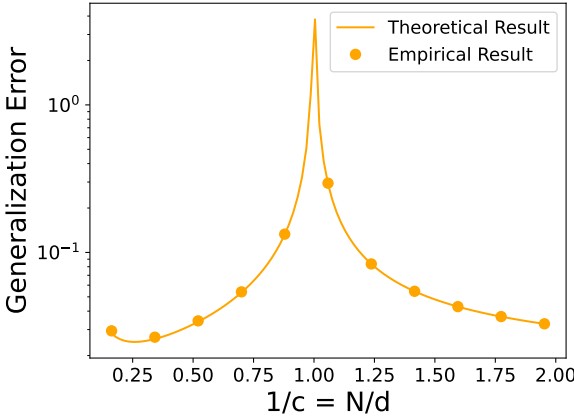

Figure 11: Figure showing the generalization error vs $1/c$ where training and test datasets are both I.I.D. The theoretical solid curve is obtained from Corollary 7. The empirical generalization error, shown by markers, is averaged over 50 trials.

## C Proofs

In all proofs, WLOG we assume $d/N = c$ since even though $d/N = c + o(1)$, the *relative* error we will accumulate from this assumption be $o(1)$. For instance, this means that the absolute error from this assumption in Theorem 1 will be $o(1/N)$, which can be absorbed into the $o(1/N)$ estimation error in the theorem.

### C.1 Proof for Theorem 1, Test Error

#### C.1.1 The Overparametrized Regime, $d > N$

We derive test error bounds for $\beta = I$ in our problem setting. We also denote $W_{opt}$ by $W$ in this subsection, for ease of notation. We begin by defining some notation. Let $X_{trn} = U\Sigma_{trn}V_{trn}^T$. Here $U$ is $d \times r$ with $U^T U = I$, $\Sigma_{trn}$ is $r \times r$, and $V_{trn}$ is $r \times N$. All of the following matrices are full rank.

1. $P = -(I - A_{trn}A_{trn}^{\dagger})U\Sigma_{trn} \in \mathbb{R}^{d \times r}$.
2. $H = V_{trn}^T A_{trn}^{\dagger} \in \mathbb{R}^{r \times d}$
3. $Z = I + V_{trn}^T A_{trn}^{\dagger} U\Sigma_{trn} \in \mathbb{R}^{r \times r}$
4. $K_1 = HH^T + Z(P^T P)^{-1}Z^T \in \mathbb{R}^{r \times r}$.

5. $U$ is $d \times r$ with $U^T U = I_{r \times r}$.
6. $\Sigma_{trn}$ is $r \times r$, with rank $r$.
7. $A_{trn} = \eta_{trn}\tilde{U}\tilde{\Sigma}\tilde{V}^T$ with $\tilde{U}$ is $d \times d$ unitary and $\tilde{\Sigma}$ is $d \times N$.

We will henceforth drop the subscript $A_{trn}$ in the expectation $\mathbb{E}_{A_{trn}}$.

**Lemma 1.** *If $d > N$ and $A_{trn}$ has full column rank, then*

$$W := X_{trn}(X_{trn} + A_{trn})^{\dagger} = U\Sigma_{trn}(P^T P)^{-1}Z^T K_1^{-1}H - U\Sigma_{trn}Z^{-1}HH^T K_1^{-1}ZP^{\dagger}. \tag{2}$$

*Proof.* Note that $P$ has full column rank and $A_{trn}$ has rank $N$. Thus, we can use corollary 2.2 from Wei (2001) to obtain

$$(A_{trn} + U\Sigma_{trn}V_{trn}^T)^{\dagger} = A_{trn}^{\dagger} + A_{trn}^{\dagger}U\Sigma_{trn}P^{\dagger} - (A_{trn}^{\dagger}H^T + A_{trn}^{\dagger}U\Sigma_{trn}(P^T P)^{-1}Z^T)K_1^{-1}(H + ZP^{\dagger}).$$

We are interested in simimplifying the expression. Multiplying this through, we obtain

$$W = U\Sigma_{trn}V_{trn}^T A_{trn}^{\dagger} + U\Sigma_{trn}V_{trn}^T A_{trn}^{\dagger}U\Sigma_{trn}P^{\dagger}$$
$$- U\Sigma_{trn}V_{trn}^T(A_{trn}^{\dagger}H^T + A_{trn}^{\dagger}U\Sigma_{trn}(P^T P)^{-1}Z^T)K_1^{-1}(H + ZP^{\dagger}).$$

Replacing $V_{trn}^T A_{trn} = H$,

$$W = U\Sigma_{trn}H + U\Sigma_{trn}HU\Sigma_{trn}P^{\dagger} - U\Sigma_{trn}V_{trn}^T(A_{trn}^{\dagger}H^T K_1^{-1}H + A_{trn}^{\dagger}H^T K_1^{-1}ZP^{\dagger}$$
$$+ A_{trn}^{\dagger}U\Sigma_{trn}(P^T P)^{-1}Z^T K_1^{-1}H + A_{trn}^{\dagger}U\Sigma_{trn}(P^T P)^{-1}Z^T K_1^{-1}ZP^{\dagger}).$$

Through further simplification, we obtain

$$W = U\Sigma_{trn}H + U\Sigma_{trn}HU\Sigma_{trn}P^{\dagger} - U\Sigma_{trn}HH^T K_1^{-1}H - U\Sigma_{trn}HH^T K_1^{-1}ZP^{\dagger}$$
$$- U\Sigma_{trn}HU\Sigma_{trn}(P^T P)^{-1}Z^T K_1^{-1}H - U\Sigma_{trn}HU\Sigma_{trn}(P^T P)^{-1}Z^T K_1^{-1}ZP^{\dagger}.$$

Setting $HU\Sigma_{trn} = Z - I$ yields

$$W = U\Sigma_{trn}H + U\Sigma_{trn}ZP^{\dagger} - U\Sigma_{trn}P^{\dagger} - U\Sigma_{trn}HH^T K_1^{-1}H - U\Sigma_{trn}HH^T K_1^{-1}ZP^{\dagger}$$
$$- U\Sigma_{trn}Z(P^T P)^{-1}Z^T K_1^{-1}H + U\Sigma_{trn}(P^T P)^{-1}Z^T K_1^{-1}H$$
$$- U\Sigma_{trn}Z(P^T P)^{-1}Z^T K_1^{-1}ZP^{\dagger} + U\Sigma_{trn}(P^T P)^{-1}Z^T K_1^{-1}ZP^{\dagger}.$$

Combining terms and replacing $HH^T + Z(P^T P)^{-1}Z^T = K_1$, we prove

$$W = -U\Sigma_{trn}P^{\dagger} + U\Sigma_{trn}(P^T P)^{-1}Z^T K_1^{-1}H + U\Sigma_{trn}(P^T P)^{-1}Z^T K_1^{-1}ZP^{\dagger},$$
$$= U\Sigma_{trn}(P^T P)^{-1}Z^T K_1^{-1}H - U\Sigma_{trn}Z^{-1}(K_1 - Z(P^T P)^{-1}Z^T)K_1^{-1}ZP^{\dagger},$$
$$= U\Sigma_{trn}(P^T P)^{-1}Z^T K_1^{-1}H - U\Sigma_{trn}Z^{-1}HH^T K_1^{-1}ZP^{\dagger}.$$

$\square$

**Lemma 2.** *For $d > N + r$, $X_{tst} - WX_{tst} = U\Sigma_{trn}(P^T P)^{-1}Z^T K_1^{-1}\Sigma_{trn}^{-1}L$.*

*Proof.* Here, $X_{tst} = UL$ and $W$ is given by Equation 2. Substituting this, we get

$$X_{tst} - WX_{tst} = UL - U\Sigma_{trn}(P^T P)^{-1}Z^T K_1^{-1}HUL + U\Sigma_{trn}Z^{-1}HH^T K_1^{-1}ZP^{\dagger}UL.$$

Note that $P^\dagger U = -\Sigma_{trn}^{-1}$ and $HU\Sigma_{tst} = V_{trn}^T A_{trn}^\dagger U\Sigma_{trn}\Sigma_{trn}^{-1}\Sigma_{tst} = (Z-I)\Sigma_{trn}^{-1}\Sigma_{tst}$ which yields

$$
\begin{aligned}
X_{tst} - WX_{tst} &= UL - U\Sigma_{trn}(P^T P)^{-1}Z^T K_1^{-1}(Z-I)\Sigma_{trn}^{-1}L - U\Sigma_{trn}Z^{-1}HH^T K_1^{-1}Z\Sigma_{trn}^{-1}L, \\
&= U\Sigma_{trn}Z^{-1}(Z - Z(P^T P)^{-1}Z^T K_1^{-1}(Z-I) - HH^T K_1^{-1}Z)\Sigma_{trn}^{-1}L, \\
&= U\Sigma_{trn}Z^{-1}(Z - (Z-I) + HH^T K_1^{-1}(Z-I) - HH^T K_1^{-1}Z)\Sigma_{trn}^{-1}L, \\
&= U\Sigma_{trn}Z^{-1}(K_1 - HH^T)K_1^{-1}\Sigma_{trn}^{-1}L, \\
&= U\Sigma_{trn}(P^T P)^{-1}Z^T K_1^{-1}\Sigma_{trn}^{-1}L.
\end{aligned}
$$

$\square$

**Lemma 3.** *Let $a^T b = 0$, and $Q$ is a Haar uniformly random orthogonal matrix. Let $\tilde{a} = Qa$, and $\tilde{b} = Qb$, then $\mathbb{E}[\tilde{a}_i \tilde{b}_i] = 0$.*

*Proof.* We note that $\mathbb{E}[\tilde{a}^T \tilde{b}] = \mathbb{E}[a^T b] = 0$. Then we get the result from symmetry. $\square$

**Lemma 4.** *For $c > 1$, we have that $\mathbb{E}[HH^T] = \dfrac{c}{\eta_{trn}^2(c-1)}I_r + o(1)$ and the variance of each entry is $O(1/(\eta_{trn}^4 N))$. For $c < 1$, we have that $\mathbb{E}[HH^T] = \dfrac{c^2}{\eta_{trn}^2(1-c)}I_r + o(1)$ and the variance is $O(1/(\eta_{trn}^4 d))$.*

*Proof.* Here we see that $HH^T = V_{trn}^T A_{trn}^\dagger (A_{trn}^\dagger)^T V_{trn} = V_{trn}^T (A_{trn}^T A_{trn})^\dagger V_{trn}$. Thus, if $V_{trn} = [v_1 \cdots v_r]$. Then we see that $HH^T$ is an $r \times r$ matrix such that

$$
(HH^T)_{ij} = v_i^T (A_{trn}^T A_{trn})^\dagger v_j = v_i^T \tilde{V}\frac{1}{\eta_{trn}^2}\tilde{\Sigma}^{-2}\tilde{V}^T v_j^T.
$$

Since $\tilde{V}$ is Haar uniformly random and independent of $\tilde{\Sigma}$, using Lemma 3 for $i \neq j$, we see that the expectation is 0. On the other hand if $i = j$, then using Lemma 6 from Sonthalia & Nadakuditi (2023), with $p = N$, $q = d$ and $A = \frac{1}{\eta_{trn}}A_{trn}$, we get that for $c > 1$ we have that $\mathbb{E}[v_i^T (A_{trn}^T A_{trn})^\dagger v_i] = \dfrac{c}{\eta_{trn}^2(c-1)} + o(1)$. while for $c < 1$, $\mathbb{E}[v_i^T (A_{trn}^T A_{trn})^\dagger v_i] = \dfrac{c^2}{\eta_{trn}^2(1-c)} + o(1)$. For the variance, let $A_{trn} = \eta_{trn}\tilde{U}\tilde{\Sigma}\tilde{V}^T$, then we have that

$$
v_i^T (A_{trn}^T A_{trn})^\dagger v_j = \frac{1}{\eta_{trn}^2}v_i^T \tilde{V}\tilde{\Sigma}^{-2}\tilde{V}^T v_j = \frac{1}{\eta_{trn}^2}a^T \tilde{\Sigma}^{-2}b = \sum_{i=1}^N \frac{1}{\eta_{trn}^2}\frac{1}{\tilde{\sigma}_i^2}a_i b_i.
$$

Where $a, b$ are orthogonal vectors (when $i \neq j$). Then for computing the variance when $c > 1$,

$$\mathbb{E}\left[\left(v_i^T(A_{trn}^T A_{trn})^{\dagger}v_j\right)^2\right] = \mathbb{E}\left[\left(\frac{1}{\eta_{trn}^2}\sum_{i=1}^N \frac{1}{\tilde{\sigma}_i^2}a_i b_i\right)^2\right]$$

$$= \frac{1}{\eta_{trn}^4}\mathbb{E}\left[\sum_{i=1}^N\sum_{j=1}^N \frac{1}{\tilde{\sigma}_i^2\tilde{\sigma}_j^2}a_i b_i a_j b_j\right]$$

$$= \left(\frac{c^2}{\eta_{trn}^4(c-1)^2} + o(1)\right)\mathbb{E}\left[\left(\sum_{i=1}^N a_i b_i\right)\left(\sum_{j=1}^N a_j b_j\right)\right]$$

$$+ \left(\frac{c^3}{\eta_{trn}^4(c-1)^3} - \frac{c^2}{\eta_{trn}^4(c-1)^2} + o(1)\right)\sum_{i=1}^N \mathbb{E}[a_i^2 b_i^2]$$

$$= 0 + \left(\frac{c^2}{\eta_{trn}^4(c-1)^3} + o(1)\right)\sum_{i=1}^N \frac{1}{N^2} + o\left(\frac{1}{N}\right)$$

$$= \frac{c^2}{\eta_{trn}^4(c-1)^3}\frac{1}{N} + o\left(\frac{1}{N}\right).$$

Here even though $a, b$ are not independent, because of the smaller variance in the entries, the error is absorbed in the $o\left(\frac{1}{N}\right)$ term. When $i = j$, we use the same proof Sonthalia & Nadakuditi (2023), to see that the variance is at most $O\left(\frac{1}{\eta_{trn}^4 N}\right)$. A very similar computation follows for the variance when $c < 1$. $\qquad\square$

We prove a general result on inverses of matrices that whose expected norms are $\Omega(1)$.

**Lemma 5.** *If* $\|\mathbb{E}[X_N]\| = \Omega(1)$ *as $N$ grows and* $\mathrm{Var}((X_N)_{ij}) = s_N$, *then* $\mathbb{E}[X_N^{-1}] = \mathbb{E}[X_N]^{-1} + O(s_N)$. *Additionally, if* $\mathrm{Var}((X_N - \mathbb{E}[X_N])_{ij}^2) = O(t_N)$, *then* $\mathrm{Var}((X_N^{-1})_{ij}) = O(s_N + t_N)$.

*Proof.* Let $\delta X_N = X_N - \mathbb{E}[X_n]$. Notice that $\delta X_N = O_P(s_N)$ and $\mathbb{E}[\delta X_N] = 0$. Additionally, by the Taylor expansion $(Y + \delta Y)^{-1} = Y^{-1} + Y^{-1}\delta Y Y^{-1} + O(\delta Y^2)$ we have that

$$X_N^{-1} = \mathbb{E}[X_N]^{-1} + \mathbb{E}[X_N]^{-1}\delta X_N \mathbb{E}[X_N]^{-1} + O(\delta X_N^2).$$

In particular, since $\mathbb{E}[X_n]^{-1} = O(1)$, we have

$$O(\mathbb{E}[X_N^{-1}] = \mathbb{E}[X_N]^{-1} + O(\mathrm{Var}((X_N)_{ij})) = \mathbb{E}[X_N]^{-1} + O(s_N).$$

Finally, note that $\mathrm{Var}((\delta X_N^2)_{ij}) = O(t_N)$ by assumption. So,

$$\mathrm{Var}((X_N^{-1})_{ij}) = \mathrm{Var}((\mathbb{E}[X_N]^{-1}\delta X_N \mathbb{E}[X_N]^{-1})_{ij}) + O(\mathrm{Var}((\delta X_N^2)_{ij})) = O(s_N + t_N)$$

since $\mathbb{E}[X_N]^{-1} = O(1)$. $\qquad\square$

**Lemma 6.** *For $c > 1$, we claim that* $\mathbb{E}[\Sigma_{trn}^{-1}P^T P\Sigma_{trn}^{-1}] = \left(1 - \frac{1}{c}\right)I_r$, *each entry has variance* $O\left(\frac{1}{d}\right)$, *and*

$$\mathbb{E}[\Sigma_{trn}(P^T P)^{-1}\Sigma_{trn}] = \frac{c}{c-1}I_r + O\left(\frac{1}{d}\right).$$

*with element-wise variance $O(1/d)$.*

*Proof.* Recall that $P = -(I - A_{trn}A_{trn}^\dagger)U\Sigma_{trn}$ Thus, we have that

$$P^T P = \Sigma_{trn}^T U^T (I - A_{trn}A_{trn}^\dagger)U\Sigma_{trn}.$$
$$= \Sigma_{trn}^T \Sigma_{trn} - \Sigma_{trn}^T U^T A_{trn}A_{trn}^\dagger U\Sigma_{trn}$$
$$= \Sigma_{trn}^T \Sigma_{trn} - \Sigma_{trn}^T U^T \tilde{U}\tilde{\Sigma}\tilde{\Sigma}^\dagger \tilde{U}^T U\Sigma_{trn}$$
$$= \Sigma_{trn}^T \Sigma_{trn} - \Sigma_{trn}^T R \begin{bmatrix} I_N & 0 \\ 0 & 0_{d-N} \end{bmatrix} R^T \Sigma_{trn}.$$

Where $R$ is a uniformly random $r \times d$ unitary matrix. Then by symmetry (of the sign of rows of $R$), we have that

$$\mathbb{E}[P^T P] = \Sigma_{trn}^2 - \Sigma_{trn}^T \left(\frac{1}{c}I_r\right)\Sigma_{trn} = \left(1 - \frac{1}{c}\right)\Sigma_{trn}^2.$$

So, we have that

$$\mathbb{E}\left[\Sigma_{trn}^{-1}P^T P\Sigma_{trn}^{-1}\right] = U^T \left(I - \mathbb{E}\left[A_{trn}A_{trn}^\dagger\right]\right)U = \left(1 - \frac{1}{c}\right)I_r.$$

Thus to compute the variance, we first compute the variance of $(A_{trn}A_{trn}^\dagger)_{ij}$. For this, we first note that

$$\begin{bmatrix} \frac{1}{c}I_N & 0 \\ 0 & 0 \end{bmatrix} = \mathbb{E}\left[\tilde{U}\tilde{\Sigma}\tilde{\Sigma}^\dagger\tilde{U}^T\right] = \mathbb{E}\left[A_{trn}A_{trn}^\dagger\right] = \mathbb{E}\left[A_{trn}A_{trn}^\dagger A_{trn}A_{trn}^\dagger\right].$$

The first equality follows from the symmetry of the signs of the rows of $\tilde{U}$. Then we can see that

$$\sum_k^d (A_{trn}A_{trn}^\dagger)_{ik}^2 = \begin{cases} \frac{1}{c} & i \leq N \\ 0 & i > N \end{cases}.$$

From Lemma 14 in Sonthalia & Nadakuditi (2023), we have that $\mathbb{E}[(A_{trn}A_{trn}^\dagger)_{ii}^2] = \frac{1}{c^2} + \frac{2}{cd} + o(1)$. Then combining this with the computation above and using symmetry, we have that for $i \neq j$ and $\min(i,j) \leq N$

$$\mathbb{E}[(A_{trn}A_{trn}^\dagger)_{ij}^2] = \frac{1}{N-1}\left(\frac{1}{c} - \frac{1}{c^2} + \frac{2}{cd} + o(1)\right).$$

Now consider the other (full) SVD of $X_{trn}$ given by $\hat{U}_{d\times d}\hat{\Sigma}_{d\times N}\hat{V}_{N\times N}^T$. Note that the top left $r \times r$ block of $\hat{\Sigma}$ is $\Sigma_{trn}$, and we can choose $\hat{U}$ so that the first $r$ columns of $\hat{U}$ give $U$. Note that since $\hat{U}^T\tilde{U}$ is still uniformly random, the symmetry argument above follows for $\hat{U}^T A_{trn}A_{trn}^\dagger\hat{U}$. Additionally, for $i,j \leq r$, $(\hat{U}^T A_{trn}A_{trn}^\dagger\hat{U})_{ij} = (U^T A_{trn}A_{trn}^\dagger U)_{ij}$ Thus, we see that for $i,j \leq r$

$$\mathbb{E}\left[(U^T A_{trn}A_{trn}^\dagger U)_{ij}^2\right] = \frac{1}{N-1}\left(\frac{1}{c} - \frac{1}{c^2} + \frac{2}{cd} + o(1)\right),$$

while for $i = j$, we get that it is $O\left(\frac{1}{N}\right)$ by Lemma 14 of Sonthalia & Nadakuditi (2023). Thus, finally, we have that arranged as a matrix

$$\mathbb{E}\left[(\Sigma_{trn}^{-1}P^T P\Sigma_{trn}^{-1}) \odot (\Sigma_{trn}^{-1}P^T P\Sigma_{trn}^{-1})\right] = O\left(\frac{1}{d}\right).$$

By an analogous symmetry argument, since $(A_{trn}A_{trn}^\dagger)^i = A_{trn}A_{trn}^\dagger$ for any $i$, we can show that

$$\text{Var}\left((U^T A_{trn}A_{trn}^\dagger U)_{ij}^2\right) = O\left(\frac{1}{d}\right).$$

We can in principle show a faster decay for this with a more involved argument, but this is enough for our purposes. We can now apply Lemma 5 with $X_N = I - (U^T A_{trn} A_{trn}^\dagger U)$ to see that

$$\mathbb{E}[\Sigma_{trn}(P^T P)^{-1}\Sigma_{trn}] = \frac{c}{c-1}I_r + O\left(\frac{1}{d}\right)$$

and has element-wise variance $O(1/d)$. $\qquad\square$

**Lemma 7.** *We have that $\mathbb{E}[Z] = I$ and $\mathrm{Var}(Z_{ij}) = O\left(\frac{\|\Sigma_{trn}\|^2}{\eta_{trn}^2 d}\right)$. Further, $E[Z\Sigma_{trn}^{-1}] = E[\Sigma_{trn}^{-1}Z] = \Sigma_{trn}^{-1}$ and each element has variance $O\left(\frac{1}{d}\right)$. Finally,*

$$\mathbb{E}[Z^{-1}] = I + O\left(\frac{\|\Sigma_{trn}\|^2}{d}\right) \text{ with } \mathrm{Var}((Z^{-1})_{ij}) = O\left(\frac{\|\Sigma_{trn}\|^2}{d} + \frac{\|\Sigma_{trn}\|^4}{d^2}\right).$$

*Proof.* The element-wise variance and expectation of $Z$ can be computed exactly as in the proof of Lemma 11 in Sonthalia & Nadakuditi (2023). Specifically, by considering the row $u_j$ of $U$ and the row $v_i$ of $V$, treating $Z_{ij}$ as $\beta$, and replacing $\theta_{trn}$ by $\sigma_j$. The expressions for the element-wise expectation and variance of $Z\Sigma_{trn}^{-1}$ and $\Sigma_{trn}^{-1}Z$ immediately follow from those of $Z$ and the fact that $\sigma_i/\sigma_j = \Theta(1)$ by Assumption 1.

For $Z^{-1}$, we continue the computation using $Z_{ij} = 1 + T_{ij}$ with $T_{ij} = \sigma_j \sum_{k=1}^{\min(d,N)} \frac{1}{\lambda_k} a_k b_k$ with $a$ and $b$ obtained using $v_j$ and $u_i$ respectively, and $\lambda_k$ a singular value of $A_{trn}$. It is easy to check that $\mathrm{Var}(T_{ij}^2) = O\left(\frac{\|\Sigma_{trn}\|^4}{N^2}\right)$ using a symmetry argument for $a_k$ and $b_k$ and the fact that $\mathbb{E}[1/\lambda_k^4] = O(1)$ by Lemma 5 of Sonthalia & Nadakuditi (2023). Now we can use Lemma 5 to conclude that

$$\mathbb{E}[Z^{-1}] = I + O\left(\frac{\|\Sigma_{trn}\|^2}{d}\right) \text{ with } \mathrm{Var}((Z^{-1})_{ij}) = O\left(\frac{\|\Sigma_{trn}\|^2}{d} + \frac{\|\Sigma_{trn}\|^4}{d^2}\right).$$

$\qquad\square$

**Lemma 8.** *For $c > 1$, $\mathbb{E}[K_1] = \frac{1}{\eta_{trn}^2}\frac{c}{c-1}I_r + \frac{c}{c-1}\Sigma_{trn}^{-2} + o(1)$ with element-wise variance $O(1/d)$. Further,*

$$\mathbb{E}[K_1^{-1}] = \eta_{trn}^2\left(1 - \frac{1}{c}\right)\left(\eta_{trn}^2\Sigma_{trn}^{-2} + I_r\right)^{-1} + o(1)$$

*with element-wise variance $O(1/d)$.*

*Proof.* From Lemma 6, we have that $\mathbb{E}[\Sigma_{trn}(P^T P)^{-1}\Sigma_{trn}] = \frac{c}{c-1}I_r + O\left(\frac{1}{d}\right)$. Recall that

$$K_1 = HH^T + Z(P^T P)^{-1}Z^T = HH^T + Z\Sigma_{trn}^{-1}(\Sigma_{trn}(P^T P)^{-1}\Sigma_{trn})\Sigma_{trn}^{-1}Z^T.$$

Then recall from Lemma 4 that

$$\mathbb{E}[HH^T] = \frac{1}{\eta_{trn}^2}\frac{c}{c-1}I_r + o(1).$$

For the second term in the expression for $K_1$, we want to use Lemmas 6 and 7, but they give expectations of each term separately. Note that

$$|\mathbb{E}[XY] - \mathbb{E}[X]\mathbb{E}[Y]| = |Cov(X,Y)| \leq \sqrt{\mathrm{Var}(X)\mathrm{Var}(Y)}$$

and also note the following fact, from Bohrnstedt & Goldberger (1969).

$$Cov(XY, WZ) = \mathbb{E}X\mathbb{E}W Cov(Y,Z) + \mathbb{E}Y\mathbb{E}Z Cov(X,W) + \mathbb{E}X\mathbb{E}Z Cov(Y,W) + \mathbb{E}Y\mathbb{E}W Cov(X,Z) + Cov(X,W)Cov(Y,Z) + Cov(Y,W)Cov(X,Z)$$

We use the facts above along with Lemmas 6 and 7 to compute the expectation. Specifically, the second term in $K_1$ is the product of three terms $Z\Sigma_{trn}^{-1}$, $(\Sigma_{trn}(P^T P)^{-1}\Sigma_{trn})$, and $\Sigma_{trn}^{-1}Z^T$. Hence we need the first fact to replace the expectation of the product of two terms with the product of the expectation of the two terms. To use this again, we would need to bound the variance of the product. Hence we need the second fact. Doing this computation, we get that

$$\mathbb{E}[K_1] = \frac{1}{\eta_{trn}^2}\frac{c}{c-1}I_r + \frac{c}{c-1}\Sigma_{trn}^{-2} + O\left(\frac{1}{d}\right) + o(1)$$

For the element-wise variance, consider $\delta K_1 = K_1 - \mathbb{E}[K_1]$. We cover the $i \neq j$ case. The $i = j$ case is analogous. From the proofs of Lemmas 4, 6, and 7, we have $Z_{ij} = I + T_{ij}$ and $(\Sigma_{trn}(P^T P)^{-1}\Sigma_{trn})_{ij} = U^T A_{trn}A_{trn}^\dagger U)_{ij}$. The expanding the product, we get that

$$(\delta K_1)_{ij} = \left(v_i(A_{trn}^T A_{trn})^\dagger v_j\right) + O\left((U^T A_{trn}A_{trn}^\dagger U)_{ij}\right) + O\left((U^T A_{trn}A_{trn}^\dagger U)_{ij}^2\right) + O(T_{ij})$$

$$+ O\left(\sum_{k=1}^N T_{ik}(U^T A_{trn}A_{trn}^\dagger U)_{kj}\right) + O\left(\sum_{k=1}^N T_{ik}(U^T A_{trn}A_{trn}^\dagger U)_{kj}^2\right) + O\left(\sum_{k=1}^N T_{ik}T_{kj}\right)$$

$$+ O\left(\sum_{k,l=1}^d T_{ik}(U^T A_{trn}A_{trn}^\dagger U)_{kl}T_{lj}\right) + O\left(\sum_{k,l=1}^d T_{ik}(U^T A_{trn}A_{trn}^\dagger U)_{kl}^2 T_{lj}\right)$$

Then since

$$\mathrm{Var}(XY) = Cov(X^2, Y^2) + (\mathrm{Var}(X) + (\mathbb{E}X)^2)(\mathrm{Var}(Y) + (\mathbb{E}Y)^2) - (Cov(X,Y) + \mathbb{E}X\mathbb{E}Y)^2$$

using this for terms five through nine, we get that $\mathrm{Var}\left((\delta K_1)_{ij}\right) = O\left(\frac{1}{d}\right)$.

For the inverse, we cover the $i \neq j$ case again. The $i = j$ case is analogous. We can perform an analogous computation to the one in the proof of Lemma 4 to get that

$$\mathrm{Var}\left((v_i(A_{trn}^T A_{trn})^\dagger v_j)^2\right) = O\left(\frac{1}{N}\right),$$

using the fact that $\mathbb{E}\left[\frac{1}{\lambda^4}\right] = O(1)$ for a random eigenvalue $\lambda_k$ of $A_{trn}$. We also use the fact that $(A_{trn}A_{trn}^\dagger)^p = A_{trn}A_{trn}^\dagger$ for any $p$ and a symmetry argument analogous to the one in the proof of Lemma 6 to note that

$$\mathbb{E}\left[(U^T A_{trn}A_{trn}^\dagger U)_{ij}^p\right] = O\left(\frac{1}{d}\right) \qquad p = 2, \ldots, 8.$$

One can also check by the arguments in the proof of Lemma 7 that

$$\mathbb{E}\left[T_{ij}^{2p}\right] = O\left(\frac{\sigma_i^p \sigma_j^p}{d^p}\right) = O(1).$$

These together with the facts about $\mathrm{Var}(XY)$ and $Cov(XY, ZW)$ above establish after a tedious but straightforward computation that

$$\mathrm{Var}((\delta K_1)_{ij}^2) = O\left(\frac{1}{d}\right).$$

We can now use Lemma 5 to establish that

$$\mathbb{E}[K_1^{-1}] = \eta_{trn}^2\left(1 - \frac{1}{c}\right)\left(\eta_{trn}^2\Sigma_{trn}^{-2} + I_r\right)^{-1} + O\left(\frac{1}{d}\right) + o(1)$$

$$= \eta_{trn}^2\left(1 - \frac{1}{c}\right)\left(\eta_{trn}^2\Sigma_{trn}^{-2} + I_r\right)^{-1} + o(1)$$

and $\mathrm{Var}((K_1^{-1})_{ij}) = O\left(\frac{1}{d}\right)$.

$\square$

**Lemma 9.** *When $c > 1$, we have for $W = W_{opt}$ that*

$$\mathbb{E}[\|W\|_F^2] = \frac{c}{c-1} \operatorname{Tr}(\Sigma_{trn}^2 (\Sigma_{trn}^2 + \eta_{trn}^2 I)^{-1}) + O\left(\frac{\|\Sigma_{trn}\|^2}{d}\right) + o(1).$$

*Proof.* We first use the estimates for the expectations from Lemmas 4, 6, 7, and 8 to get an estimate for the expectation of $\|W\|_F^2$. We get this estimate by treating various matrices in the product as independent. We then bound the deviation of the true expectation from this estimate using the variance estimates above. We begin the calculation as

$$\|W\|_F^2 = \operatorname{Tr}(W^T W)$$

Using Lemma 1, we see that the trace has three terms. The first term is

$$\operatorname{Tr}\left(H^T (K_1^{-1})^T Z((P^T P)^{-1})^T \Sigma_{trn}^T U^T U \Sigma_{trn} (P^T P)^{-1} Z^T K_1^{-1} H\right).$$

Here we have that $U$ is $d \times r$ with orthonormal columns. Hence we get that $U^T U = I$. Then since the trace is invariant under cyclic permutations, we get the following term

$$\operatorname{Tr}\left((\Sigma_{trn}(P^T P)^{-1}\Sigma_{trn})(\Sigma_{trn}^{-1} Z^T) K_1^{-1} H H^T (K_1^{-1})^T (Z\Sigma_{trn}^{-1})(\Sigma_{trn}(P^T P)^{-1}\Sigma_{trn})^T\right).$$

Now we use our random matrix theory estimates for various terms in the product. From Lemma 7, we have that $\mathbb{E}_{A_{trn}}[Z\Sigma_{trn}^{-1}] = \Sigma_{trn}^{-1}$. Thus, that first term's expectation can be estimated by

$$\operatorname{Tr}\left((\Sigma_{trn}(P^T P)^{-1}\Sigma_{trn})\Sigma_{trn}^{-1} K_1^{-1} H H^T (K_1^{-1})^T \Sigma_{trn}^{-1}(\Sigma_{trn}(P^T P)^{-1}\Sigma_{trn})^T\right).$$

Then using Lemma 4, we can further estimate this by

$$\frac{1}{\eta_{trn}^2} \frac{c}{c-1} \operatorname{Tr}\left((\Sigma_{trn}(P^T P)^{-1}\Sigma_{trn})\Sigma_{trn}^{-1} K_1^{-1} (K_1^{-1})^T \Sigma_{trn}^{-1}(\Sigma_{trn}(P^T P)^{-1}\Sigma_{trn})^T\right) + o(1).$$

Here, the error contribution of the $o(1)$ error from Lemma 4 is still $o(1)$ since we will see that all the other estimates are $O(1)$. Then we use Lemma 6, to replace $\Sigma_{trn}(P^T P)^{-1}\Sigma_{trn}$ to get

$$\frac{1}{\eta_{trn}^2} \frac{c}{c-1} \left(1 - \frac{1}{c}\right)^{-2} \operatorname{Tr}\left(\Sigma_{trn}^{-1} K_1^{-1} (K_1^{-1})^T (\Sigma_{trn}^T)^{-1}\right) + o(1).$$

Finally, we use Lemma 8 to replace the last term and get

$$\frac{1}{\eta_{trn}^2} \frac{c}{c-1} \left(\frac{c}{c-1}\right)^2 \operatorname{Tr}\left(\Sigma_{trn}^{-2} \eta_{trn}^4 \left(1 - \frac{1}{c}\right)^2 (I_r + \eta_{trn}^2 \Sigma_{trn}^{-2})^{-2}\right) + o(1).$$

This immediately simplifies to

$$\eta_{trn}^2 \frac{c}{c-1} \operatorname{Tr}\left(\Sigma_{trn}^2 (\Sigma_{trn}^2 + \eta_{trn}^2 I_r)^{-2}\right) + o(1). \tag{3}$$

The second term in $\operatorname{Tr}(W^T W)$ is

$$-2 \operatorname{Tr}\left(H^T (K_1^{-1})^T Z^T ((P^T P)^{-1})^T \Sigma_{trn}^T U^T U \Sigma_{trn} Z^{-1} H H^T Z P^\dagger\right).$$

We can rearrange this using cyclic invariance to

$$-2 \operatorname{Tr}\left((K_1^{-1})^T Z^T \Sigma_{trn}^{-1}(\Sigma_{trn}(P^T P)^{-1}\Sigma_{trn})^T \Sigma_{trn} Z^{-1} H H^T Z P^\dagger H^T\right).$$

Let us focus on the $P^\dagger H^T$ term. Since $P^T P$ is invertible, we have that $P$ has full column rank. Hence we have that

$$P^\dagger = (P^T P)^{-1} P^T.$$

Further, since $P = -(I - A_{trn}A_{trn}^\dagger)U\Sigma_{trn}$ and $H = V_{trn}^T A_{trn}^\dagger$, we have that

$$P^\dagger H^T = (P^T P)^{-1}\Sigma_{trn}^T U^T (I - A_{trn}A_{trn}^\dagger)(A_{trn}^\dagger)^T V_{trn}.$$

Finally, we notice that

$$A_{trn}A_{trn}^\dagger (A_{trn}^\dagger)^T = (A_{trn}^\dagger)^T.$$

Thus, we have that

$$P^\dagger H^T = (P^T P)^{-1}\Sigma_{trn}^T U^T (I - A_{trn}A_{trn}^\dagger)(A_{trn}^\dagger)^T V_{trn} = 0. \qquad (4)$$

Finally, the last term in $\mathrm{Tr}(W^T W)$ is

$$\mathrm{Tr}\left((P^\dagger)^T Z^T (K_1^{-1})^T H H^T (Z^{-1})^T \Sigma_{trn}^T U^T U \Sigma_{trn} Z^{-1} H H^T K_1^{-1} Z P^\dagger\right).$$

We note that

$$P^\dagger (P^\dagger)^T = (P^T P)^\dagger = (P^T P)^{-1}.$$

We use this observation along with cyclic invariance to get that the last term is the same as

$$\mathrm{Tr}\left((K_1^{-1})^T H H^T \Sigma_{trn}^2 Z^{-1} H H^T K_1^{-1} Z \Sigma_{trn}^{-1} (\Sigma_{trn}(P^T P)^{-1}\Sigma_{trn})\Sigma_{trn}^{-1} Z^T\right).$$

We again use Lemmas 4 and 7 to get that its expectation is estimated by

$$\frac{1}{\eta_{trn}^4}\left(\frac{c}{c-1}\right)^2 \mathrm{Tr}\left((K_1^{-1})^T \Sigma_{trn}^2 K_1^{-1} \Sigma_{trn}^{-1}(\Sigma_{trn}(P^T P)^{-1}\Sigma_{trn})\Sigma_{trn}^{-1}\right) + O\left(\frac{\|\Sigma_{trn}\|^2}{d}\right) + o(1).$$

The contribution of the $O\left(\frac{\|\Sigma_{trn}\|^2}{d}\right)$ error from Lemma 7 is still $O\left(\frac{\|\Sigma_{trn}\|^2}{d}\right)$ since the estimate for the expectation is $O(1)$. We now use Lemma 6, and 8 to see that the final term's expectation can be estimated by

$$\frac{1}{\eta_{trn}^4}\left(\frac{c}{c-1}\right)^3 \eta_{trn}^4 \left(\frac{c-1}{c}\right)^{-2}(I_r + \eta_{trn}\Sigma_{trn}^{-2})^{-2} + O\left(\frac{\|\Sigma_{trn}\|^2}{d}\right) + o(1)$$

$$= \frac{c}{c-1}\mathrm{Tr}(\Sigma_{trn}^4(\Sigma_{trn}^2 + \eta_{trn}^2 I_r)^{-2}) + O\left(\frac{\|\Sigma_{trn}\|^2}{d}\right) + o(1). \qquad (5)$$

Finally, to bound the deviation from this estimate, note that for real valued random variables $X, Y$ we have that $|\mathbb{E}[XY] - \mathbb{E}[X]\mathbb{E}[Y]| = |Cov(X,Y)| \le \sqrt{\mathrm{Var}(X)\mathrm{Var}(Y)}$ and for real valued random variables $X, Y, Z, W$, we have the following fact, from Bohrnstedt & Goldberger (1969).

$$Cov(XY, WZ) = \mathbb{E}X\mathbb{E}W Cov(Y,Z) + \mathbb{E}Y\mathbb{E}Z Cov(X,W) + \mathbb{E}X\mathbb{E}Z Cov(Y,W) +$$
$$\mathbb{E}Y\mathbb{E}W Cov(X,Z) + Cov(X,W)Cov(Y,Z) + Cov(Y,W)Cov(X,Z)$$

We repeatedly apply these two to upper bound the deviation between the product of the expectations in the estimates above and the expectation of the product. It is then straightforward to see that since all variances are $O(1/d)$ except for those of $Z^{-1}$ and $Z$, which are both $O(1)$ whenever $\Sigma_{trn} = O(\sqrt{d})$, the estimation error is $O(1/\sqrt{d}) = o(1)$.

So, we can conclude that each of the estimates in equations 3, 4 and 5 have error $o(1)$. Combining the terms together, we get from equations 3, 4 and 5 that

$$\|W\|_F^2 = \frac{c}{c-1}\mathrm{Tr}\left(\Sigma_{trn}^2(\Sigma_{trn}^2 + \eta_{trn}^2 I_r)(\Sigma_{trn} + \eta_{trn}^2 I_r)^{-2}\right) + O\left(\frac{\|\Sigma_{trn}\|^2}{d}\right) + o(1)$$

$$= \frac{c}{c-1}\mathrm{Tr}(\Sigma_{trn}^2(\Sigma_{trn}^2 + \eta_{trn}^2 I)^{-1}) + O\left(\frac{\|\Sigma_{trn}\|^2}{d}\right) + o(1).$$

$\square$

**Theorem 7.** *When $d > N + r$ and $\beta = I$, then the test error $\mathcal{R}(W, X_{tst})$ for $W = W_{opt}$ is given by*

$$\frac{\eta_{trn}^4}{N_{tst}} \| \left(\Sigma_{trn}^2 + \eta_{trn}^2 I\right)^{-1} L\|_F^2 + \frac{\eta_{tst}^2}{d} \frac{c}{c-1} \operatorname{Tr}(\Sigma_{trn}^2(\Sigma_{trn}^2 + \eta_{trn}^2 I)^{-1}) + O\left(\frac{\|\Sigma_{trn}\|^2}{d^2}\right) + o\left(\frac{1}{d}\right).$$

*Proof.* From Lemmas 1 and 2, we have that

$$\mathcal{R}(W, X_{tst}) = \mathbb{E}\left[\frac{1}{N_{tst}}\|U\Sigma_{trn}(P^T P)^{-1}Z^T K_1^{-1}\Sigma_{trn}^{-1}L\|_F^2 + \frac{\eta_{tst}^2}{d}\|W\|_F^2\right]$$

To compute the expectation of the first term, we observe that it is given by

$$\frac{1}{N_{tst}}Tr(U\Sigma_{trn}(P^T P)^{-1}Z^T K_1^{-1}\Sigma_{trn}^{-1}LL^T\Sigma_{trn}^{-1}K_1^{-1}Z(P^T P)^{-1}\Sigma_{trn}U^T).$$

We apply cyclic invariance to get that it is the same as

$$\frac{1}{N_{tst}}Tr(\Sigma_{trn}^{-1}K_1^{-1}Z\Sigma_{trn}^{-1}(\Sigma_{trn}(P^T P)^{-1}\Sigma_{trn})(\Sigma_{trn}(P^T P)^{-1}\Sigma_{trn})\Sigma_{trn}^{-1}Z^T K_1^{-1}\Sigma_{trn}^{-1}LL^T).$$

We finally use Lemmas 6, 7, and 8 to estimate it by

$$\frac{1}{N_{tst}}Tr\left(\Sigma_{trn}^{-2}\left(\frac{c}{c-1}\right)^2\left(\frac{c-1}{c}\right)^2\left(\Sigma_{trn}^{-2} + \frac{1}{\eta_{trn}^2}I\right)^{-2}\Sigma_{trn}^{-2}LL^T\right) + o\left(\frac{1}{d}\right)$$

$$= \frac{\eta_{trn}^4}{N_{tst}}Tr\left((\Sigma_{trn}^2 + \eta_{trn}^2 I)^{-2}LL^T\right) + o\left(\frac{1}{d}\right)$$

$$= \frac{\eta_{trn}^4}{N_{tst}}\| \left(\Sigma_{trn}^2 + \eta_{trn}^2 I\right)^{-1}L\|_F^2 + o\left(\frac{1}{d}\right)$$

Since test and train data are decoupled, we can treat $LL^T/N_{tst}$ as a constant as $N$ grows, noting that due the $\Sigma_{trn}^{-2}$, the final estimate is $o(1)$. So, repeating the deviation argument at the end of the proof of Lemma 9 above, we then have that the deviation from this estimate is $o\left(\frac{1}{d}\right)$.

Combining this with Lemma 9, we get that

$$\frac{\eta_{trn}^4}{N_{tst}}\| \left(\Sigma_{trn}^2 + \eta_{trn}^2 I\right)^{-1}L\|_F^2 + \frac{\eta_{tst}^2}{d}\frac{c}{c-1}\operatorname{Tr}(\Sigma_{trn}^2(\Sigma_{trn}^2 + \eta_{trn}^2 I)^{-1}) + O\left(\frac{\|\Sigma_{trn}^2\|}{d^2}\right) + o\left(\frac{1}{d}\right).$$

$\square$

### C.1.2 The Underparametrized Regime, $d < N$

We derive test error bounds for $\beta = I$ in our problem setting. We also denote $W_{opt}$ by $W$ in this subsection, for ease of notation.

We begin by defining some notation. Let $X_{trn} = U\Sigma_{trn}V_{trn}^T$. Here $U$ is $d \times r$ with $U^T U = I$, $\Sigma_{trn}$ is $r \times r$, and $V_{trn}$ is $r \times N$. All of the following matrices are full rank.

1. $Q = V^T(I - A_{trn}^\dagger A_{trn}) \in \mathbb{R}^{N \times r}$.
2. $H = V_{trn}^T A_{trn}^\dagger \in \mathbb{R}^{r \times d}$
3. $Z = I + V_{trn}^T A_{trn}^\dagger U\Sigma_{trn} \in \mathbb{R}^{r \times r}$
4. $K = -A_{trn}^\dagger U\Sigma_{trn} \in \mathbb{R}^{N \times r}$.
5. $H_1 = K^T K + Z^T(QQ^T)^{-1}Z \in \mathbb{R}^{r \times r}$.
6. $U$ is $d \times r$ with $U^T U = I_{r \times r}$.
7. $\Sigma_{trn}$ is $r \times r$, with rank $r$.
8. $A_{trn} = \eta_{trn}\tilde{U}\tilde{\Sigma}\tilde{V}^T$ with $\tilde{U}$ is $d \times d$ unitary and $\tilde{\Sigma}$ is $d \times N$.

We will henceforth drop the subscript $A_{trn}$ in the expectation $\mathbb{E}_{A_{trn}}$.

**Lemma 10.** *When $d < N - r$, we have that*

$$W = -U\Sigma_{trn}H_1^{-1}K^T A_{trn}^\dagger + U\Sigma_{trn}H_1^{-1}Z^T(QQ^T)^{-1}H.$$

*Proof.* We know that $W = X(X + A_{trn})^\dagger$. By Corollary 2.3 of Wei (2001), setting $X = -CB$ with $C = -U\Sigma_{trn}$ and $B = V^T$, we have that

$$(X + A_{trn})^\dagger = A_{trn}^\dagger - Q^\dagger H - (K + Q^\dagger Z)H_1^{-1}(K^T A_{trn}^\dagger - Z^T(QQ^T)^{-1}H).$$

So, using the facts that $X = U\Sigma_{trn}V^T$, $K = -A_{trn}^\dagger U\Sigma_{trn}$, we have that

$$
\begin{aligned}
W &= X(X + A_{trn}^\dagger) \\
&= U\Sigma_{trn}V^T A_{trn}^\dagger - U\Sigma_{trn}Q^\dagger H + U\Sigma_{trn}V^T A_{trn}^\dagger U\Sigma_{trn}H_1^{-1}K^T A_{trn}^\dagger \\
&\quad - U\Sigma_{trn}V^T Q^\dagger Z H_1^{-1}K^T A_{trn}^\dagger - U\Sigma_{trn}V^T A_{trn}^\dagger U\Sigma_{trn}H_1^{-1}Z^T(QQ^T)^{-1}H \\
&\quad + U\Sigma_{trn}V^T Q^\dagger Z H_1^{-1}Z^T(QQ^T)^{-1}H.
\end{aligned}
$$

Using the fact that $H = V^T A_{trn}^\dagger$, we get that

$$
\begin{aligned}
W &= U\Sigma_{trn}H - U\Sigma_{trn}Q^\dagger H + U\Sigma_{trn}HU\Sigma_{trn}H_1^{-1}K^T A_{trn}^\dagger - U\Sigma_{trn}V^T Q^\dagger Z H_1^{-1}K^T A_{trn}^\dagger \\
&\quad - U\Sigma_{trn}HU\Sigma_{trn}H_1^{-1}Z^T(QQ^T)^{-1}ZZ^{-1}H + U\Sigma_{trn}V^T Q^\dagger Z H_1^{-1}Z^T(QQ^T)^{-1}ZZ^{-1}H.
\end{aligned}
$$

Using the fact that $Z = I + V^T A_{trn}^\dagger U\Sigma_{trn} = I + HU\Sigma_{trn}$, we get that

$$
\begin{aligned}
W &= U\Sigma_{trn}H - U\Sigma_{trn}Q^\dagger H + U\Sigma_{trn}(Z - I)H_1^{-1}K^T A_{trn}^\dagger - U\Sigma_{trn}V^T Q^\dagger Z H_1^{-1}K^T A_{trn}^\dagger \\
&\quad - U\Sigma_{trn}(Z - I)H_1^{-1}Z^T(QQ^T)^{-1}ZZ^{-1}H + U\Sigma_{trn}V^T Q^\dagger Z H_1^{-1}Z^T(QQ^T)^{-1}ZZ^{-1}H.
\end{aligned}
$$

Using the fact that $H_1 = K^T K + Z^T(QQ^T)^{-1}Z$, we get that

$$
\begin{aligned}
W &= U\Sigma_{trn}H - U\Sigma_{trn}Q^\dagger H + U\Sigma_{trn}ZH_1^{-1}K^T A_{trn}^\dagger - U\Sigma_{trn}H_1^{-1}K^T A_{trn}^\dagger \\
&\quad - U\Sigma_{trn}V^T Q^\dagger Z H_1^{-1}K^T A_{trn}^\dagger - U\Sigma_{trn}ZH_1^{-1}(H_1 - K^T K)Z^{-1}H \\
&\quad + U\Sigma_{trn}H_1^{-1}Z^T(QQ^T)^{-1}H + U\Sigma_{trn}V^T Q^\dagger Z H_1^{-1}(H_1 - K^T K)Z^{-1}H \\
&= U\Sigma_{trn}H - U\Sigma_{trn}Q^\dagger H + U\Sigma_{trn}ZH_1^{-1}K^T A_{trn}^\dagger - U\Sigma_{trn}H_1^{-1}K^T A_{trn}^\dagger \\
&\quad - U\Sigma_{trn}V^T Q^\dagger Z H_1^{-1}K^T A_{trn}^\dagger - U\Sigma_{trn}H + U\Sigma_{trn}ZH_1^{-1}K^T K Z^{-1}H \\
&\quad + U\Sigma_{trn}H_1^{-1}Z^T(QQ^T)^{-1}H + U\Sigma_{trn}V^T Q^\dagger H - U\Sigma_{trn}V^T Q^\dagger Z H_1^{-1}K^T K Z^{-1}H.
\end{aligned}
$$

Cancelling terms, we get that

$$
\begin{aligned}
W &= U\Sigma_{trn}ZH_1^{-1}K^T A_{trn}^\dagger - U\Sigma_{trn}H_1^{-1}K^T A_{trn}^\dagger - U\Sigma_{trn}V^T Q^\dagger Z H_1^{-1}K^T A_{trn}^\dagger \\
&\quad + U\Sigma_{trn}ZH_1^{-1}K^T K Z^{-1}H + U\Sigma_{trn}H_1^{-1}Z^T(QQ^T)^{-1}H \\
&\quad - U\Sigma_{trn}V^T Q^\dagger Z H_1^{-1}K^T K Z^{-1}H.
\end{aligned}
$$

And we rearrange to get that

$$
\begin{aligned}
W &= -U\Sigma_{trn}H_1^{-1}K^T A_{trn}^\dagger + U\Sigma_{trn}H_1^{-1}Z^T(QQ^T)^{-1}H + U\Sigma_{trn}(I - V^T Q^\dagger)ZH_1^{-1}K^T A_{trn}^\dagger \\
&\quad + U\Sigma_{trn}(I - V^T Q^\dagger)ZH_1^{-1}K^T K Z^{-1}H \\
&= -U\Sigma_{trn}H_1^{-1}K^T A_{trn}^\dagger + U\Sigma_{trn}H_1^{-1}Z^T(QQ^T)^{-1}H,
\end{aligned}
$$

where the last equality is because $Q = V^T(I - A_{trn}^\dagger A_{trn})$ has full rank, so $Q^\dagger = Q^T(QQ^T)^{-1}$, so $V^T Q^\dagger = V^T(I - A_{trn}^\dagger A_{trn})V(V^T(I - A_{trn}^\dagger A_{trn})V)^{-1} = I$. $\qquad\square$

**Lemma 11.** *For $d < N - r$, with notation as in Lemma 10 have that $X_{tst} - WX_{tst} = U\Sigma_{trn}H_1^{-1}Z^T(QQ^T)^{-1}\Sigma_{trn}^{-1}L$.*

*Proof.* Note that $X_{tst} - WX_{tst} = UL - U\Sigma_{trn}H_1^{-1}K^TA_{trn}^\dagger UL - U\Sigma_{trn}H_1^{-1}Z^T(QQ^T)^{-1}HUL$. Remember that $K = -A_{trn}U\Sigma$, so $A_{trn}U\Sigma_{tst} = -K\Sigma_{trn}^{-1}\Sigma_{tst}$ and $HU\Sigma_{tst} = (HU\Sigma)\Sigma_{trn}^{-1}\Sigma_{tst} = (Z - I)\Sigma_{trn}^{-1}\Sigma_{tst}$ This gives us the following equality.

$$\begin{aligned}
X_{tst} - WX_{tst} &= UL - U\Sigma_{trn}H_1^{-1}K^TK\Sigma_{trn}^{-1}L - U\Sigma_{trn}H_1^{-1}Z^T(QQ^T)^{-1}Z\Sigma_{trn}^{-1}L \\
&\quad + U\Sigma_{trn}H_1^{-1}Z^T(QQ^T)^{-1}\Sigma_{trn}^{-1}L \\
&= U(I - \Sigma_{trn}H_1^{-1}(K^TK + Z^T(QQ^T)^{-1}Z)\Sigma_{trn}^{-1} + \Sigma_{trn}H_1^{-1}Z^T(QQ^T)^{-1}\Sigma_{trn}^{-1})L.
\end{aligned}$$

Using the fact that $H_1 = K^TK + Z^T(QQ^T)^{-1}Z$, we get that

$$X_{tst} - WX_{tst} = UL - U\Sigma_{trn}H_1^{-1}H_1\Sigma_{trn}^{-1}L + U\Sigma_{trn}H_1^{-1}Z^T(QQ^T)^{-1}\Sigma_{trn}^{-1}L = U\Sigma_{trn}H_1^{-1}Z^T(QQ^T)^{-1}\Sigma_{trn}^{-1}L.$$

$\square$

**Lemma 12.** *For $c < 1$, we have that $\mathbb{E}[\Sigma_{trn}^{-1}K^TK\Sigma_{trn}^{-1}] = \dfrac{1}{\eta_{trn}^2}\dfrac{c}{1-c} + o(1)$ and the variance of the $ij^{th}$ entry is $O\left(\frac{1}{N}\right)$.*

*Proof.* Note that $K^TK = \Sigma_{trn}U^T(A_{trn}A_{trn}^T)^\dagger U\Sigma_{trn}$. So, $(K^TK)_{ij} = \sigma_i u_i^T(A_{trn}A_{trn}^T)^\dagger u_j \sigma_j$. Using ideas from Sonthalia & Nadakuditi (2023), we see that if $i \neq j$, then the expectation is 0. On the other hand if $i = j$, then using Lemma 6 from Sonthalia & Nadakuditi (2023), with $p = N$, $q = d$, $A = \frac{1}{\eta_{trn}}A_{trn}^T$, we get that

$$\mathbb{E}[(\Sigma_{trn}^{-1}K^TK\Sigma_{trn}^{-1})_{ii}] = \frac{1}{\eta_{trn}^2}\frac{c}{1-c} + o(1).$$

The result on the expectation follows immediately from this.

For the variance, pick arbitrary $i \neq j$ and fix them. Consider $a = \tilde{U}^*u_i$ and $b = \tilde{U}^*u_j$. They are uniformly random orthogonal unit vectors, not necessarily independent. Now note that

$$\begin{aligned}
(\Sigma_{trn}^{-1}(K^TK)\Sigma_{trn}^{-1})_{ij} &= u_i^T(\tilde{U}\tilde{\Sigma}\tilde{\Sigma}^*\tilde{U}^*)^\dagger u_j \\
&= u_i^T\tilde{U}(\tilde{\Sigma}\tilde{\Sigma}^*)^\dagger\tilde{U}^*u_j \\
&= a^T(\tilde{\Sigma}\tilde{\Sigma}^*)^\dagger b \\
&= \sum_{k=1}^d \frac{1}{\tilde{\sigma}_k^2}a_k b_k.
\end{aligned}$$

So, we get that

$$\begin{aligned}
\mathbb{E}[((\Sigma_{trn}^{-1}(K^TK)\Sigma_{trn}^{-1})_{ij})^2] &= \mathbb{E}\left[\sum_{k=1}^d\sum_{l=1}^d\frac{1}{\tilde{\sigma}_k^2\tilde{\sigma}_l^2}a_kb_ka_lb_l\right] \\
&= \left(\frac{c^2}{(1-c)^2} + o(1)\right)\mathbb{E}\left[\left(\sum_{k=1}^d a_kb_k\right)^2\right] \\
&\quad + \left(\frac{c^2}{(1-c)^3} - \frac{c^2}{(1-c)^2} + o(1)\right)\mathbb{E}\left[\sum_{k=1}^d a_k^2b_k^2\right] \\
&= \left(\frac{c^3}{(1-c)^3} + o(1)\right)\mathbb{E}\left[\sum_{k=1}^d a_k^2b_k^2\right] \\
&= \frac{c^3}{(1-c)^3}\sum_{k=1}^d \mathbb{E}[a_k^2]\mathbb{E}[b_k^2] + o\left(\frac{1}{d}\right),
\end{aligned}$$

where the last line holds due to the following reasoning, even though $a$ and $b$ are not independent. We then use the fact that

$$\mathbb{E}[a_k^2 b_k^2] - \mathbb{E}[a_k^2]\mathbb{E}[b_k^2] \leq \sqrt{\mathrm{Var}(a_k^2)\mathrm{Var}(b_k^2)}$$

and Lemma 13 of Sonthalia & Nadakuditi (2023), to get that $\mathrm{Var}\left(\sum_{k=1}^d a_k^2\right) = O\left(\frac{1}{d}\right)$. So, by symmetry of coordinates, $\mathrm{Var}(a_k^2) = O\left(\frac{1}{d^2}\right)$. The same holds for $b_k$, giving us that

$$\left|\mathbb{E}[a_k^2 b_k^2] - \mathbb{E}[a_k^2]\mathbb{E}[b_k^2]\right| \leq O\left(\frac{1}{d^2}\right).$$

This gives us that

$$\mathrm{Var}\left((\Sigma_{trn}^{-1}(K^T K)\Sigma_{trn}^{-1})_{ij}^2\right) = \frac{c^3}{d(1-c)^3} + o\left(\frac{1}{d}\right) \qquad i \neq j.$$

For $i = j$, we use Sonthalia & Nadakuditi (2023) to see that the variance is $O\left(\frac{1}{d}\right) = O\left(\frac{1}{N}\right)$ since $d = cN$. $\quad\square$

**Lemma 13.** *For $c < 1$, we have that $\mathbb{E}[\Sigma_{trn}^{-1}K^T A_{trn}^\dagger (A_{trn}^\dagger)^T K \Sigma_{trn}^{-1}] = \frac{1}{\eta_{trn}^2}\frac{c^2}{(1-c)^3} + o(1)$ and the variance of the $ij^{th}$ entry is $O\left(\frac{1}{N}\right)$.*

*Proof.* Let $M := \Sigma_{trn}^{-1}K^T A_{trn}^\dagger (A_{trn}^\dagger)^T K \Sigma_{trn}^{-1}$ and note that

$$\Sigma_{trn}^{-1}K^T A_{trn}^\dagger (A_{trn}^\dagger)^T K \Sigma_{trn}^{-1} = \Sigma_{trn} U^T (A_{trn}A_{trn}^T)^\dagger (A_{trn}A_{trn}^T)^\dagger U \Sigma_{trn}.$$

So, $M_{ij} = \sigma_i u_i^T (A_{trn}A_{trn}^T)^\dagger (A_{trn}A_{trn}^T)^\dagger u_j \sigma_j$. Using ideas from Sonthalia & Nadakuditi (2023), we see that if $i \neq j$, then the expectation is 0. On the other hand if $i = j$, then using Lemma 6 from Sonthalia & Nadakuditi (2023), with $p = N$, $q = d$, we get that

$$\mathbb{E}[M_{ii}] = \frac{\sigma_i^2}{\eta_{trn}^2}\frac{c^2}{(1-c)^3} + o(1).$$

For the variance, pick arbitrary $i \neq j$ and fix them. Consider $a = \tilde{U}^* u_i$ and $b = \tilde{U}^* u_j$. They are uniformly random orthogonal unit vectors, not necessarily independent. Now note that

$$\begin{aligned}
M_{ij} &= u_i^T (A_{trn}A_{trn}^T)^\dagger (A_{trn}A_{trn}^T)^\dagger u_j \\
&= u_i^T (\tilde{U}\tilde{\Sigma}\tilde{\Sigma}^*\tilde{\Sigma}\tilde{\Sigma}^*\tilde{U}^*)^\dagger u_j \\
&= u_i^T \tilde{U}(\tilde{\Sigma}\tilde{\Sigma}^*\tilde{\Sigma}\tilde{\Sigma}^*)^\dagger \tilde{U}^* u_j \\
&= a^T (\tilde{\Sigma}\tilde{\Sigma}^*\tilde{\Sigma}\tilde{\Sigma}^*)^\dagger b \\
&= \sum_{k=1}^d \frac{1}{\tilde{\sigma}_k^4} a_k b_k.
\end{aligned}$$

So, we get that

$$
\begin{aligned}
\mathbb{E}[M_{ij}^2] &= \mathbb{E}\left[\left(\sum_{k=1}^{d}\frac{1}{\tilde{\sigma}_k^4}a_k b_k\right)^2\right] \\
&= \mathbb{E}\left[\sum_{k=1}^{d}\sum_{l=1}^{d}\frac{1}{\sigma_k^4\sigma_l^4}a_k b_k a_l b_l\right] \\
&= \left(\frac{c^4(c^2+22/6c+1)}{(1-c)^7}+o(1)\right)\mathbb{E}\left[\left(\sum_{k=1}^{d}a_k b_k\right)^2\right]+(\chi(c)+o(1))\mathbb{E}\left[\sum_{k=1}^{d}a_k^2 b_k^2\right] \\
&= (\chi(c)+o(1))\mathbb{E}\left[\sum_{k=1}^{d}a_k^2 b_k^2\right] \\
&= (\chi(c)+o(1))\mathbb{E}\left[\sum_{k=1}^{d}a_k^2 b_k^2\right] \\
&= \chi(c)\sum_{k=1}^{d}\mathbb{E}[a_k^2]\mathbb{E}[b_k^2]+o\left(\frac{1}{d}\right),
\end{aligned}
$$

where the last line holds due to the argument in the proof of Lemma 12. Here $\chi(c)$ is some function of $c$. This gives us that $\mathrm{Var}[M_{ij}] = \frac{1}{d}\chi(c)+o\left(\frac{1}{d}\right)$ for $i \neq j$. For $i = j$, we use Sonthalia & Nadakuditi (2023) to see that the variance is $O\left(\frac{1}{d}\right)$. □

**Lemma 14.** *For $c < 1$, we have that $\mathbb{E}[QQ^T] = (1-c)I_r$ and the variance of each entry is $O\left(\frac{1}{d}\right)$. Further,*

$$
\mathbb{E}[(QQ^T)^{-1}] = \frac{1}{1-c}I_r + O\left(\frac{1}{d}\right).
$$

*and each element has variance $O(1/d)$*

*Proof.* Recall that $Q = V^T(I - A_{trn}^\dagger A_{trn})$. We thus have that

$$
\begin{aligned}
QQ^T &= V^T(I - A_{trn}^\dagger A_{trn})V. \\
&= V^T V - V^T A_{trn}^\dagger A_{trn} V \\
&= I_r - V^T \tilde{V}\tilde{\Sigma}^\dagger\tilde{\Sigma}\tilde{V}^T V \\
&= I_r - R\begin{bmatrix} I_d & 0 \\ 0 & 0_{N-d}\end{bmatrix}R^T.
\end{aligned}
$$

Where $R$ is a uniformly random $r \times N$ unitary matrix. Then by symmetry (of the sign of rows of $R$), we have that

$$
\mathbb{E}[QQ^T] = I_r - cI_r = (1-c)I_r.
$$

Next notice that

$$
\mathbb{E}[QQ^T] = V^T(I - \mathbb{E}[A_{trn}^\dagger A_{trn}])V,
$$

thus to compute the variance, we first compute the variance of $(A_{trn}^\dagger A_{trn})_{ij}$. For this, we first note that

$$
\begin{bmatrix} cI_d & 0 \\ 0 & 0\end{bmatrix} = \mathbb{E}[A_{trn}^\dagger A_{trn}] = \mathbb{E}[A_{trn}^\dagger A_{trn}A_{trn}^\dagger A_{trn}].
$$

Since $A_{trn}^\dagger A_{trn}$ is symmetric, we can see that

$$
\sum_{k}^{d}((A_{trn}^\dagger A_{trn})_{ik})^2 = \begin{cases} c & i \leq d \\ 0 & i > d\end{cases}.
$$

From Lemma 15 in Sonthalia & Nadakuditi (2023), we have that $\mathbb{E}[((A_{trn}^\dagger A_{trn})_{ii})^2] = c^2 + \frac{2c}{N} + o(1)$. Then combining this with the computation above and using symmetry, we have that for $i \neq j$ and $\min(i,j) \leq d$

$$\mathbb{E}[(A_{trn}^\dagger A_{trn})_{ij}^2] = \frac{1}{d-1}\left(\frac{1}{c} - \frac{1}{c^2} + \frac{3}{cd} + o(1)\right).$$

Now consider the other (full) SVD of $X_{trn}$ given by $\hat{U}_{d\times d}\hat{\Sigma}_{d\times N}\hat{V}_{N\times N}^T$. Note that the top left $r \times r$ block of $\hat{\Sigma}$ is $\Sigma_{trn}$, and the first $r$ rows of $\hat{V}$ give $V$. Note that since $\hat{V}^T\tilde{V}$ is still uniformly random, the variance argument above follows for $\hat{V}^T A_{trn}^\dagger A_{trn}\hat{V}$. Additionally, for $i, j \leq r$, $(\hat{V}^T A_{trn}^\dagger A_{trn}\hat{V})_{ij} = (V^T A_{trn}^\dagger A_{trn}V)_{ij}$ Thus, we see that for $i, j \leq r$,

$$\mathbb{E}[((V^T A_{trn}^\dagger A_{trn}V)_{ij})^2] = \frac{1}{d-1}\left(c - c^2 + \frac{2}{cd} + o(1)\right).$$

Thus, finally, we have that arranged as a matrix $\mathbb{E}[QQ^T \odot QQ^T] = O\left(\frac{1}{d}\right)$. By an analogous symmetry argument, we can show that $\mathrm{Var}\left((V^T A_{trn}^\dagger A_{trn}V)_{ij}^2\right) = O\left(\frac{1}{d}\right)$. In principle, one can get a faster decay bound with a more sophisticated argument, but this is sufficient for our purposes. Now, by Lemma 5, we get that

$$\mathbb{E}[(QQ^T)^{-1}] = \frac{1}{1-c}I_r + O\left(\frac{1}{d}\right).$$

and each element has variance $O(1/d)$.

$\square$

**Lemma 15.** *For $c < 1$,*

$$\mathbb{E}\left[\Sigma_{trn}^{-1}H_1\Sigma_{trn}^{-1}\right] = \frac{1}{1-c}\Sigma_{trn}^{-2} + \frac{1}{\eta_{trn}^2}\frac{c}{1-c}I_r + o(1)$$

*and the variance of each element is $O\left(\frac{1}{d}\right)$. Additionally*

$$\mathbb{E}\left[\Sigma_{trn}H_1^{-1}\Sigma_{trn}\right] = (1-c)\eta_{trn}^2(\eta_{trn}^2\Sigma_{trn}^{-2} + cI_r)^{-1} + o(1),$$

*and the variance of each term is $O\left(\frac{1}{d}\right)$*

*Proof.* Recall that

$$H_1 = K^TK + Z^T(QQ^T)^{-1}Z = K^TK + Z^T\Sigma_{trn}^{-1}(\Sigma_{trn}(QQ^T)^{-1}\Sigma_{trn})\Sigma_{trn}^{-1}Z.$$

Using Lemmas 7, 12 and 14 along with an argument analogous to the one in Lemma 8, we get that

$$\mathbb{E}[\Sigma_{trn}^{-1}H_1\Sigma_{trn}^{-1}] = \frac{1}{1-c}\Sigma_{trn}^{-2} + \frac{1}{\eta_{trn}^2}\frac{c}{1-c}I_r + O\left(\frac{1}{d}\right) + o(1)$$

and the variance of each element is $O\left(\frac{1}{d}\right)$.

For the inverse, we define $\delta H_1 := H_1 - \mathbb{E}[H_1]$ and by an argument analogous to the one in the proof of Lemma 8, we get that

$$\mathbb{E}\left[\Sigma_{trn}H_1^{-1}\Sigma_{trn}\right] = (1-c)\eta_{trn}^2(\eta_{trn}^2\Sigma_{trn}^{-2} + cI_r)^{-1} + o(1)$$

and the variance of each term is $O\left(\frac{1}{d}\right)$.

$\square$

**Lemma 16.** *When $c < 1$, we have for $W = W_{opt}$ that*

$$\mathbb{E}[\|W\|_F^2] = \frac{c^2}{1-c}\mathrm{Tr}\left(\Sigma_{trn}^2\left(\Sigma_{trn}^2 + \frac{1}{\eta_{trn}^2}I_r\right)(\Sigma_{trn}^2c + \eta_{trn}^2I_r)^{-2}\right) + o(1).$$

*Proof.* Again, like in Lemma 9, we first use the estimates for the expectations from the lemmas above to get an estimate for the expectation of $\|W\|_F^2$, and then bound the deviation from it using the variance estimates in this section. We see that the first term in $\text{Tr}(W^T W)$ is

$$\text{Tr}((A_{trn}^\dagger)^T K (H_1^{-1})^T \Sigma_{trn}^2 H_1^{-1} K^T A_{trn}^\dagger) = \text{Tr}(K^T A_{trn}^\dagger (A_{trn}^\dagger)^T K (H_1^{-1})^T \Sigma_{trn}^2 H_1^{-1}).$$

Then using Lemma 13 along with cyclic invariance of traces, we see that this is estimated by

$$\frac{1}{\eta_{trn}^2} \frac{c^2}{(1-c)^3} \text{Tr}(\Sigma_{trn}(H_1^{-1})^T \Sigma_{trn}^2 H_1^{-1} \Sigma_{trn}) + o(1).$$

Then using Lemma 15, we get that this is estimated by

$$\eta_{trn}^2 \frac{c^2}{(1-c)^3} (1-c)^2 (cI_r + \eta_{trn}^2 \Sigma_{trn}^{-2})^{-2} + o(1)$$

$$= \eta_{trn}^2 \frac{c^2}{1-c} \text{Tr}\left(\Sigma_{trn}^4 (\Sigma_{trn}^2 c + \eta_{trn}^2 I_r)^{-2}\right) + o(1).$$

The second term is

$$\text{Tr}(((QQ^T)^{-1})^T Z (H_1^{-1})^T \Sigma_{trn}^2 H_1^{-1} Z^T (QQ^T)^{-1} HH^T).$$

We can rewrite this as

$$\text{Tr}(((QQ^T)^{-1})^T Z \Sigma_{trn}^{-1} (\Sigma_{trn}(H_1^{-1})^T \Sigma_{trn})(\Sigma_{trn} H_1^{-1} \Sigma_{trn}) \Sigma_{trn}^{-1} Z^T (QQ^T)^{-1} HH^T).$$

Using Lemmas 4 and 7, we can estimate its expectation by

$$\frac{1}{\eta_{trn}^2} \frac{c^2}{1-c} \text{Tr}\left(((QQ^T)^{-1})^T \Sigma_{trn}^{-1} (\Sigma_{trn}(H_1^{-1})^T \Sigma_{trn})(\Sigma_{trn} H_1^{-1} \Sigma_{trn}) \Sigma_{trn}^{-1} (QQ^T)^{-1}\right) + o(1).$$

Then using Lemma 14 and the fact that $H_1^T = H_1$, we get that this be further estimated by

$$\frac{1}{\eta_{trn}^2} \frac{c^2}{(1-c)^3} \text{Tr}(\Sigma_{trn}^{-1} (\Sigma_{trn}(H_1^{-1}) \Sigma_{trn})^2 \Sigma_{trn}^{-1}) + o(1).$$

Then using Lemma 15, we can simplify this estimate to

$$\frac{1}{\eta_{trn}^2} \frac{c^2}{(1-c)^3} (1-c)^2 \eta_{trn}^4 (cI_r + \eta_{trn}^2 \Sigma_{trn}^{-2})^{-2} + o(1)$$

$$= \eta_{trn}^2 \frac{c^2}{1-c} \text{Tr}\left(\Sigma_{trn}^2 (\Sigma_{trn}^2 c + \eta_{trn}^2 I_r)^{-2}\right) + o(1).$$

The cross term in $\text{Tr}(W^T W)$ is

$$-2 \text{Tr}((A_{trn}^\dagger)^T K (H_1^{-1})^T \Sigma_{trn}^2 H_1^{-1} Z^T (QQ^T)^{-1} H).$$

Here the term (after cyclically permuting) that we should focus on is

$$\text{Tr}(H (A_{trn}^\dagger)^T K) = -\text{Tr}(V_{trn}^T A_{trn}^\dagger (A_{trn}^\dagger)^T A_{trn}^\dagger \Sigma_{trn} U).$$

Here since $A_{trn} = \eta_{trn} \tilde{U} \tilde{\Sigma} \tilde{V}^T$ and $\tilde{U}, \tilde{V}$ are independent of each other, we see that using ideas from Lemma 8 in Sonthalia & Nadakuditi (2023) and extending them to rank $r$ as before, the expectation of this term is 0 with $O(1/d)$ variance. Thus, the whole cross-term has an expectation equal to 0.

Again, to bound the deviation from this estimate, note that for real valued random variables $X, Y$ we have that $|\mathbb{E}[XY] - \mathbb{E}[X]\mathbb{E}[Y]| = |Cov(X,Y)| \leq \sqrt{\text{Var}(X)\text{Var}(Y)}$. For real valued random variables $X, Y, Z, W$, we have the following fact, from Bohrnstedt & Goldberger (1969).

$$Cov(XY, WZ) = \mathbb{E}X\mathbb{E}W Cov(Y, Z) + \mathbb{E}Y\mathbb{E}Z Cov(X, W) + \mathbb{E}X\mathbb{E}Z Cov(Y, W) +$$
$$\mathbb{E}Y\mathbb{E}W Cov(X, Z) + Cov(X, W)Cov(Y, Z) + Cov(Y, W)Cov(X, Z).$$

We repeatedly apply these two to upper bound the deviation between the product of the expectations in the estimates above and the expectation of the product. It is then straightforward to see that since all variances are $O(1/d)$, the estimation error is $O(1/d) = o(1)$.

Finally, combining the terms, we get that

$$\mathbb{E}[\|W\|_F^2] = \frac{c^2}{1-c} \operatorname{Tr}\left( \Sigma_{trn}^2 \left( \Sigma_{trn}^2 + \frac{1}{\eta_{trn}^2} I_r \right) (\Sigma_{trn}^2 c + \eta_{trn}^2 I_r)^{-2} \right) + o(1).$$

$\square$

**Theorem 8.** *When $d < N - r$ and $\beta = I$, then the test error $\mathcal{R}(W, X_{tst})$ for $W = W_{opt}$ is given by*

$$\frac{\eta_{trn}^4}{N_{tst}} \| \left( \Sigma_{trn}^2 c + \eta_{trn}^2 I \right)^{-1} L \|_F^2$$

$$+ \frac{\eta_{tst}^2}{d} \frac{c^2}{1-c} \operatorname{Tr}\left( \Sigma_{trn}^2 \left( \Sigma_{trn}^2 + \frac{1}{\eta_{trn}^2} I_r \right) (\Sigma_{trn}^2 c + \eta_{trn}^2 I_r)^{-2} \right) + o\left( \frac{1}{d} \right).$$

*Proof.* Note that $\mathcal{R}(W, X_{tst}) = \frac{1}{N_{tst}} \| U \Sigma_{trn} H_1^{-1} Z^T (QQ^T)^{-1} \Sigma_{trn}^{-1} L \|_F^2 + \frac{\eta_{tst}^2}{d} \|W\|_F^2$.

To compute the first term, we observe that it is given by

$$\frac{1}{N_{tst}} Tr(U \Sigma_{trn} H_1^{-1} Z^T (QQ^T)^{-1} \Sigma_{trn}^{-1} LL^T \Sigma_{trn}^{-1} (QQ^T)^{-1} Z H_1^{-1} \Sigma_{trn} U^T).$$

This can be rewritten using cyclic invariance as

$$\frac{1}{N_{tst}} Tr(U^T U \Sigma_{trn} H_1^{-1} Z^T \Sigma_{trn}^{-1} \Sigma_{trn} (QQ^T)^{-1} \Sigma_{trn}^{-1} LL^T \Sigma_{trn}^{-1} (QQ^T)^{-1} \Sigma_{trn} \Sigma_{trn}^{-1} Z H_1^{-1} \Sigma_{trn}).$$

We apply Lemmas 14, 15 and 7 to get that its expectation can be estimated by

$$\frac{1}{N_{tst}} Tr\left( ((c-1)\eta_{trn}^2 (\eta_{trn}^2 I + c\Sigma_{trn}^2)^{-1})^2 \left( \frac{1}{1-c} \right)^2 LL^T \right) + o(1/d)$$

$$= \frac{\eta_{trn}^4}{N_{tst}} Tr\left( \left( \Sigma_{trn}^2 c + \eta_{trn}^2 I \right)^{-2} LL^T \right) + o(1/d)$$

$$= \frac{\eta_{trn}^4}{N_{tst}} \| \left( \Sigma_{trn}^2 c + \eta_{trn}^2 I \right)^{-1} L \|_F^2 + o(1/d).$$

We get $o\left( \frac{1}{d} \right)$ due to the $\Sigma_{trn}^{-2}$ term. Again, we can argue as in the proof of Lemma 16 to bound the deviation of the true expectation from this estimate by $o(1/d)$, noting that since train and test data assumptions are decoupled, $LL^T / N_{tst}$ can be treated as constant as $N$ grows.

Combining this with Lemma 9, we get that

$$\frac{\eta_{trn}^4}{N_{tst}} \| \left( \Sigma_{trn}^2 c + \eta_{trn}^2 I \right)^{-1} L \|_F^2$$

$$+ \frac{\eta_{tst}^2}{d} \frac{c^2}{1-c} \operatorname{Tr}\left( \Sigma_{trn}^2 \left( \Sigma_{trn}^2 + \frac{1}{\eta_{trn}^2} I_r \right) (\Sigma_{trn}^2 c + \eta_{trn}^2 I_r)^{-2} \right) + o\left( \frac{1}{d} \right).$$

$\square$

**Theorem 1** (In-Subspace Test Error). *Let $r < |d - N|$. Let the SVD of $X_{trn}$ be $U\Sigma_{trn} V_{trn}^T$, let $L := U^T X_{tst}$, $\beta_U := U^T \beta$, and $c := d/N$. Under our setup and Assumptions 1 and 2, the test error (Equation 1) is given by the following. If $c < 1$ (under-parameterized regime)*

$$\mathcal{R}(W_{opt}, UL) = \frac{\eta_{trn}^4}{N_{tst}} \left\| \beta_U^T (\Sigma_{trn}^2 c + \eta_{trn}^2 I)^{-1} L \right\|_F^2$$

$$+ \frac{\eta_{tst}^2}{d} \frac{c^2}{1-c} \operatorname{Tr}\left( \beta_U \beta_U^T \Sigma_{trn}^2 \left( \Sigma_{trn}^2 + \frac{1}{\eta_{trn}^2} I \right) (\Sigma_{trn}^2 c + \eta_{trn}^2 I)^{-2} \right) + o\left( \frac{1}{N} \right)$$

*If $c > 1$ (over-parameterized regime)*

$$\mathcal{R}(W_{opt}, UL) = \frac{\eta_{trn}^4}{N_{tst}} \left\| \beta_U^T (\Sigma_{trn}^2 + \eta_{trn}^2 I)^{-1} L \right\|_F^2$$
$$+ \frac{\eta_{tst}^2}{d} \frac{c}{c-1} \operatorname{Tr}(\beta_U \beta_U^T (I + \eta_{trn}^2 \Sigma_{trn}^{-2})^{-1}) + O\left( \frac{\|\Sigma_{trn}\|^2}{N^2} \right) + o\left( \frac{1}{N} \right)$$

*Proof.* The version for $\beta = I$ follows immediately from Theorem 7 and Theorem 8.

We now demonstrate how the the general version is a straightforward repetition of the proofs of the two theorems. First denote by $Z_{opt}$ the minimum norm solution to the denoising problem (where $\beta = I$). Then $Z_{opt} = X_{trn}(X_{trn} + A_{trn})^\dagger$ and note that

$$W_{opt} = Y_{trn}(X_{trn} + A_{trn})^\dagger = \beta^T X_{trn}(X_{trn} + A_{trn})^\dagger = \beta^T Z_{opt}$$

We present the adaptation of Lemma 9, the other lemmas can be adapted accordingly.

We first use the estimates for the expectations from the lemmas to get an estimate for $\|W_{opt}\|_F^2 = \|\beta^T Z_{opt}\|_F^2$, and then bound the deviation from it using the variance estimates above. We begin the calculation as

$$\|\beta^T Z_{opt}\|_F^2 = \operatorname{Tr}(Z_{opt}^T \beta \beta^T Z_{opt})$$

Using Lemma 1, we see that the trace has three terms. The first term is

$$\operatorname{Tr}(H^T (K_1^{-1})^T Z ((P^T P)^{-1})^T \Sigma_{trn}^T U^T \beta \beta^T U \Sigma_{trn} (P^T P)^{-1} Z^T K_1^{-1} H)$$

Using $\beta_U^T = \beta_{opt}^T U$ Then since the trace is invariant under cyclic permutations, we get the following term

$$\operatorname{Tr}(\beta_U^T \Sigma_{trn} (P^T P)^{-1} Z^T K_1^{-1} H H^T (K_1^{-1})^T Z ((P^T P)^{-1})^T \Sigma_{trn}^T \beta_U)$$

The rest of the proof for this term is the same as Lemma 9.

The second term in $\operatorname{Tr}(W^T \beta \beta^T W)$ is

$$-2 \operatorname{Tr}(H^T (K_1^{-1})^T Z^T ((P^T P)^{-1})^T \Sigma_{trn}^T \beta_U \beta_U^T \Sigma_{trn} Z^{-1} H H^T Z P^\dagger)$$

Then the rest of the proof for this term is identical to the one in the proof of Lemma 9.

Finally, the last term in $\operatorname{Tr}(W^T \beta \beta^T W)$ is

$$\operatorname{Tr}((P^\dagger)^T Z^T (K_1^{-1})^T H H^T (Z^{-1})^T \Sigma_{trn}^T \beta_U \beta_U^T \Sigma_{trn} Z^{-1} H H^T K_1^{-1} P^\dagger)$$

The rest of the proof is the same again, after using the cyclic invariance of the trace.

$\square$

## C.2 Proof of Corollary 1, The Distribution Shift Bound

We first prove Theorem 5, bounding the difference in generalization error in terms of the change in the test set. Recall the theorem below.

**Theorem 5** (Test Set Shift Bound). *Under the assumptions of Theorem 1, consider a linear regressor $W_{opt}$ trained on training data $X_{trn} = U \Sigma_{trn} V_{trn}^T$ with $\Sigma_{trn}$ such that $\sigma_r(X_{trn}) > M$, and tested on test data $X_{tst,1} = U L_1$ and $X_{tst,2} = U L_2$ with noise $A_{tst,1}, A_{tst,2}$ with the same variance $\eta_{tst}^2/d$. Then, the generalization errors $\mathcal{R}_1$ and $\mathcal{R}_2$ differ for $c < 1$ by*

$$|\mathcal{R}_2 - \mathcal{R}_1| \le \frac{\sigma_1(\beta)^2}{N_{tst}} \frac{\eta_{trn}^4 r}{(\sigma_r(X_{trn})^2 f(c) + \eta_{trn}^2)^2} \|L_2 L_2^T - L_1 L_1^T\|_F + o\left( \frac{1}{N} \right)$$

*where $f(c) = c$ for $c < 1$ and $f(c) = 1$ for $c \ge 1$. We add $O(\|\Sigma_{trn}\|_F^2/N^2)$ to the bound when $c > 1$.*

*Proof.* We will first show this for $c < 1$. Let $\mathcal{R}_i := \mathcal{R}(W_{opt}, X_{tst,i})$. Remember that the test error is given by

$$\mathcal{R}_i = \frac{\eta_{trn}^4}{N_{tst}} \left\| \beta_U^T (\Sigma_{trn}^2 c + \eta_{trn}^2 I)^{-1} L_i \right\|_F^2$$
$$+ \eta_{tst}^2 \eta_{trn}^2 \frac{1}{d} \frac{c^2}{1-c} \text{Tr} \left( \beta_U \beta_U^T \Sigma_{trn}^2 \left( \Sigma_{trn}^2 + \frac{1}{\eta_{trn}^2} I \right) (\Sigma_{trn}^2 c + \eta_{trn}^2 I)^{-2} \right) + o\left( \frac{1}{N} \right)$$

Note that the second term above has no dependence on $X_{tst,i}$, so the difference is given by

$$\mathcal{R}_2 - \mathcal{R}_1 = \frac{\eta_{trn}^4}{N_{tst}} \left( \left\| \beta_U^T (\Sigma_{trn}^2 c + \eta_{trn}^2 I)^{-1} L_2 \right\|_F^2 - \left\| \beta_U^T (\Sigma_{trn}^2 c + \eta_{trn}^2 I)^{-1} L_1 \right\|_F^2 \right)$$
$$+ o\left( \frac{1}{N} \right)$$
$$= \frac{\eta_{trn}^4}{N_{tst}} Tr \left( (\Sigma_{trn}^2 c + \eta_{trn}^2 I)^{-1} \beta_U \beta_U^T (\Sigma_{trn}^2 c + \eta_{trn}^2 I)^{-1} (L_2 L_2^T - L_1 L_1^T) \right) + o\left( \frac{1}{N} \right)$$
$$\overset{(i)}{\leq} \frac{\eta_{trn}^4}{N_{tst}} \| (\Sigma_{trn}^2 c + \eta_{trn}^2 I)^{-1} \beta_U \beta_U^T (\Sigma_{trn}^2 c + \eta_{trn}^2 I)^{-1} \|_F \| (L_2 L_2^T - L_1 L_1^T) \|_F + o\left( \frac{1}{N} \right)$$
$$= \frac{\eta_{trn}^4}{N_{tst}} \| \beta_U \beta_U^T (\Sigma_{trn}^2 c + \eta_{trn}^2 I)^{-2} \|_F \| (L_2 L_2^T - L_1 L_1^T) \|_F + o\left( \frac{1}{N} \right)$$
$$\overset{(ii)}{\leq} \frac{\eta_{trn}^4}{N_{tst}} \| \beta_U \beta_U^T \|_2 \| (\Sigma_{trn}^2 c + \eta_{trn}^2 I)^{-2} \|_F \| (L_2 L_2^T - L_1 L_1^T) \|_F + o\left( \frac{1}{N} \right)$$

where $(i)$ above is by the Cauchy-Schwarz inequality for the Frobenius norm and $(ii)$ above holds since $\|AB\|_F \leq \|A\|_2 \|B\|_F$. So, for $\Sigma_{trn}$ with lower bounded diagonal entries $\sigma_i > M$, we have that

$$|\mathcal{R}_2 - \mathcal{R}_1| \leq \frac{\eta_{trn}^4 r}{N_{tst} (\sigma_r(X_{trn})^2 c + \eta_{trn}^2)^2} \| \beta_U \beta_U^T \|_2 \| (L_2 L_2^T - L_1 L_1^T) \|_F + o\left( \frac{1}{N} \right)$$
$$= \frac{\eta_{trn}^4 r}{N_{tst} (\sigma_r(X_{trn})^2 c + \eta_{trn}^2)^2} \| U^T \beta \beta^T U \|_2 \| (L_2 L_2^T - L_1 L_1^T) \|_F + o\left( \frac{1}{N} \right)$$
$$= \frac{\eta_{trn}^4 r}{N_{tst} (\sigma_r(X_{trn})^2 c + \eta_{trn}^2)^2} \| \beta \beta^T \|_2 \| (L_2 L_2^T - L_1 L_1^T) \|_F + o\left( \frac{1}{N} \right)$$
$$= \frac{\sigma_1(\beta)^2}{N_{tst}} \frac{\eta_{trn}^4 r}{(\sigma_r(X_{trn})^2 c + \eta_{trn}^2)^2} \| L_2 L_2^T - L_1 L_1^T \|_F + o\left( \frac{1}{N} \right)$$

Similarly, for $c > 1$, we have that

$$|\mathcal{R}_2 - \mathcal{R}_1| \leq \frac{\sigma_1(\beta)^2}{N_{tst}} \frac{\eta_{trn}^4 r}{(\sigma_r(X_{trn})^2 + \eta_{trn}^2)^2} \| L_2 L_2^T - L_1 L_1^T \|_F + O\left( \frac{\|\Sigma_{trn}\|_F^2}{N^2} \right) + o\left( \frac{1}{N} \right)$$

$\square$

We now prove our corollary below.

**Corollary 1** (Distribution Shift Bound). *Consider a linear denoiser $W_{opt}$ trained on training data $X_{trn} = U\Sigma_{trn} V_{trn}^T$. Let it be tested on test data $X_{tst,1} = UL_1$ and $X_{tst,2} = UL_2$ generated possibly dependently from distributions supported in the span of $U$ with mean $U\mu_i$ and covariance $\Sigma_{U,i} = U\Sigma_i U^T$ respectively. Then, the difference in* generalization errors $\mathcal{G}_i := \mathbb{E}_{X_{tst,i}}[\mathcal{R}(W_{opt}, X_{tst,i})]$ *is bounded for $c < 1$ by*

$$|\mathcal{G}_2 - \mathcal{G}_1| \leq \frac{\sigma_1(\beta)^2 \eta_{trn}^4 r}{(\sigma_r(X_{trn})^2 f(c) + \eta_{trn}^2)^2} \| \Sigma_2 - \Sigma_1 + \mu_2 \mu_2^T - \mu_1 \mu_1^T \|_F + o\left( \frac{1}{N} \right)$$

*where $f(c) = c$ for $c < 1$ and $f(c) = 1$ otherwise. We add $O(\|\Sigma_{trn}\|_F^2 / N^2)$ when $c \geq 1$.*

*Proof.* Let $\bar{L}_i := L_i - [\mu_i \; \mu_i \; \ldots \; \mu_i]$ be the centered version of the test data matrix. In that case, $\mathbb{E}_{X_{tst,i}}[\bar{L}_i] = \mathbb{E}_{X_{tst,i}}[U^T \bar{X}_{tst,i}] = 0$ and

$$\mathbb{E}_{X_{tst,i}}[\bar{L}_i \bar{L}_i^T] = \mathbb{E}_{X_{tst,i}}[U^T \bar{X}_{tst,i} \bar{X}_{tst,i}^T U] = N_{tst} \Sigma_i$$

Now note the following elementary computation.

$$\begin{aligned}
\mathbb{E}_{X_{tst,i}}[L_i L_i^T] &= \mathbb{E}_{X_{tst,i}}[(\bar{L}_i + [\mu_i \; \mu_i \; \ldots \; \mu_i])(\bar{L}_i + [\mu_i \; \mu_i \; \ldots \; \mu_i])^T] \\
&= \mathbb{E}_{X_{tst,i}}[\bar{L}_i \bar{L}_i^T] + 0 + 0 + N_{tst} \mu_i \mu_i^T \\
&= N_{tst} \Sigma_{trn} + N_{tst} \mu_i \mu_i^T
\end{aligned}$$

We can now follow the initial part of the proof of Theorem 5 to get the following for $c < 1$.

$$\begin{aligned}
\mathcal{G}_2 - \mathcal{G}_1 &= \frac{\eta_{trn}^4}{N_{tst}} Tr\left(\beta_U \beta_U^T (\Sigma_{trn}^2 c + \eta_{trn}^2 I)^{-2} (\mathbb{E}_{X_{tst,2}}[L_2 L_2^T] - \mathbb{E}_{X_{tst,1}}[L_1 L_1^T])\right) + o\left(\frac{1}{N}\right) \\
&= \eta_{trn}^4 Tr\left(\beta_U \beta_U^T (\Sigma_{trn}^2 c + \eta_{trn}^2 I)^{-2} (\Sigma_2 - \Sigma_1 + \mu_2 \mu_2^T - \mu_1 \mu_1^T)\right) + o\left(\frac{1}{N}\right)
\end{aligned}$$

Now, we can follow the rest of the proof of Theorem 5 to complete the proof. $\qquad\square$

## C.3 Proofs for Theorem 2, Out-of-Subspace Generalization

**Theorem 2** (Out-of-Subspace Shift Bound). *If we have the same training data and solution $W_{opt}$ assumptions as in Theorem 1. Then, for **any** $X_{tst}$ for which there exists an $L$ and an $\alpha > 0$ such that $\|X_{tst} - UL\|_F \leq \alpha$, and $A_{tst}$ that satisfies 1,2 from Assumption 2, we have that the generalization error $\mathcal{R}(W_{opt}, X_{tst})$ satisfies*

$$|\mathcal{R}(W_{opt}, X_{tst}) - \mathcal{R}(W_{opt}, UL)| \leq \alpha^2 \sigma_1 (W_{opt} + I)^2.$$

*Proof.* Here we see that

$$\begin{aligned}
\|(I - W) X_{tst} - (I - W) UL\|_F^2 &= \|(I - W)(X_{tst} - UL)\|_F^2 \\
&\leq \sigma_1 (W - I)^2 \|X_{tst} - UL\|_F^2 \\
&= \alpha^2 \sigma_1 (W - I)^2
\end{aligned}$$

The inequality is due to Cauchy-Schwarz inequality. Then using the reverse triangle inequality, we have that

$$\left|\|(I - W) X_{tst}\|_F^2 - \|(I - W) UL\|_F^2\right| \leq \alpha^2 \sigma_1 (W + I)^2.$$

$\qquad\square$

## C.4 Proofs for Corollary 4, Generalization Error

**Corollary 4** (Generalization Error). *Let $r < |d - N|$. Let the SVD of $X_{trn}$ be $U \Sigma_{trn} V_{trn}^T$, let $L := U^T X_{tst}$, $\beta_U := U^T \beta$, and $c := d/N$. Under our setup and Assumptions 1 and 2, with the further assumption that the columns of $L$ are drawn IID from a distribution with mean $\mu$ and Covariance $\Sigma$, the test error (Equation 1) is given by the following.*
*If $c < 1$ (under-parameterized regime)*

$$\begin{aligned}
\mathbb{E}_L[\mathcal{R}(W_{opt}, UL)] = {}& \eta_{trn}^4 \left\| \beta_U^T (\Sigma_{trn}^2 c + \eta_{trn}^2 I)^{-1} (\Sigma + \mu \mu^T)^{1/2} \right\|_F^2 \\
& + \frac{\eta_{tst}^2}{d} \frac{c^2}{1 - c} Tr\left(\beta_U \beta_U^T \Sigma_{trn}^2 \left(\Sigma_{trn}^2 + \frac{1}{\eta_{trn}^2} I\right) (\Sigma_{trn}^2 c + \eta_{trn}^2 I)^{-2}\right) + o\left(\frac{1}{N}\right)
\end{aligned}$$

*If $c > 1$ (over-parameterized regime)*

$$\mathbb{E}_L[\mathcal{R}(W_{opt}, UL)] = \eta_{trn}^4 \left\| \beta_U^T (\Sigma_{trn}^2 + \eta_{trn}^2 I)^{-1} (\Sigma + \mu\mu^T)^{1/2} \right\|_F^2$$
$$+ \frac{\eta_{tst}^2}{d} \frac{c}{c-1} \operatorname{Tr}(\beta_U \beta_U^T (I + \eta_{trn}^2 \Sigma_{trn}^{-2})^{-1}) + O\left( \frac{\|\Sigma_{trn}\|^2}{N^2} \right) + o\left( \frac{1}{N} \right)$$

*Proof.* We begin by noting that the variance term is independent of $X_{tst}$. Hence we only need to focus on the bias term. Let $\bar{L} := L - [\mu\ \mu\ \dots\ \mu]$ be the centered version of the test data matrix. In that case, $\mathbb{E}_{X_{tst,i}}[\bar{L}] = \mathbb{E}_{X_{tst,i}}[U^T \bar{X}_{tst,i}] = 0$ and

$$\mathbb{E}_{X_{tst,i}}[\bar{L}\bar{L}^T] = \mathbb{E}_{X_{tst,i}}[U^T \bar{X}_{tst,i} \bar{X}_{tst,i}^T U] = N_{tst}\Sigma$$

Now note the following elementary computation.

$$\mathbb{E}_{X_{tst,i}}[LL^T] = \mathbb{E}_{X_{tst,i}}[(\bar{L} + [\mu\ \mu\ \dots\ \mu])(\bar{L} + [\mu\ \mu\ \dots\ \mu])^T]$$
$$= \mathbb{E}_{X_{tst,i}}[\bar{L}\bar{L}^T] + 0 + 0 + N_{tst}\mu\mu^T$$
$$= N_{tst}\Sigma_{trn} + N_{tst}\mu\mu^T$$

Consider the following sequence on computations about the bias term.

$$\mathbb{E}_{X_{tst}}\left[ \frac{\eta_{trn}^4}{N_{tst}} \left\| \beta_U^T (\Sigma_{trn}^2 c + \eta_{trn}^2 I)^{-1} L \right\|_F^2 \right]$$
$$= \frac{\eta_{trn}^4}{N_{tst}} Tr\left( \beta_U^T (\Sigma_{trn}^2 c + \eta_{trn}^2 I)^{-1} \mathbb{E}_{X_{tst}}[LL^T](\Sigma_{trn}^2 c + \eta_{trn}^2 I)^{-1} \beta_U \right)$$
$$= \frac{\eta_{trn}^4}{N_{tst}} Tr\left( \beta_U^T (\Sigma_{trn}^2 c + \eta_{trn}^2 I)^{-1} (\Sigma + \mu\mu^T)(\Sigma_{trn}^2 c + \eta_{trn}^2 I)^{-1} \beta_U \right)$$
$$= \frac{\eta_{trn}^4}{N_{tst}} \left\| \beta_U^T (\Sigma_{trn}^2 c + \eta_{trn}^2 I)^{-1} (\Sigma + \mu\mu^T)^{1/2} \right\|_F^2$$

This establishes our claim. □

### C.5   Proof for Theorem 4, Test Error for $W^*$

**Theorem 4** (Test Error for $W^*$). *In the same setting as Theorem 1, we have that* $W^* = \beta_U^T \left( I + \frac{\eta_{trn}^2}{c} \Sigma_{trn}^{-2} \right)^{-1} U^T$ *and*

$$\mathcal{R}(W^*, UL) = \frac{\eta_{trn}^4 N^2}{N_{tst} d^2} \left\| \beta_U^T \left( \Sigma_{trn}^2 + \frac{\eta_{trn}^2 N}{d} I \right)^{-1} L \right\|_F^2 + \frac{\eta_{tst}^2}{d} Tr\left( \beta_U \beta_U^T \left( I + \frac{\eta_{trn}^2 N}{d} \Sigma_{trn}^{-2} \right)^{-2} \right).$$

*Proof.* To prove the first part of the theorem, we first note that

$$\mathbb{E}_{A_{trn}} \left[ \|Y_{trn} - W(X_{trn} + A_{trn})\|_F^2 \right] = \|Y_{trn} - WX_{trn}\|_F^2 + \frac{\eta_{trn}^2 N}{d} \|W\|_F^2.$$

Solving this is equivalent to solving

$$\| [Y_{trn}\ \ 0] - W [X_{trn}\ \ \mu I] \|_F^2.$$

where $\mu^2 = \frac{\eta_{trn}^2 N}{d}$. We know from classical linear algebra that the solution to the above is

$$W^* = [\beta^T X_{trn}\ \ 0] [X_{trn}\ \ \mu I]^\dagger.$$

Using Lemmas 5 and 6 from Sonthalia et al. (2023), we have that if $X_{trn} = U\Sigma_{trn}V_{trn}^T$ where $U$ is $d$ by $d$, $\Sigma_{trn}$ is $d$ by $d$ and $V_{trn}$ is $N \times d$, then

$$
\begin{bmatrix} X_{trn} & \mu I \end{bmatrix} = U \underbrace{\begin{bmatrix} \sqrt{\sigma_1(X_{trn})^2 + \mu^2} & 0 & \cdots & & & & 0 \\ 0 & \ddots & & 0 & & & \\ \vdots & & \sqrt{\sigma_r(X_{trn})^2 + \mu^2} & & & & \vdots \\ & & 0 & \mu & 0 & & \\ & & & 0 & \ddots & 0 & \\ 0 & & & & & 0 & \mu \end{bmatrix}}_{\hat{\Sigma}} \begin{bmatrix} V_{trn}\Sigma_{trn}\hat{\Sigma}^{-1} \\ \mu U \hat{\Sigma}^{-1} \end{bmatrix}^T .
$$

Thus, we have that

$$
W^* = \begin{bmatrix} \beta^T U \Sigma_{trn} V_{trn}^T & 0 \end{bmatrix} \begin{bmatrix} V_{trn}\Sigma_{trn}\hat{\Sigma}^{-1} \\ \mu U \hat{\Sigma}^{-1} \end{bmatrix} \begin{bmatrix} \frac{1}{\sqrt{\sigma_1(X_{trn})^2+\mu^2}} & 0 & \cdots & & & & 0 \\ 0 & \ddots & & 0 & & & \\ \vdots & & \frac{1}{\sqrt{\sigma_r(X_{trn})^2+\mu^2}} & & & & \vdots \\ & & 0 & \frac{1}{\mu} & 0 & & \\ & & & 0 & \ddots & 0 & \\ 0 & & & & & 0 & \frac{1}{\mu} \end{bmatrix} U^T .
$$

Simplifying, we get

$$
W^* = \beta_U^T \Sigma_{trn}^2 \hat{\Sigma}^{-2} U^T
$$

$$
= \beta_U^T \begin{bmatrix} \frac{\sigma_1(X_{trn})^2}{\sigma_1(X_{trn})^2+\mu^2} & 0 & \cdots & & & 0 \\ 0 & \ddots & & 0 & & \\ \vdots & & \frac{\sigma_r(X_{trn})^2}{\sigma_r(X_{trn})^2+\mu^2} & & & \vdots \\ & & 0 & 0 & 0 & \\ & & & 0 & \ddots & 0 \\ 0 & & & & 0 & 0 \end{bmatrix} U^T
$$

$$
= \beta_U^T \Sigma_{trn}^2 (\Sigma_{trn}^2 + \mu^2 I)^{-1} U^T
$$

$$
= \beta_U^T \Sigma_{trn}^2 \left( \Sigma_{trn}^2 + \frac{\eta_{trn}^2 N}{d} I \right)^{-1} U^T
$$

$$
= \beta_U^T \left( I + \frac{\eta_{trn}^2 N}{d} \Sigma_{trn}^{-2} \right)^{-1} U^T
$$

Hence we have finished proving the first part.

For the second part, we note that similar to before, we need to calculate

$$
\frac{1}{N_{tst}} \mathbb{E}_{A_{tst}} \left[ \| Y_{tst} - W^*(X_{tst} + A_{tst}) \|_F^2 \right] = \frac{1}{N_{tst}} \| Y_{tst} - W^* X_{tst} \|_F^2 + \frac{\eta_{tst}^2}{d} \| W^* \|_F^2 .
$$

For the first term recall that $X_{tst} = UL$ and $Y_{tst} = \beta^T X_{tst}$. Hence we have that

$$
\frac{1}{N_{tst}} \| Y_{tst} - W^* X_{tst} \|_F^2 = \frac{1}{N_{tst}} \left\| \beta_U^T \left( I - \left( I + \frac{\eta_{trn}^2 N}{d} \Sigma_{trn}^{-2} \right)^{-1} \right) L \right\|_F^2 = \frac{1}{N_{tst}} \frac{\eta_{trn}^4 N^2}{d^2} \left\| \beta_U^T \left( \Sigma_{trn}^2 + \frac{\eta_{trn}^2 N}{d} \right)^{-1} L \right\|_F^2
$$

For the second term, we have that

$$\frac{\eta_{tst}^2}{d}\|W^*\|_F^2 = \frac{\eta_{tst}^2}{d}\operatorname{Tr}\left(\beta_U^T\left(I + \frac{\eta_{trn}^2 N}{d}\Sigma_{trn}^{-2}\right)^{-2}\beta_U\right) = \frac{\eta_{tst}^2}{d}\operatorname{Tr}\left(\beta_U\beta_U^T\left(I + \frac{\eta_{trn}^2 N}{d}\Sigma_{trn}^{-2}\right)^{-2}\right)$$

$\square$

### C.6 Proof for Corollary 2, Relative Excess Error

**Corollary 2** (Relative Excess Error). *Let* $\|\Sigma_{trn}\|_F^2 = \Omega(N^{1/2+\epsilon})$. *As* $d, N \to \infty$ *with* $d/N \to c$, *the relative excess error tends to* $\frac{c}{1-c}$ *in the underparametrized regime. In the overparametrized regime, when* $\|\Sigma_{trn}\|_F^2 = o(N)$, *it tends to* $\frac{1}{c-1}$ *and to* $\frac{1}{c-1} + k$ *for some constant* $k$ *when* $\|\Sigma_{trn}\|_F^2 = \Theta(N)$.

*Proof.* Recall from Theorem 4 that the test error for $W^*$ is given by

$$\mathcal{R}(W^*, UL) = \frac{\eta_{trn}^4 N^2}{d^2}\left\|\beta_U^T\left(\Sigma_{trn}^2 + \frac{\eta_{trn}^2 N}{d}I\right)^{-1}L\right\|^2 + \frac{\eta_{tst}^2}{d}Tr\left(\beta_U\beta_U^T\left(I + \frac{\eta_{trn}^2 N}{d}\Sigma_{trn}^{-2}\right)^{-2}\right)$$

We prove this for $c > 1$, the proof for $c < 1$ is analogous and in fact simpler. Notice that when $|\Sigma_{trn}\|_F^2 = \Omega(N^{1/2+\epsilon})$, in both $\mathcal{R}(W_{opt}, X_{tst})$ and $\mathcal{R}(W^*, X_{tst})$, the bias terms are $O(1/d^{1+2\epsilon})$ while the variance terms are $\Theta(1/d)$. In particular, as $d, N \to \infty$, with $d/N \to c$, the limit of the excess risk is given by only considering the variance terms and the estimation errors.

$$\lim_{d,N\to\infty, d/N\to c}\frac{\mathcal{R}(W_{opt}, X_{tst}) - \mathcal{R}(W^*, X_{tst})}{\mathcal{R}(W^*, X_{tst})}$$

$$= \lim_{d,N\to\infty, d/N\to c}\frac{\frac{\eta_{tst}^2}{d}Tr\left(\beta_U\beta_U^T\left(I + \frac{\eta_{trn}^2 N}{d}\Sigma_{trn}^{-2}\right)^{-2}\right) - \frac{\eta_{tst}^2}{d}\frac{c}{c-1}\operatorname{Tr}(\beta_U\beta_U^T(I + \eta_{trn}^2\Sigma_{trn}^{-2})^{-1})}{\frac{\eta_{tst}^2}{d}Tr\left(\beta_U\beta_U^T\left(I + \frac{\eta_{trn}^2 N}{d}\Sigma_{trn}^{-2}\right)^{-2}\right)}$$

$$+ \lim_{d,N\to\infty, d/N\to c}\frac{O\left(\frac{\|\Sigma_{trn}\|_F^2}{N^2}\right) + o\left(\frac{1}{N}\right)}{\frac{\eta_{tst}^2}{d}Tr\left(\beta_U\beta_U^T\left(I + \frac{\eta_{trn}^2 N}{d}\Sigma_{trn}^{-2}\right)^{-2}\right)}$$

$$= \lim_{d,N\to\infty, d/N\to c}\frac{Tr\left(\beta_U\beta_U^T\left(I + \frac{\eta_{trn}^2}{c}\Sigma_{trn}^{-2}\right)^{-2}\right) - \frac{c}{c-1}\operatorname{Tr}(\beta_U\beta_U^T(I + \eta_{trn}^2\Sigma_{trn}^{-2})^{-1})}{Tr\left(\beta_U\beta_U^T\left(I + \frac{\eta_{trn}^2}{c}\Sigma_{trn}^{-2}\right)^{-2}\right)}$$

$$+ \lim_{d,N\to\infty, d/N\to c}\frac{O\left(\frac{c\|\Sigma_{trn}\|_F^2}{N}\right) + o\left(c\right)}{\eta_{tst}^2 Tr\left(\beta_U\beta_U^T\left(I + \frac{\eta_{trn}^2}{c}\Sigma_{trn}^{-2}\right)^{-2}\right)}$$

$$= \lim_{d,N\to\infty, d/N\to c}\frac{Tr\left(\beta_U\beta_U^T\right) - \frac{c}{c-1}\operatorname{Tr}(\beta_U\beta_U^T)}{Tr\left(\beta_U\beta_U^T\right)} + \lim_{d,N\to\infty, d/N\to c}\frac{O\left(\frac{c\|\Sigma_{trn}\|_F^2}{N}\right) + o(1)}{\eta_{tst}^2 Tr\left(\beta_U\beta_U^T\right)}$$

$$= 1 - \frac{c}{c-1} + \lim_{d,N\to\infty, d/N\to c}O\left(\frac{\|\Sigma_{trn}\|_F^2}{N}\right)$$

$$= \begin{cases}\frac{1}{c-1} & ; \|\Sigma_{trn}\|_F^2 = o(N) \\ \frac{1}{c-1} + k & ; \|\Sigma_{trn}\|_F^2 = \Theta(N)\end{cases}$$

for some unknown problem-dependent constant $k$. This establishes the claim for $c > 1$, and the proof for when $c < 1$ is analogous and in fact simpler.

$\square$

## C.7 Proof for Corollary 6

**Corollary 6** (IID Test Data). *Let $r < |d - N|$. Let the SVD of $X_{trn}$ be $U\Sigma_{trn}V_{trn}^T$, let $L := U^T X_{tst}$, $\beta_U := U^T \beta$, and $c := d/N$. Under our setup and Assumptions 1 and 2, with the further assumption that the columns of $L$ are drawn IID from a distribution with mean zero and Covariance $\Sigma$, the test error (Equation 1) is given by the following.*
*If $c < 1$ (under-parameterized regime)*

$$\mathbb{E}_L[\mathcal{R}(W_{opt}, UL)] = \eta_{trn}^4 \left\| \beta_U^T (\Sigma_{trn}^2 c + \eta_{trn}^2 I)^{-1} \Sigma^{1/2} \right\|_F^2$$
$$+ \frac{\eta_{tst}^2}{d} \frac{c^2}{1-c} \operatorname{Tr}\left( \beta_U \beta_U^T \Sigma_{trn}^2 \left( \Sigma_{trn}^2 + \frac{1}{\eta_{trn}^2} I \right) (\Sigma_{trn}^2 c + \eta_{trn}^2 I)^{-2} \right) + o\left(\frac{1}{N}\right)$$

*If $c > 1$ (over-parameterized regime)*

$$\mathbb{E}_L[\mathcal{R}(W_{opt}, UL)] = \eta_{trn}^4 \left\| \beta_U^T (\Sigma_{trn}^2 + \eta_{trn}^2 I)^{-1} \Sigma^{1/2} \right\|_F^2$$
$$+ \frac{\eta_{tst}^2}{d} \frac{c}{c-1} \operatorname{Tr}(\beta_U \beta_U^T (I + \eta_{trn}^2 \Sigma_{trn}^{-2})^{-1}) + O\left(\frac{\|\Sigma_{trn}\|^2}{N^2}\right) + o\left(\frac{1}{N}\right)$$

*Proof.* Follows immediately from Corollary 4. $\square$

## C.8 Proofs for Theorem 6, IID Training Data With Isotropic Covariance

**Theorem 6** (I.I.D. Training Data With Isotropic Covariance). *Let $c = d/N$ and $c_r = r/N$. Then if $c < 1$*

$$\mathbb{E}_{X_{trn}}[\mathcal{R}] = \frac{\eta_{trn}^4}{N_{tst}} \|(\Sigma_{trn}^2 c + \eta_{trn}^2 I)^{-1} L\|_F^2$$
$$+ \eta_{tst}^2 \frac{r}{d} \frac{1}{1-c} \left( T_1(c_r, \eta_{trn}^2/c) + \frac{1}{\eta_{trn}^2} T_2(c_r, \eta_{trn}^2/c) \right) + o\left(\frac{1}{N}\right)$$

*and if $c > 1$*

$$\mathbb{E}_{X_{trn}}[\mathcal{R}] = \frac{\eta_{trn}^4}{N_{tst}} \|(\Sigma_{trn}^2 + \eta_{trn}^2 I)^{-1} L\|_F^2 + \eta_{tst}^2 \frac{r}{d} \frac{c}{c-1} T_3(c_r, \eta_{trn}^2) + O\left(\frac{1}{N}\right)$$

*where $T_1(c_r, z) = T_3(cr, z) - zT_2(cr, z)$, and*

$$T_2(c_r, z) = \frac{1 + c_r + zc_r}{2\sqrt{(1 - c_r + c_r z)^2 + 4c_r^2 z}} - \frac{1}{2}, \quad T_3(c_r, z) = \frac{1}{2} + \frac{1 + zc_r - \sqrt{(1 - c_r + zc_r)^2 + 4c_r^2 z}}{2c_r}.$$

*Proof.* Then if $X_{trn}$ is the data matrix, the singular values squared for $X_{trn}$ are the eigenvalues of

$$X_{trn}^T X_{trn} = Z^T U^T U Z = Z^T Z$$

Then $Z^T Z$ is a $N \times N$ matrix, and due to the normalization of the variance of the entries, this is a Wishart Matrix. Further, we know that the eigenvalue distribution can be approximated by the Marchenko Pastur distribution with shape parameter $r/N$ Marcenko & Pastur (1967); Götze & Tikhomirov (2011; 2003; 2004; 2005); Bai et al. (2003).

Then we have that for the $c < 1$ case, we have the variance is

$$\frac{1}{d} \frac{c}{1-c} \sum_{i=1}^r \frac{1}{c^2} \left( \frac{\sigma_i^4}{(\sigma_i^2 + \sigma_{trn}^2/c)^2} + \frac{1}{\sigma_{trn}^2} \frac{\sigma_i^2}{(\sigma_i^2 + \sigma_{trn}^2)^2} \right)$$

Then we simplify this as the following.

$$\frac{r}{d}\frac{1}{c(1-c)}\left(\mathbb{E}\left[\frac{\sigma_i^4}{(\sigma_i^2 + \sigma_{trn}^2/c)^2}\right] + \frac{1}{\sigma_{trn}^2}\mathbb{E}\left[\frac{\sigma_i^2}{(\sigma_i^2 + \sigma_{trn}^2)^2}\right]\right)$$

If $\lambda$ is an eigenvalue of the training data gram matrix, then the variance term of the generalization error has terms of the following form.

$$\frac{\lambda^2}{(\lambda + 1/c)^2}, \quad \frac{\lambda}{(\lambda + 1/c)^2}, \quad \frac{\lambda}{\lambda + 1}$$

The value of these for the Marchenko Pastur distribution can be found in Sonthalia et al. (2023).

$$\mathbb{E}\left[\frac{\lambda}{\lambda + \eta_{trn}^2}\right] = \frac{1}{2} + \frac{1 + \eta_{trn}^2 c_r - \sqrt{(1 - c_r + \eta_{trn}^2 c_r)^2 + 4c_r^2\eta_{trn}^2}}{2c_r}$$

$$\mathbb{E}\left[\frac{\lambda}{(\lambda + \eta_{trn}^2)^2}\right] = \frac{1 + c_r + \eta_{trn}^2 c_r}{2\sqrt{(1 - c_r + c_r\eta_{trn}^2)^2 + 4c_r^2\eta_{trn}^2}} - \frac{1}{2} + o(1)$$

$$\mathbb{E}\left[\frac{\lambda^2}{(\lambda + \eta_{trn}^2)^2}\right] = \mathbb{E}\left[\frac{\lambda}{\lambda + \eta_{trn}^2}\right] - \eta_{trn}^2\left(\mathbb{E}\left[\frac{\lambda}{(\lambda + \eta_{trn}^2)^2}\right]\right)$$

$c_r = r/N$

The proofs for the rest of the terms are similar. $\qquad\square$

### C.9 Proofs for Corollary 7, IID Training and Test Data With Isotropic Covariance

**Corollary 7** (I.I.D. Train and Tests Data With Isotropic Covariance). *Let $c = d/N$ and $c_r = r/N$. Then if $c < 1$*

$$\mathbb{E}_{X_{trn}}[\mathcal{R}] = \eta_{trn}^4 \cdot r \cdot \kappa \cdot T_4(c_r, \eta_{trn}^2/c)$$
$$+ \frac{r}{d}\frac{1}{1-c}\left(T_1(c_r, \eta_{trn}^2/c) + \frac{1}{\eta_{trn}^2}T_2(c_r, \eta_{trn}^2/c)\right) + o\left(\frac{1}{N}\right)$$

*and if $c > 1$*

$$\mathbb{E}_{X_{trn}}[\mathcal{R}] = \eta_{trn}^4 \cdot r \cdot \kappa \cdot T_4(c_r, \eta_{trn}^2) + \frac{r}{d}\frac{c}{c-1}T_3(c_r, \eta_{trn}^2) + O\left(\frac{1}{N}\right)$$

*where $T_1(c_r, z) = T_3(c_r, z) - zT_2(c_r, z)$, and*

$$T_2(c_r, z) = \frac{1 + c_r + zc_r}{2\sqrt{(1 - c_r + c_r z)^2 + 4c_r^2 z}} - \frac{1}{2}, \quad T_3(c_r, z) = \frac{1}{2} + \frac{1 + zc_r - \sqrt{(1 - c_r + zc_r)^2 + 4c_r^2 z}}{2c_r},$$

$$T_4(c_r, z) = \frac{zc_r^2 + c_r^2 + zc_r - 2c_r + 1}{2z^2 c_r\sqrt{(1 - c_r + c_r z)^2 + 4c_r^2 z}} - \frac{1}{2z^2}\left(1 - \frac{1}{c_r}\right).$$

*Proof.* For the bias, we get

$$\frac{\eta_{trn}^4}{N_{tst}}\frac{1}{c^2}\mathbb{E}\left[\frac{1}{(\sigma_i^2 + \eta_{trn}^2/c)^2}\right]\|L\|_F^2$$

The value of these for the Marchenko Pastur distribution can be found in Sonthalia et al. (2023).

$$\mathbb{E}\left[\frac{1}{(\lambda + \eta_{trn}^2)^2}\right] = \frac{\eta_{trn}^2 c_r^2 + c_r^2 + \eta_{trn}^2 c_r - 2c_r + 1}{2\eta_{trn}^4 c_r\sqrt{4\eta_{trn}^2 c_r^2 + (1 - c_r + \eta_{trn}^2 c_r)^2}} + \frac{1}{2\eta_{trn}^4}\left(1 - \frac{1}{c_r}\right)$$

$\qquad\square$

# D  Numerical Details

In this section, we include the computational details required to reproduce the data and figures in the paper. The code for the experiments can be found in the following anonymized repository [Link].

**Data**   For our transfer learning results, we use real datasets namely CIFAR Krizhevsky (2009), STL10 Coates et al. (2011) and SVHN Netzer et al. (2011). We will mostly be working with the training and test split of CIFAR, training split of STL10 and training split of SVHN. We will also use the test split of STL10 for our data augmentation results, refer figure 7h and section D.3, to avoid overlaps between training and test data.

To verify the application of our results to I.I.D. data, we generate datasets from certain distributions, the details of which are presented in the upcoming sections.

The test data is normalized so that each coordinate has mean zero and a standard deviation of 5. This is done before we do any other pre-processing.

**Compute Time**   For figures 2, 1, 9 and 8, we use the same training data from CIFAR train split. Thus, we combine our code implementation for these figures. This saves up compute time for mean empirical error since inversion of the matrix $X_{trn} + A_{trn}$, for obtaining $W_{opt}$, occurs once for each empirical run for all 4 figures. The code was implemented using Google Colab with A100 Nvidia GPU which took approximately 1 hour for the 200 trials for each value of r. Since the results are computed for 4 values of r, the entire experiment was completed within approximately 4 hours.

Figures 7 and 7h took approximately 4 hours each using A100 Nvidia GPU on Google Colab. Figures 5 and 6 were computed together in approximately 40 minutes. Figure 10 took approximately 1 hour to compute. Figure 11 only took around 10 minutes due to less number of $N$ values and only 50 trials. All the above was implemented using A100 GPU on Colab. Figure 2c took approximately 4.5 hours using T4 Nvidia GPU on Google Colab.

## D.1  Principal Component Regression

We use four datasets for the set of results obtained through principal component regression namely, CIFAR train split, CIFAR test split, STL10 dataset and SVHN dataset.

**In-Subspace**   For figure 2a, the test data lies in the same low-dimensional subspace as the training dataset. The experimental setting is as follows.

- Training data, of order $d \times N$, is sampled from flattened CIFAR train split such that $d = 3072$ and N ranges between 1050 and 10500 with an increment of 550 for the results.
- We project our training data over the first $r$ principal components where $r$ refers to the rank and varies as 25, 50, 100 and 150.
- Test datasets, of order $d \times N_{tst}$, are sampled from CIFAR test split, STL10 train split and SVHN train split where $d = 3072$ and $N_{tst} = 2500$.
- We also project these test datasets onto the low-dimensional subspace using the projection matrices.
- For denoising, we generate Gaussian noise matrix $A_{trn}$ with norm $\sqrt{N}$ for the training data and $A_{tst}$ with norm $\sqrt{N_{tst}}$ for the test datasets.

The theoretical error is calculated using the formula in Theorem 1 and the empirical error is the mean squared error.

**Out-of-Subspace**   Next, we test our formulas for test datasets which lie outside the training distribution space.

**Small $\alpha$**   We detail the numerical setup required to generate figure 2b.

- Training data, of order $d \times N$, is sampled from flattened CIFAR train split such that $d = 3072$ and N ranges between 1050 and 10500 with an increment of 550 for the results.
- We project our training data over the first $r$ principal components where $r$ refers to the rank and varies as 25, 50, 100 and 150.

- Test datasets, of order $d \times N_{tst}$, are sampled from CIFAR test split, STL10 train split and SVHN train split where $d = 3072$ and $N_{tst} = 2500$.
- We project these test datasets onto the low-dimensional subspace using the projection matrices.
- We add a small amount of full-dimensional Gaussian noise to the projected datasets to generate out-of-subspace datasets with small $\alpha$. Here, we consider the case where $\alpha = 0.1$.
- For denoising, we generate Gaussian noise matrix $A_{trn}$ with norm $\sqrt{N}$ for the training data and $A_{tst}$ with norm $\sqrt{N_{tst}}$ for the test datasets.

The empirical error shown in figure 2b is the square root of the mean squared error. The theoretical bounds on the error are calculated using Theorem 2.

**Large $\alpha$.** For figure 8, the experimental setup is as follows.

- Training data, of order $d \times N$, is sampled from flattened CIFAR train split such that $d = 3072$ and N ranges between 1050 and 10500 with an increment of 550 for the results.
- We project our training data over the first $r$ principal components where $r$ refers to the rank and varies as 25, 50, 100 and 150.
- Test datasets, of order $d \times N_{tst}$, are sampled from CIFAR test split, STL10 train split and SVHN train split where $d = 3072$ and $N_{tst} = 2500$.
- We do not project these test datasets onto the low-dimensional subspace. We retain their high dimensions. The values of $\alpha$ for different values of $r$ are provided in figure 8.
- For denoising, we generate Gaussian noise matrix $A_{trn}$ with norm $\sqrt{N}$ for the training data and $A_{tst}$ with norm $\sqrt{N_{tst}}$ for the test datasets.

### D.2 Linear Regression

To consider the linear regression case for figure 1,

- Training data, of order $d \times N$, is sampled from flattened CIFAR train split such that $d = 3072$ and N ranges between 1050 and 10500 with an increment of 550 for the results.
- We project our training data over the first $r$ principal components where $r$ refers to the rank and varies as 25, 50, 100 and 150.
- Gaussian noise matrix with norm $\sqrt{N}$ is added to the training data.
- We generate normally-distributed $\beta_{opt}$ of order $d \times 1$ with norm 1. The learned estimator is computed as $\beta^T = \beta_{opt}^T W$ where $W$ is the minimum norm solution to the least squares denoising problem. For theoretical error, we compute $\hat{\beta}^T = \beta_{opt} U$.
- Test datasets, of order $d \times N_{tst}$, are sampled from CIFAR test split, STL10 train split and SVHN train split where $d = 3072$ and $N_{tst} = 2500$.
- We also project these test datasets onto the low-dimensional subspace using the projection matrices.
- Gaussian noise matrix with norm $\sqrt{N_{tst}}$ is added to the test datasets.
- Finally, the test datasets, $X_{tst}$, are replaced with $\beta^T X_{tst}$ to compute the error for the linear regression problem.

### D.3 Data Augmentation

To emphasize the application of our results to non-I.I.D. data, we consider two cases of data augmentation to our training data.

**Without Independence** The experimental setting to obtain the empirical generalization error is as follows.

- We sample 1000 images from the CIFAR train split as the first batch of our training data. For experimental results
- We augment the above batch with the same batch to vary $N$ between 1000 and 6000 with an increment of 1000. We project the dataset onto its first $r$ principal components where $r = 25, 50, 100$ and 150.
- We add gaussian noise with norm $\sqrt{N}$ to the training data as before. Note that the noise on augmented batches would be independent of the noise in the original batch. This is the only assumption required for our result.

- Test datasets, of order $d \times N_{tst}$, sampled from CIFAR test split, STL10 train split and SVHN train split where $d = 3072$ and $N_{tst} = 2500$ are also projected onto the low-dimensional subspace.

We calculate the theoretical generalization error for more values of $c$ to obtain smoother curves. Note that the left singular vectors i.e., the columns of matrix $U$, do not change when we augment our training batches. We utilize this to speed-up our computation for theoretical curves.

- We sample 1000 images from the CIFAR train split as the first batch of our training data.
- We obtain the projection matrix $P = UU^T$ and the matrix $L = U^T X_{tst}$ from the SVD of the first batch itself.
- The generalization error is computed from the formula in Theorem 1 for values of $N$ between 1000 and 6000 with an increment of 50.
- We scale the singular values by a factor of $N/1000$ to account for the augmenting.

**Without Identicality**  To generate figure 7h,

- We use training data, of order $d \times N$, such that $d = 3072$ and N ranges between 1050 and 10500 with an increment of 550 for the results.
- We use $N/2$ images from the CIFAR training split and $N/2$ images from the STL10 training split concatenated together for our training data.
- We project our training data over the first $r$ principal components where $r$ refers to the rank and varies as 25, 50, 100 and 150.
- Test datasets, of order $d \times N_{tst}$, are sampled from CIFAR test split, STL10 test split and SVHN train split where $d = 3072$ and $N_{tst} = 2500$. This is done to avoid any overlaps between training and test data.
- We also project these test datasets onto the low-dimensional subspace using the projection matrices.
- For denoising, we generate Gaussian noise matrix $A_{trn}$ with norm $\sqrt{N}$ for the training data and $A_{tst}$ with norm $\sqrt{N_{tst}}$ for the test datasets.

### D.4  I.I.D. Data

We also perform experiments to verify our results in cases where training and test datasets are I.I.D. The numerical details for those experiments are presented in this section.

**I.I.D. Test Data**  To generate figure 9,

- Training data, of order $d \times N$, is sampled from flattened CIFAR train split such that $d = 3072$ and N ranges between 1050 and 10500 with an increment of 550 for the results.
- We project our training data over the first $r$ principal components where $r$ refers to the rank and varies as 25, 50, 100 and 150.
- We generate $L$ from Gaussian distribution of norm $\sqrt{N_{tst}}$ where $N_{tst} = 2500$.
- We obtain our I.I.D. test data of order $d \times N_{tst}$ as $X_{tst} = UL$ where $U$ contains the left singular vectors of the projected training data.
- For denoising, we generate Gaussian noise matrix $A_{trn}$ with norm $\sqrt{N}$ for the training data and $A_{tst}$ with norm $\sqrt{N_{tst}}$ for the test datasets.

**I.I.D. Train Data**  To generate figure 10,

- We generate the left singular matrix $U$ from the SVD of a Gaussian matrix of order $d \times r$ where $M = 3072$ and $r = 50$.
- We generate the training matrix $X_{trn} = UZ$ where $Z$ is of order $r \times N$ such that each column is normally distributed with mean 0 and variance $1/r$.
- Here, $N$ varies from 1050 to 10500 with an increment of 550.
- Test datasets, of order $d \times N_{tst}$, are sampled from CIFAR test split, STL10 train split and SVHN train split where $d = 3072$ and $N_{tst} = 2500$.
- We also project these test datasets onto the $r$-dimensional subspace using projection matrices.
- For denoising, we generate Gaussian noise matrix $A_{trn}$ with norm $\sqrt{N}$ for the training data and $A_{tst}$ with norm $\sqrt{N_{tst}}$ for the test datasets.

**I.I.D Train and Test Data**  To generate figure 11,

- We generate the left singular matrix $U$ from the SVD of a Gaussian matrix of order $d \times r$ where $M = 3072$ and $r = 50$.
- We generate the training matrix $X_{trn} = UZ$ where $Z$ is of order $r \times N$ such that each column is normally distributed with mean 0 and variance $1/r$.
- Here, $N$ varies from 500 to 6010 with an increment of 550 for the empirical markers and with an increment of 55 for theoretical values on the solid curve.
- We generate $L$ from Gaussian distribution of norm $\sqrt{N_{tst}}$ where $N_{tst} = 5000$.
- We obtain our I.I.D. test data of order $d \times N_{tst}$ as $X_{tst} = UL$ where $U$ contains the left singular vectors of the projected training data.
- For denoising, we generate Gaussian noise matrix $A_{trn}$ with norm $\sqrt{N}$ for the training data and $A_{tst}$ with norm $\sqrt{N_{tst}}$ for the test datasets.

### D.5 Full Dimensional Denoising

To generate figure 2c,

- Training data, of order $d \times N$, is sampled from flattened CIFAR train split such that $d = 3072$ and N ranges between 1050 and 10500 with an increment of 550 for the results.
- We project our training data over the first $r$ principal components where $r$ is the minimum of $d$ and $N$. This implies that the data is full dimensional.
- Test datasets, of order $d \times N_{tst}$, are sampled from CIFAR test split, STL10 train split and SVHN train split where $d = 3072$ and $N_{tst} = 2500$.
- We also project these test datasets onto the low-dimensional subspace using the projection matrices.
- For denoising, we generate Gaussian noise matrix $A_{trn}$ with norm $\sqrt{N}$ for the training data and $A_{tst}$ with norm $\sqrt{N_{tst}}$ for the test datasets.

### D.6 Optimal $\eta_{trn}$

To generate figures 5 and 6,

- Training data, of order $d \times N$, is sampled from flattened CIFAR train split such that $d = 3072$ and N ranges between 500 and 5500 as {500, 750, 1000, 1250, 1500, 1750, 2000, 2250, 2500, 2600, 2700, 2800, 2900, 3000, 3020, 3130, 3200, 3300, 3400, 3500, 3750, 4000, 4250, 4500, 4750, 5000, 5250, 5500}.
- We project our training data over the first $r$ principal components where $r = 50$.
- Test datasets, of order $d \times N_{tst}$, are the training dataset with new noise and sampled from CIFAR test split, STL10 train split and SVHN train split where $d = 3072$ and $N_{tst} = N$.
- We compute generalization error for 2000 $\eta_{trn}$ values ranging from $1/3.5$ to 100 for each $N$ from our formula in Theorem 1.
- We report the optimal $\eta_{trn}$ found to minimise the generalization error in figure 5 and the optimal generalization error in figure 6.

