# OpenReview forum: "Double Descent and Overfitting under Noisy Inputs and Distribution Shift for Linear Denoisers"
_TMLR — Accepted by TMLR_

### Review · Reviewer_brba · 2024-02-10

**Summary Of Contributions:**

This paper studies the machine-learning problem of supervised denoising and noisy-input regression under distribution shifts. Based on a couple of assumptions about the training/testing data distributions (e.g., low-rank) and noise distributions (e.g., Marchenko-Pastur distribution), the paper presents a sequence of theoretical results regarding the generalization error bound. Empirically, the paper demonstrates the double-descent phenomenon on image denoising tasks using benchmark datasets, which verifies their theoretical results when there is a distribution shift.

**Audience:**

Yes

**Broader Impact Concerns:**

No ethical concerns since the paper is mainly theoretical.

**Claims And Evidence:**

Yes

**Requested Changes:**

1. Clarify why assuming noisy inputs and clean outputs for regression tasks.

2. Clarify why restricting to linear relationship for denoising tasks.

3. Provide clarifications to my questions related to the imposed assumptions and theoretical results.

4. Minor concerns about presentations: The theoretical results are not easy to parse - it would be helpful to provide more intuitive (informal) statements of your main theoretical results for better understanding.

5. A typo: On page 3, there is a question mark.

**Strengths And Weaknesses:**

The paper clearly presents the mathematical formulations of different problems and their required assumptions, with detailed discussions and related work comparisons. The proposed theoretical framework is general and seems applicable to a broad family of denoising and regression problems. Under distribution shifts, the paper proves the generalization error bound and how it implies the double descent phenomenon. The paper mainly focuses on theoretical topics but also provides experimental results on image benchmark datasets to verify their theoretical findings.

Although I can see a connection between regression tasks and denoising tasks, why they can be unified in a single framework requires further clarification. While reading the paper, I sometimes find it difficult to switch between regression and denoising tasks, as they are motivated from different perspectives and applied to different applications. In particular, for linear regression tasks, it is typical to consider that arbitrary or structured noises corrupt the outputs. However, you mainly focus on the setting with noisy inputs but clean outputs, so I'd like to know whether this setting is worth studying for regression tasks. For denoising tasks, it needs to be clarified why we should restrict the relationship between the noisy inputs and their clean counterparts to be linear.

Moreover, I have the following questions about the imposed assumptions and theoretical results. Regarding the low-rank condition of the data under distribution shift scenarios specified in Assumption 1, it is unclear why the training data and testing data should share the same (or almost similar) linear subspace. In addition, the last condition specified regarding noise requires $d/N = c + o(1)$ as $N$ grows, whereas $d$ is usually a constant for real-world denoising tasks. So, I’m not sure if this is a realistic assumption for modeling the noises. For the theoretical results presented in Theorem 2 and Theorem 3, I fail to understand why they imply a double descent phenomenon. Can you elaborate further?

---

> ### Author Response · Authors · 2024-02-16
> **Response - Part 1**
>
> We thank the reviewer for the comments, feedback, and questions.
>
> > Although I can see a connection between regression tasks and denoising tasks, why they can be unified in a single framework requires further clarification. While reading the paper, I sometimes find it difficult to switch between regression and denoising tasks, as they are motivated from different perspectives and applied to different applications.
>
> This is a good question. The general setup for our problem is as follows. There is a linear function of the data $y = \beta^T x$ that is the true data-generating process. Here $\beta \in \mathbb{R}^{d \times k}$ so can be multivariate.
>
> During training, we, however, do observe $(x_1, y_1), \ldots, (x_n, y_n)$. We instead observe $(x_1+noise, y_1), \ldots, (x_n+noise, y_n)$. So, we observe noisy versions of the independent variable $x$ and noiseless versions of the dependent variable $y$.
>
> Hence, this can be thought of as a version of linear regression with noise in the independent variables. Similar forms of regression have been previously studied and are usually referred to as errors in variable regression. These consider both noise in both inputs and outputs, while we only consider noise in inputs. We discuss later in the response why this is reasonable. We also believe this is reasonable as, for many applications, we have noisy samples of the independent and dependent variables.
>
> **Denoising**: We would like the reviewer to think of denoising as a special case of the above error in variable regression. Specifically, this is the case when the true data-generating function is the identity $\beta = I$. That $y = x$. However, since we observe noisy versions of $x$, this becomes the denoising problem.
>
> > In particular, for linear regression tasks, it is typical to consider that arbitrary or structured noises corrupt the outputs. However, you mainly focus on the setting with noisy inputs but clean outputs, so I'd like to know whether this setting is worth studying for regression tasks. For denoising tasks, it needs to be clarified why we should restrict the relationship between the noisy inputs and their clean counterparts to be linear.
>
> As the reviewer mentions, studying the case when we have noisy outputs is typical. However, we do not think studying the noisy input case is unreasonable. This has been studied before and is usually referred to as an error in variable regression.
>
> Also, consider the following scientific motivation. We conduct some biological experiments where we measure various quantities such as heart rate, oxygen levels, and various other physiological properties, and would like to predict whether a person is standing, sitting, or lying down. In this case, the inputs such as heart rate, oxygen levels etc are not exact values and have noise. However, the labels we are trying to predict are fairly clean.
>
> **Linear relationship for denoising**. We assume a linear relationship between the *noiseless* inputs and the outputs, and **not** between the noisy inputs and outputs. The latter would be quite unreasonable, as the reviewer points out. So, for the denoising case, the linear relationship between the *noiseless* inputs and the outputs is just $y = I x$.
>
> > Moreover, I have the following questions about the imposed assumptions and theoretical results. Regarding the low-rank condition of the data under distribution shift scenarios specified in Assumption 1, it is unclear why the training data and testing data should share the same (or almost similar) linear subspace.
>
> In practice, the reviewer is correct –  the training and test data need not live in the same low-dimensional subspace. However, this is an assumption that we need for Theorem 1. Theorem 2 allows us to provide some bounds in the case when they do not live in the same subspace.

---

> > ### Author Response · Authors · 2024-02-16
> > **Response - Part 2**
> >
> > > In addition, the last condition specified regarding noise requires $d/N = c + o(1)$ as $N$ grows, whereas $d$ is usually a constant for real-world denoising tasks. So, I’m not sure if this is a realistic assumption for modeling the noises.
> >
> > The reviewer is correct that for real data $d$ is constant. However, in many cases, $N$ is also roughly fixed. Maybe with some work we can double or triple the amount of data points. But it is very unlikely that we are going to go from having $N$ to $N^2$ data points. Hence, $d$ and $N$ are roughly proportional for many real tasks.
> > Also note that we are merely controlling the generalization error in terms of the size of d and N. The little o notation is used to reflect that the deviation from our expression is small when this "roughly fixed" order of d and N is large. Further, as seen in Figures 1, 2, 3, and 4, the error terms are small, and the theoretical predictions align with the experiments.
> >
> > > or the theoretical results presented in Theorem 2 and Theorem 3, I fail to understand why they imply a double descent phenomenon. Can you elaborate further?
> >
> > For Theorem 3, the term on the third line of each expression has a 1/(|c-1|) in the denominator. Hence giving us a double descent.
> > For Theorem 2, we bound the difference between the true risk and the risk predicted in Theorem 1. In particular, this gives a lower bound for the true risk - risk from theorem 1 minus $\alpha^2 \sigma_1(W+I)$. Note that this lower bound goes to infinity at c = 1 for small $\alpha$. This suggests a peak at c = 1.
> >
> > **Response to Requested Changes**
> >
> > > Clarify why assuming noisy inputs and clean outputs for regression tasks.
> >
> > We hope that the above response clarifies why our assumptions are reasonable. Additionally, to get to the complete error in variable regression, we must add the output noise back. This is a comparatively simple step, given our paper and prior work.
> >
> > > Clarify why restricting to linear relationship for denoising tasks.
> >
> > We believe that this might be a misunderstanding. We do not assume that the "relationship between the noisy inputs and their clean counterparts is linear," as the review says. We assume that the **noiseless** inputs and their clean counterparts have a linear relationship. This is the standard linear modeling assumption.
> >
> > > Provide clarifications to my questions related to the imposed assumptions and theoretical results.
> >
> > We hope that our response provides the requested clarification.
> >
> > > Minor concerns about presentations: The theoretical results are not easy to parse - it would be helpful to provide more intuitive (informal) statements of your main theoretical results for better understanding.
> >
> > We provide some intuition after the statement of Theorem 1 in terms of a split into bias and variance terms, and the insights section provides some insights about this split  as well as other aspects of the theorem. We believe that adding informal statements can be misconstrued for results as subtle as this one, and we would like to avoid that. Please let us know if you are not satisfied.
> >
> > > A typo: On page 3, there is a question mark.
> >
> > Fixed
> >
> >
> > If the reviewer still has concerns, please let us know.

---

> ### Comment · Reviewer_brba · 2024-02-22
>
> Thanks for responding to my comments. Let me clarify my comment, "Clarify why restricting to linear relationship for denoising tasks," which I think the authors might have misunderstood. I do not think your response "We would like the reviewer to think of denoising as a special case of the above error in variable regression. Specifically, this is the case when the true data-generating function is the identity $\beta = I$" answers my question.
>
> For image-denoising tasks, the goal is to learn a generalizable function to map the noisy image to the corresponding clean image. Based on the optimization problem introduced in Section 2, you restrict this function to be linear. For denoising autoencoders, this essentially suggests that the encoder and decoder are both linear, which I do not understand why. As far as I know, the state-of-the-art image denoisers are highly non-linear. I would like to know why studying linear denoisers is important.

---

> > ### Author Response · Authors · 2024-02-26
> > **Response**
> >
> > Thank you for that clarification. We did, indeed, previously misunderstand the comment.
> >
> > The reviewer is correct that the encoder and decoder are linear in our setup. The reviewer is also correct that state-of-the-art methods are highly non-linear.
> >
> > However, even the understanding of linear systems is incomplete. We believe that our paper helps understand the linear case.
> > We hope that understanding the linear case can act as an important step in theoretically understanding non-linear models.

---

### Review · Reviewer_3dcE · 2024-02-12

**Summary Of Contributions:**

This paper provides a theoretical upper bound on the test error of linear regression problems in the scenario of noisy data but noiseless labels. The curve of the upper bound shows a double-descent-like shape. The paper also provides numerical verification on the test error bound, as well as some insights on the role of data noise and data augmentation.

**Audience:**

Yes

**Broader Impact Concerns:**

no concerns

**Claims And Evidence:**

Yes

**Requested Changes:**

See "weaknesses" and "questions"

**Strengths And Weaknesses:**

Strength:

The setting of *noisy* data but noiseless labels is relatively rarely studied in the literature. The role of noise in the data has not been well understood. It is an interesting topic to theoretically analyze the effect of data noise, including its effect on the test error/loss as discussed in this paper.

The test error bound exhibits a perfect double descent curve. It is also well matched by numerical verification on some real data.

The observation of the data noise having a regularization effect is interesting.

Weaknesses:

The analysis is limited to the very simple linear regression problem. It remains unclear in non-linear problems. The analysis techniques seem not to extend to more complicated problems.

The title of the paper should be more precise to reflect the setting of linear regression.

The assumption of data in a low-dimensional subspace seems too strong. It is believable that the data lies in a low-dimensional *manifold*, but I don’t see any reason to restrict the data to a subspace (it is much stronger to let the manifold to be “straight”).

Questions:

As discussed in the beginning of Page 13, minimizing the MSE error itself may have the effect of regularizing the weights $||W||_F$. In addition, in the optimization problem (first equation of page 4) there is also an explicit minimization of $||W||_F$ after minimizing the MSE error. I am wondering: is it necessary to have the latter minimization (as in page 4)? What is the relation between these two types of regularization of weights? Or more precisely, does the regularization effect of noise (as discussed in page 13) really have any impact on the analysis, as the minimization of $||W||_F$ (in page 4) already had the effect?

What is the technical novelty of this paper when compared with the paper “Benign overfitting in linear regression” (by Bartlett et al)?

---

> ### Author Response · Authors · 2024-02-16
> **Response**
>
> We thank the reviewer for their feedback, comments, and questions.
>
> **Weaknesses**
>
> We agree with the reviewer that our work does not answer the question for non-linear data or non-linear models.
>
> We have updated the title to add the word Linear.
>
>
> We hope that it is clear from the paper now that we do not claim to solve the problem for non-linear data/models and that the limitations sections on page 4 are clear.
>
> Nevertheless, we believe that the setup being studied here is still a non-trivial step in the direction of being able to work with non-linear problems.
>
> **Questions**
>
> > norm minimization
>
> This is a great question, and we would like to make a few points.
>
> The first is that many standard algorithms, while only optimizing for MSE, actually solve the problem given on page 4. Specifically, there are two conventional methods. The first is to use the Moore-penrose pseudo-inverse. However, it is known that this results in the minimum norm solutions. The second is to use gradient descent. Then, for gradient descent, if we assume nice initialization (i.e., doesn't live in some low dimensional orthogonal subspace), then it converges to the Moore-Penrose solution.
>
> **Hence, it is reasonable to study the minimum norm solution**
>
> > noise regularization
>
> Additionally, the noise regularization has an impact! To see this, let us look at the expression in Theorem 1. Let us look at $c > 1$. The term on the second line comes from the $\|W_{opt}\|$. Specifically,
>
> $$ \|W_{opt}\|^2_F = \frac{c}{c-1}Tr(\beta_U \beta_U^T (I + \eta_{trn}^2\Sigma_{trn}^2)^{-1}) $$
>
> Here $\eta_{trn}$ controls the variance of the noise. Decreasing the variance reduces the regularization and increases the above quantity. On the other hand, increasing the variance increases the regularization and decreases the above quantity.
>
> **Hence, the noise regularization has an effect**
>
> > novelty compared to Bartlet et al.
>
> As detailed on page 3 in the section titled "Theoretical Work on Generalization in Non-Classical Regimes" and then again on page 5 when discussing the assumption on the data, *we have a different setup from the Bartlet et al. paper*.
> Specifically:
>
> 1) They look at output noise univariate regression. We look at the input noise version of the problem.
> 2) They have sub-gaussian data that is well conditioned. We have low dimensional data
> 3) They do not consider covariate shifts or transfer learning. We do.
>
> While similar, the settings of the problems are thus different.
>
> In terms of the technical tools used, we both use tools from high-dimensional statistics and random matrix theory. However, the specific techniques and their applications are different.

---

> > ### Comment · Reviewer_3dcE · 2024-03-05
> > **thanks for the response**
> >
> > I thank the authors for the reply and the update on the paper.
> >
> > However, the reply does not answer my questions.
> >
> > In Q1, I asked the necessirty of taking the argmin in Eq.(4). Because it states that minimizing the MSE loss itself already have the effect of reducing $||W||$. Hence, I don't see the point why there is a need to take the argmin after minimizing the loss.
> >
> > In Q2, I asked about the **technical novelty** (instead of the setting) of this paper, compared to Bartlet et al.. I would like to know how much do the techniques used in prior work apply to the setting of this paper.

---

> > > ### Author Response · Authors · 2024-03-06
> > > **Response**
> > >
> > > Thank you for the clarifications.
> > >
> > > -----
> > >
> > > **Q1**
> > >
> > > Yes, it is needed. Even with noise regularization, there are many solutions in the $c > 1$ case, and we need the norm minimization to pick a canonical solution. This is because we train one specific noise instance.
> > >
> > > For example, let $X = \begin{bmatrix} 1 \\\\ 1 \end{bmatrix}$. So $d = 2, n=1$, and  $c = 2$. Let $\beta = I$ and $Y = X$.
> > >
> > > Suppose the noise we received was $A_{trn} = \begin{bmatrix} 0.1 \\\\ -0.07 \end{bmatrix}$
> > >
> > > Then we solve for $W = \begin{bmatrix} w_{11} & w_{12} \\\\ w_{21} & w_{22} \end{bmatrix}$ that minimizes $\|\|Y - W(X+A_{trn})\|\|_F^2 = \left\|\left\|\begin{bmatrix} 1 \\\\ 1 \end{bmatrix} - W\begin{bmatrix} 1.1 \\\\ 0.93 \end{bmatrix} \right\|\right\|_F^2$.
> > >
> > > One possible solution is $W = \begin{bmatrix} 1/1.1 & 0 \\\\ 0 & 1/(0.93) \end{bmatrix}$, another solution is $W = \begin{bmatrix} 0 & 1/0.93 \\\\ 1/1.1 & 0\end{bmatrix}$.
> > >
> > > However, the minimum norm solution is $W = \frac{1}{\|\|X+A\|\|_2}Y(X+A)^T = \frac{1}{\sqrt{1.1^2 + 0.97^2}} \begin{bmatrix} 1.1 & 0.97 \\\\ 1.1 & 0.97 \end{bmatrix}$
> > >
> > > Notice that these solutions have very different properties! The first two are full rank, while the minimum norm solution is rank 1.
> > >
> > > Hence, they will have very different generalization properties. Also, note that the pseudo-inverse solution is the minimum norm solution, and that gradient descent (for ordinary least squares regression) converges to the minimum norm solution. Hence, this is a reasonable solution to use as the canonical solution.
> > >
> > > -----
> > >
> > > **Q2**
> > >
> > > We are not sure how to best answer Q2. We hope the following is helpful.
> > >
> > > First, the type of results are also different; they give high probability bounds on the risk. We give an asymptotically exact expression for the risk.
> > >
> > > Looking through the proof from Bartlett et al. 2019, they do their proof in 4 steps.
> > > First, they decompose the risk into two trace terms. From our proof sketch on page 8, this is roughly equivalent to the first displayed equation. However, even this is different. We have an exact expression, and they provide bounds (Lemma 7 in their paper).
> > >
> > > From here, the proofs diverge quite significantly. They mention that the heart of the proof is bounding the second trace term. To do this, they use concentration inequalities for sub-Gaussians and quadratic forms and show the concentration of the **whole spectrum**. That is, the spectrum is concentrated in a small interval. They then do some algebra to get the results, and *does not involve any random matrix theory*.
> > >
> > > For us, we get concentration as well. However, it is not the concentration of the spectrum but the concentration of the trace. We do this by using ideas from random matrix theory. That is, we show the distribution of the spectrum converges to a fixed distribution, and then we bound the deviation from that distribution. Then, we bound products of random variables. To the best of our understanding, this is not needed in Bartlett et al. 2019.
> > >
> > > Hence, we use **different techniques**.
> > >
> > > ----
> > >
> > > To answer the broader question about the techniques and prior work. Our proof techniques are similar to those of Sonthalia and Nadakuditi 2023 and Dobriban and Wagner 2015.
> > >
> > > Specifically, DW15 uses random matrix theory to answer questions in a similar setup to Bartlett et al. 2019. Sonthalia and Nadakuditi 2023 show how these techniques can be adapted to the input noise setting. We build on SN23, and while the proof structure remains the same, we need a variety of new concentration results and novel linear algebra results.
> > >
> > > -----
> > >
> > > Please let us know if there are more questions or concerns.

---

### Review · Reviewer_5Zxt · 2024-02-13

**Summary Of Contributions:**

This paper studies the data-dependent generalization bound for low-rank regression problem. While relaxing some conditions in the theorems, they also conduct numerical experiments of real data to verify the correctness of the theoretical results.

**Audience:**

Yes

**Claims And Evidence:**

Yes

**Requested Changes:**

Please see the weakness section. I also selected "No" in Claims and Evidence because the descriptions of the contributions are inaccurate.

**Strengths And Weaknesses:**

Weakness:
My main concern is the wording of the claims. In the abstract and the introduction, the authors are trying to claim that they obtain a very strong bound with relaxed conditions (even without iid assumption) and the results are also verified by real-data experiment, but never mention wordings like "data-dependent". When reading these parts, I was surprised at their result.

However, after reading the main theorems, I realized that they are able to relax some strong distributional conditions because they directly put the data matrix into the bound. It then becomes much less surprising: when people add distributional assumptions, they actually try to remove the data-dependent terms from their error bound. For example, in Hastie et. al. (2021), when deriving the asymptotic formula of tr(Xn^{\top}Xn) (Xn is the data matrix), they start from data-dependent representation tr(Xn^{\top}Xn).

The authors are strongly suggested to rewrite their abstract and introduction to help readers build a more concrete expectation on this paper.

Strengths:
While the weakness mentioned above makes the reading experience less pleasant, the paper still provides solid theoretical contributions. The derivations are non-trivial, and the assumptions are relaxed to the best. The technical sections are clear and easy to understand.

Hastie, Trevor, et al. "Surprises in high-dimensional ridgeless least squares interpolation." Annals of statistics 50.2 (2022): 949.

---

> ### Author Response · Authors · 2024-02-16
> **Response**
>
> We thank the reviewer for their comments, and feedback.
>
> > Weakness: My main concern is the wording of the claims. In the abstract and the introduction, the authors are trying to claim that they obtain a very strong bound with relaxed conditions (even without iid assumption) and the results are also verified by real-data experiment, but never mention wordings like "data-dependent". When reading these parts, I was surprised at their result.
>
> > ... The authors are strongly suggested to rewrite their abstract and introduction to help readers build a more concrete expectation on this paper.
>
> We are grateful to the reviewer for pointing this out. We agree with their concern and have edited the paper to highlight that we provide data-dependent, instance-specific bounds. The changes are highlighted in blue in the revised version.
>
> > Strengths: While the weakness mentioned above makes the reading experience less pleasant, the paper still provides solid theoretical contributions. The derivations are non-trivial, and the assumptions are relaxed to the best. The technical sections are clear and easy to understand.
>
> We thank the reviewer for their enthusiasm about the paper and its techniques.
>
> We hope our changes in the wording of the abstract and contributions aligns with their request. If they have any further concerns, we are happy to address them.

---

> > ### Comment · Reviewer_5Zxt · 2024-02-22
> >
> > I appreciate the authors addressing my concerns. The changes in the abstract and introduction look good to me.
> >
> > I went through the comments of other reviewers, and I agree that there is a gap between the settings considered in this paper and what is used in real practice. But I still value the theoretical contribution of this paper, as they provide detailed data-dependent bounds in the low-rank setting with non-trivial derivations, which can contribute to the area of double-descent related theories. (But I think this topic would be more suitable for Statistics or Data Science journals though.)

---

> > > ### Author Response · Authors · 2024-02-26
> > > **Thank you**
> > >
> > > Thank you again for the feedback, the paper is better for it. Also thank you for finding value in our theoretical contributions.

---

### Author Response · Authors · 2024-02-16
**General Response**

We thank the reviewers for their feedback. We have incorporated comments and uploaded a revision. The changes are highlighted in blue. If the reviewers have more concerns, please let us know and we are excited to engage with the reviewers.

---

### Decision · Action_Editor_Mj8F · 2024-03-15

**Recommendation:** Accept with minor revision

**Comment:**

The reviewers seem to agree that the theoretical contributions are solid and correct: [5Zxt: the paper still provides solid theoretical contributions. The derivations are non-trivial, and the assumptions are relaxed to the best], and that the theory is clear and easy to follow: [brba: The paper clearly presents the mathematical formulations of different problems and their required assumptions] [5Zxt: The technical sections are clear and easy to understand].

The main criticism was about the limitations of the framework (linear denoisers, etc). Reviewers [brba,5Zxt] requested the authors to make these limitations more explicit in the paper, especially in the abstract and introduction. The authors updated the paper accordingly. Ultimately, [5Zxt] and [3dcE] recommended to accept the paper, but [brba] was still in favor of rejection because of these limitations. To be honest, while I share the opinion that there are strong restrictions in the applicability of these results, one must also note that this is the case of most papers studying the theoretical aspects of benign overfitting: most are in the framework of linear regression or 2-layers neural network. The theory is of course far to be able to explain the double descent phenomenon in very complex models. Yet, these papers shine a new light on the tradeoffs in high-dimensional machine learning, and this paper extends on existing works in various relevant directions (non iid data, distribuction shift, etc.). Thus, I support its publication in TMLR.

Note that [brba] actually mentioned he/she would not oppose publication, on the condition that the authors would add more discussions on the limitations of their theoretical findings, I suggest to respect the reviewer's opinion here. The authors could maybe mention what would be desirable extensions of their results to deal with more realistic settings (non-linear denoisers, see also the comment by [3dcE] that the data might be in a low-dimensional manifold, not necessarily a subspace, etc.). Also, I note that the authors discuss extensively the existing literature on benign overfitting in the iid case, but they missed an earlier work on time-dependent observations: Nakakita and Imaizumi, Benign Overfitting in Time Series Linear Model with Over-Parameterization, https://arxiv.org/abs/2204.08369. The nature of the dependence seems different in Nakakita and Imaizumi's paper, so it should not remove any novelty from this submission, but this should be discussed by the authors. Thus, I will request a minor revision before publication.

**Audience:**

This is a theoretical contribution, that should be of interest for all researchers working on the new tradeoffs in machine learning theory.

**Claims And Evidence:**

The authors study linear denoising and regression with noisy inputs in the overparametrized setting. This setting has been widely studied in the past 2/3 years, notably to identify situations where apparent overfitting can still lead to good generalization ("benign overfitting"). However, the authors try to make the setting more realistic by 1) relaxing the iid assumption in the test and train set, and 2) allowing the train and test set to have different distribution (distribution shift). They leverage on a low-rank assumption on the data to prove generalization bounds that exhibit a double descent. The results of their numerical experiments match closely their theoretical findings.